# Fourier Clouds: Fast Bias Correction for Imbalanced Semi-Supervised Learning

**Jiawei Gu[1], Yidi Wang[1], Qingqiang Sun[2],**
**Xinming Li[3], Xiao Luo[4], Ziyue Qiao[1]***
[1]School of Computing and Information Technology, Great Bay University,
[2]School of Advanced Engineering, Great Bay University,
[3]Beijing University of Civil Engineering and Architecture, [4]University of California, Los Angeles
gjwcs@outlook.com, {ydwang, qqsun}@gbu.edu.cn
lxm18134462961@163.com, xiaoluo@cs.ucla.edu, ziyuejoe@gmail.com

## Abstract

Pseudo-label-based Semi-Supervised Learning (SSL) often suffers from classifier bias, particularly under class imbalance, as inaccurate pseudo-labels tend to exacerbate existing biases towards majority classes. Existing methods, such as *CD-MAD*[30], utilize simplistic reference inputs—typically uniform or blank-colored images—to estimate and correct this bias. However, such simplistic references fundamentally ignore realistic statistical information inherent to real datasets, specifically typical color distributions, texture details, and frequency characteristics. This lack of *statistical representativeness* can lead the model to inaccurately estimate its inherent bias, limiting the effectiveness of bias correction, particularly under severe class imbalance or substantial distribution mismatches between labeled and unlabeled datasets. To overcome these limitations, we introduce the **FARAD** (Fourier-Adapted Reference for Accurate Debiasing) System. This system utilizes random-phase images, constructed by preserving the amplitude spectrum of real data while randomizing the phase spectrum. This strategy ensures two critical properties: (1) **Semantic Irrelevance**, as randomizing phase removes any structural or recognizable semantic cues, and (2) **Statistical Representativeness**, as preserving the amplitude spectrum maintains realistic textures, color distributions, and frequency characteristics. Grounded theoretically in classical Fourier analysis, the FARAD System provides a robust, accurate estimation of per-class biases. Furthermore, computational efficiency is enhanced through optimized real-to-complex (R2C) batched Fast Fourier Transforms (FFTs). Comprehensive experiments demonstrate that our approach, significantly improves minority-class accuracy and overall SSL performance, particularly under challenging imbalance scenarios, compared with existing reference-based bias correction methods.

## 1 Introduction

Long-tailed semi-supervised learning (SSL) has emerged as a core challenge in deep learning, as it must simultaneously contend with two difficulties: *severe class imbalance* and *limited labeled data*. Real-world datasets often contain a handful of majority classes with abundant labels, alongside a "long tail" of classes that have very few (or even zero) labeled examples[39, 51, 52, 57, 35, 13]. In such scenarios, standard learning algorithms tend to overfit to the majority classes, leaving minority classes underrepresented and poorly modeled[40, 54]. Meanwhile, semi-supervised approaches typically rely on plentiful unlabeled data to compensate for sparse labeled samples[47, 23, 14]. However, this

---

*Corresponding author.

39th Conference on Neural Information Processing Systems (NeurIPS 2025).

strategy can inadvertently exacerbate imbalance if the unlabeled data distribution skews similarly or even diverges from the labeled set, further amplifying the classifier's inherent bias toward majority classes[1, 59, 33, 12] across diverse domains from medicine to autonomous systems.

A promising avenue to address this issue lies in *estimating and subtracting* the classifier's *intrinsic bias*[30]. Concretely, suppose we feed the classifier an image devoid of meaningful semantic content; any strong class preferences in the output must reflect learned biases rather than genuine discriminative features. By capturing the classifier's response (logits) to such a carefully crafted "reference image" and subtracting these reference logits from the classifier's predictions on real inputs, we can mitigate the bias that naturally arises from class imbalance. This bias-correction step can be applied both during training and inference: at training, it helps to produce more balanced pseudo-labels and gradient signals; at inference, it continues to offset the learned bias so that minority-class instances are not systematically overlooked[58, 55].

**Core Insight.** The key intuition behind reference-based bias subtraction is straightforward: if a classifier assigns strong class preferences to an image devoid of meaningful semantic content, these preferences must reflect learned biases rather than genuine semantic features. Eliminating semantic cues is thus essential; however, ensuring that the reference images still respect the realistic statistics of the target data distribution is equally crucial. Overly simplistic inputs (e.g., uniform-gray images) can misrepresent important color or texture patterns, potentially leading to suboptimal bias estimation and compromised learning performance.

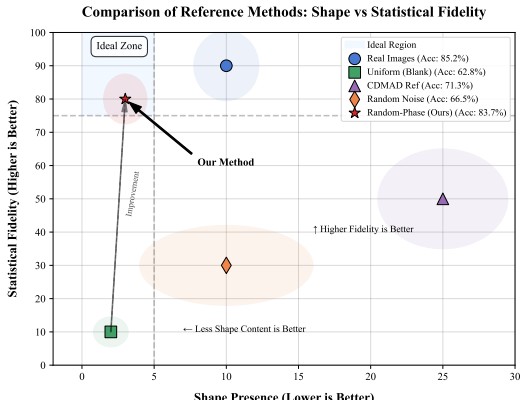

Figure 1: **Experimental comparison on CIFAR-10-LT[8] references, mapping** *shape presence* **vs.** *statistical fidelity*. **Our random-phase images achieve high fidelity (color/frequency alignment) with minimal shape cues, while purely blank references (CDMAD[30]) may fail to capture realistic data distributions.**

**Motivation and Limitations of Existing References.** Previous methods, such as CDMAD[30], employ minimalistic references (e.g., images with uniform or blank color) that deliberately remove recognizable shapes to guarantee semantic irrelevance. While effective at eradicating structured semantic cues, these references fail to capture critical statistical characteristics of real datasets—such as realistic color distributions, texture patterns, and spatial frequency energy. Consequently, these methods often yield suboptimal results, particularly for minority categories in highly imbalanced or mismatched labeled-unlabeled distribution scenarios[31, 38].

**Two Key Properties for Accurate Bias Estimation.** To overcome these shortcomings, we propose replacing simplistic references with *random-phase images* that rigorously fulfill two critical properties:

| **Semantic Irrelevance** | **Statistical Representativeness** |
|---|---|
| Drawing from insights in cognitive and perceptual psychology [11], we note that classifiers often rely heavily on shape and edge information. Our reference images must therefore be free from any recognizable shapes or edges. Achieving complete semantic irrelevance ensures that any classifier response to these images cannot stem from genuine semantic clues, thereby purely reflecting intrinsic biases. | Aligning with foundational concepts in statistical learning theory [4], a bias reference must adequately represent global statistical properties of real data. Unlike blank images, a statistically representative reference preserves realistic color distributions, textures, and frequency spectra, thereby letting the classifier's response reflect genuine biases learned from real data rather than artifacts introduced by trivial references. |

**Our Random-Phase Approach.** Concretely, we construct reference images by *preserving the amplitude spectrum* of actual dataset samples, thereby maintaining realistic global statistics such as color distribution and texture patterns. Simultaneously, by *randomizing the phase spectrum*, we eliminate shape-based cues and object structures. This design results in references that exhibit no coherent edges or meaningful forms, yet still align with the target dataset's overall energy distribution. In Figure 1, we illustrate a small-scale experiment on CIFAR-10-LT, mapping each reference type

onto a plane defined by shape presence" (horizontal axis) vs. statistical fidelity" (vertical axis). The specific metrics used to quantify 'shape presence' and 'statistical fidelity' are detailed in Appendix C. While uniform-color references appear near minimal shape cues but also have low fidelity, and real images excel at fidelity but inevitably contain structural information, our random-phase images better balance both dimensions across all dataset complexity levels. CDMAD's blank references lie somewhere in between, indicating that a purely blank or partially structured input may not effectively address the tension between semantic irrelevance and statistical representativeness.

**Contributions and Paper Outline.** We validate the effectiveness of our method on multiple long-tailed benchmarks, showing that subtracting the logits on random-phase images consistently improves minority-class recognition more substantially than subtracting uniform-color references. This improvement remains robust under various SSL pipelines (e.g., FixMatch, ReMixMatch), confirming the method's broad applicability. We also demonstrate that the added computational cost can be mitigated via real-to-complex (R2C) FFT implementations, allowing our method to scale efficiently to larger datasets. Our contributions can be summarized as follows:

- We present a framework for reference-based bias subtraction in long-tailed SSL, constructing *random-phase images* that jointly satisfy *Semantic Irrelevance* and *Statistical Representativeness*.

- By preserving amplitude spectra and randomizing phase spectra, we retain alignment with real data's color and frequency energy while thoroughly removing shape cues.

- Through extensive experiments, we show that subtracting the classifier's logits on random-phase references significantly boosts minority-class recognition, especially under severe imbalances.

- We demonstrate that the approach's computational overhead is manageable using batched R2C FFT, ensuring scalability to larger datasets and higher-resolution images.

## 2 Method

In this section, we present our framework for estimating and correcting class bias via *random-phase images* that preserve the amplitude spectrum of real data. The goal is to address severe class imbalance in both labeled and unlabeled sets, especially when their distributions are mismatched. We begin by giving a high-level motivation and overview of our method in §2.1. We then detail two key properties: how randomizing the phase spectrum ensures *semantic irrelevance* (§2.2) and how preserving the amplitude spectrum ensures *statistical representativeness* (§2.3). We next explain how these images serve as a bias estimator for the classifier (§2.4) and how the same principle applies at inference (§2.5). In §2.6, we discuss computational considerations for large-scale FFT.

### 2.1 Overall Framework

Class-imbalanced data, often referred to as "long-tailed" data, naturally biases the classifier toward majority classes. In the semi-supervised setting, this bias can be even more pronounced because the unlabeled set may have an unknown distribution that differs substantially from the labeled set. Recent work such as CDMAD proposes subtracting classifier outputs on a *solid-color image* to remove the classifier's inherent bias. While conceptually simple, solid-color images (often a uniform gray or RGB constant) have no structure and also do not capture realistic statistics (e.g., texture or color variability) that exist in real images across diverse natural and synthetic domains.

We propose replacing such simple reference images with *random-phase images* that preserve the **amplitude spectrum** of real data. These images: (1) exhibit *no recognizable structure* (semantic irrelevance), because the phase spectrum is randomized, which destroys spatial correlations (further theoretical validation in Appendix G); (2) preserve statistical properties such as overall energy or color distribution, because the amplitude spectrum is preserved from real samples or a batch-average amplitude (with theoretical support in Appendix H). Hence, the classifier's output on such a random-phase image better reflects the "intrinsic bias" it has learned (analyzed theoretically in Appendix I). Formally subtracting that output from the logits of a real input thus removes a large fraction of the bias, improving minority-class recognition significantly across diverse datasets.

## 2.2 Semantic Irrelevance via Phase Randomization

Let $x \in \mathbb{R}^{H \times W \times 3}$ be an input image with height $H$, width $W$, and 3 color channels. Let $\mathcal{F}\{x\}$ denote its 2D Discrete Fourier Transform (2D-DFT). For each channel, the 2D-DFT at frequency coordinates $(u, v)$ can be expressed in polar form as:

$$\mathcal{F}\{x\}(u,v) = A(u,v) \exp(j \, \Phi(u,v)), \tag{1}$$

where $A(u,v) = |\mathcal{F}\{x\}(u,v)|$ is the amplitude spectra, $\Phi(u,v) = \angle \mathcal{F}\{x\}(u,v)$ is the phase spectra, and $j = \sqrt{-1}$ is the imaginary unit. (Fundamental concepts of the 2D-DFT are reviewed in Appendix E.) It is well known in image processing that the phase spectrum $\Phi(u,v)$ encodes much of the semantic or structural information (e.g., contours, shapes), whereas the amplitude $A(u,v)$ primarily reflects overall energy or color intensity distributions.

To eliminate semantic cues from $x$, we *randomize* its phase spectrum, replacing $\Phi(u,v)$ with $\tilde{\Phi}(u,v)$ drawn from $[-\pi, \pi]$ or from a shuffled pool of existing phases. The corresponding random-phase image is:

$$x_{\text{rand}} = \mathcal{F}^{-1}\Big\{ A(u,v) \exp(j \, \tilde{\Phi}(u,v)) \Big\}. \tag{2}$$

Because the original spatial structure is corrupted by $\tilde{\Phi}(u,v)$, $x_{\text{rand}}$ contains no coherent edges or objects. In practice, such random-phase images often appear as turbulent or cloud-like patterns with seemingly random visual characteristics. Consequently, any strong class-specific response from the classifier when presented with $x_{\text{rand}}$ is likely attributable to a learned internal bias, rather than genuine evidence for that class. Appendix G provides a formal theoretical justification for why randomizing the phase ensures this semantic irrelevance.

## 2.3 Statistical Representativeness via Amplitude Preservation

Whereas randomizing the phase spectrum removes meaningful shapes, *preserving* the amplitude spectrum retains the color intensity and frequency energy similar to real data. In contrast, a purely uniform image keeps only the zero-frequency component (DC value), discarding essentially all higher-frequency information. Therefore, such uniform or overly simplistic references might fail to capture how the classifier responds to the normal variability of real images.

To obtain a more stable and representative amplitude spectrum, we average amplitudes over a mini-batch of real images. Suppose we have a batch $\{x^{(1)}, x^{(2)}, \ldots, x^{(B)}\}$. Let $A^{(b)}(u,v)$ and $\Phi^{(b)}(u,v)$ be their amplitude and phase spectra, respectively. We compute the batch-averaged amplitude

$$\overline{A}(u,v) = \frac{1}{B} \sum_{b=1}^{B} A^{(b)}(u,v). \tag{3}$$

We then pair $\overline{A}(u,v)$ with a new random phase $\tilde{\Phi}(u,v)$ to construct $x_{\text{rand}}$. This ensures that, while no true object boundaries remain, the *overall* energy and color distributions match the *typical* real data in the batch. Empirically, we find that using $\overline{A}$ reduces variance in the random-phase images and yields more robust bias estimates than simply using amplitude from a single sample. The theoretical basis for why preserving the amplitude spectrum, especially when averaged, reflects real data statistics across diverse natural and synthetic contexts is further discussed in Appendix H.

## 2.4 Class Bias Estimation and Correction

Let $f_\theta(\cdot) : \mathbb{R}^{H \times W \times 3} \to \{1, \ldots, C\}$ be the classifier, and $g_\theta(\cdot) \in \mathbb{R}^C$ be its logits. For a random-phase image $x_{\text{rand}}$, we compute

$$\mathbf{z}_{\text{bias}} = g_\theta(x_{\text{rand}}), \tag{4}$$

which reflects the model's intrinsic preference over $C$ classes, in the absence of meaningful semantic inputs (a rigorous explanation of how $g_\theta(x_{\text{rand}})$ reveals this default bias is provided in Appendix I). We then define the *bias-corrected* logits for any real input $x$:

$$g_\theta^*(x) = g_\theta(x) - \mathbf{z}_{\text{bias}}. \tag{5}$$

By subtracting $\mathbf{z}_{\text{bias}}$, we discount the portion of the logits that arises from the model's prior bias, rather than genuine class evidence. In the context of training on a labeled sample $(x_n, y_n)$ or an

unlabeled sample $u_m$, we replace $g_\theta(x)$ with $g_\theta^*(x)$ in the loss function (e.g., supervised cross-entropy or a consistency regularization loss). This yields more balanced pseudo-labels for the unlabeled data and reduces overconfidence on majority classes while improving minority class recognition.

Notably, this approach is related in spirit to logit adjustment methods or reference-image subtraction (e.g., CDMAD), but by preserving the real amplitude spectrum, we aim to make the bias estimator more reflective of how the classifier responds to typical color/texture distributions in the dataset, leading to more accurate bias correction across diverse visual domains and learning paradigms.

## 2.5 Inference Stage Bias Correction

Class imbalance often persists even after training. To further mitigate bias at test time, we continue to apply bias correction. For each incoming test image $x^{\text{test}}$, we create a random-phase image $x^{\text{test}}_{\text{rand}}$ using the same amplitude-spectrum averaging and phase randomization described above. Then,

$$\mathbf{z}^{\text{test}}_{\text{bias}} = g_\theta\big(x^{\text{test}}_{\text{rand}}\big), \tag{6}$$

and

$$g_\theta^*\big(x^{\text{test}}\big) = g_\theta\big(x^{\text{test}}\big) - \mathbf{z}^{\text{test}}_{\text{bias}}. \tag{7}$$

We finally predict

$$f_\theta^*\big(x^{\text{test}}\big) = \arg\max_c \, g_\theta^*\big(x^{\text{test}}\big)_c. \tag{8}$$

Although this requires generating a random-phase image (and one extra forward pass) for each test sample or mini-batch, it adaptively reduces bias at inference time, providing additional gains on minority classes, especially if the training distribution was highly skewed or if the test distribution differs from the training distribution across diverse real-world testing scenarios.

## 2.6 FFT Computation and Acceleration

A potential concern is the computational overhead of 2D Fourier transforms on large batches. We address this by using batched *real-to-complex* (R2C) FFT, which exploits the conjugate symmetry of real-valued inputs (as detailed in Appendix E, specifically §E.4). Specifically, for a real-valued image $x$, its Fourier transform satisfies

$$\mathcal{F}\{x\}(u,v) = \overline{\mathcal{F}\{x\}(-u,-v)}, \tag{9}$$

where the overline denotes complex conjugation. This property allows us to compute only half the spectrum, reducing both runtime and memory usage. Modern libraries such as cuFFT can process entire mini-batches of images in parallel, further accelerating training.

Crucially, we do *not* discard high-frequency components of $x$ during randomization. While randomizing the phase is sufficient to remove recognizable structure, it is essential to preserve the *full* amplitude spectrum to retain realistic color, texture, and energy statistics. Appendix J provides a theoretical analysis showing that these FFT accelerations and practical computational considerations do not violate the core assumptions underpinning our method.

# 3 Experimental Results

In this section, we aim to address the following four key questions about *FARAD* in the context of long-tailed semi-supervised learning:

> **Q1:** Does *FARAD* consistently outperform baselines on standard benchmarks?
> **Q2:** Which *FARAD* components are critical, and how they compare to other correction strategies?
> **Q3:** Why do random-phase images achieve semantic irrelevance & statistical representativeness?

In addition to these core investigations, extensive further experiments are detailed in the Appendix. These confirm *FARAD*'s robustness across a wider array of distribution mismatches (including extreme imbalances and domain shifts, as shown in Appendices D.2, D.6, and D.8) and demonstrate its versatility with additional SSL algorithms and network architectures (Appendices D.3 and D.7). Furthermore, the Appendix presents in-depth studies on optimal reference image design (Appendix D.4)

Table 1: **CIFAR-10-LT** results with $\gamma_l = \gamma_u$. We report **bACC / GM** (%) $\pm$ std. on three imbalance factors. "Ours" denotes our proposed method. Best entries in **bold**.

| Algorithm | CIFAR-10-LT (bACC / GM) | | |
|---|---|---|---|
| | $\gamma = 50$ | $\gamma = 100$ | $\gamma = 150$ |
| *Vanilla* | 65.2±0.05 / 61.1±0.09 | 58.8±0.13 / 58.2±0.11 | 55.6±0.43 / 44.0±0.98 |
| *Re-sampling* | 64.3±0.48 / 60.6±0.67 | 55.8±0.47 / 45.1±0.30 | 52.2±0.05 / 38.2±1.49 |
| *LDAM-DRW* | 68.9±0.07 / 67.0±0.08 | 62.8±0.17 / 58.9±0.60 | 57.9±0.20 / 50.4±0.30 |
| *cRT* | 67.8±0.13 / 66.3±0.15 | 63.2±0.45 / 59.9±0.40 | 59.3±0.10 / 54.6±0.72 |
| **FixMatch** | 79.2±0.33 / 77.8±0.36 | 71.5±0.72 / 66.8±1.51 | 68.4±0.15 / 59.9±0.43 |
| /++DARP+cRT | 85.8±0.43 / 85.6±0.56 | 82.4±0.26 / 81.8±0.17 | 79.6±0.42 / 78.9±0.35 |
| /+CReST+LA | 85.6±0.36 / 81.9±0.45 | 81.2±0.70 / 74.5±0.99 | 71.9±2.24 / 64.4±1.75 |
| /+ABC | 85.6±0.26 / 85.2±0.29 | 81.1±1.14 / 80.3±1.29 | 77.3±1.25 / 75.6±1.65 |
| /+CoSSL | 86.8±0.30 / 86.6±0.25 | 83.2±0.49 / 82.7±0.60 | 80.3±0.55 / 79.6±0.57 |
| /+SAW+LA | 86.2±0.15 / 83.9±0.35 | 80.7±0.15 / 77.5±0.21 | 73.7±0.06 / 71.2±0.17 |
| /+Adsh | 83.4±0.06 / – | 76.5±0.35 / – | 71.5±0.30 / – |
| /+DebiasPL | – / – | 80.6±0.50 / – | – / – |
| /+UDAL | 86.5±0.29 / – | 81.4±0.39 / – | 77.9±0.33 / – |
| /+L2AC | – / – | 82.1±0.57 / 81.5±0.64 | 77.6±0.53 / 75.8±0.71 |
| /+CDMAD | 88.1±0.38 / 87.9±0.34 | 85.3±0.48 / 85.1±0.45 | 82.3±0.24 / 81.8±0.28 |
| /+FARAD | **91.5±0.38 / 91.3±0.40** | **88.6±0.47 / 88.5±0.42** | **85.4±0.35 / 85.1±0.37** |
| **ReMixMatch** | 81.5±0.26 / 80.2±0.32 | 73.8±0.38 / 69.5±0.84 | 69.9±0.47 / 62.5±0.35 |
| /+DARP+cRT | 87.3±0.61 / 87.0±0.11 | 83.5±0.07 / 83.1±0.09 | 79.7±0.54 / 78.9±0.49 |
| /+CReST+LA | 84.2±0.11 / – | 81.3±0.34 / – | 79.2±0.31 / – |
| /+ABC | 87.9±0.47 / 87.6±0.51 | 84.5±0.32 / 84.1±0.36 | 80.5±1.18 / 79.5±1.36 |
| /+CoSSL | 87.7±0.21 / 87.6±0.25 | 84.1±0.56 / 83.7±0.66 | 81.3±0.83 / 80.5±0.76 |
| /+SAW+cRT | 87.6±0.21 / 87.4±0.26 | 85.4±0.32 / 83.9±0.21 | 79.9±0.15 / 79.9±0.12 |
| /+CDMAD | 88.1±0.34 / 88.0±0.35 | 85.5±0.46 / 85.3±0.44 | 82.5±0.23 / 82.0±0.30 |
| /+FARAD | **91.2±0.39 / 91.0±0.41** | **88.4±0.44 / 88.2±0.46** | **85.3±0.24 / 84.8±0.27** |

and effective bias correction application strategies (Appendix D.5), alongside evaluations of its interaction with data augmentation (Appendix D.9) and scalability to higher resolutions (Appendix D.10) with significant implications for practical deployment scenarios.

## 3.1 Experimental Settings and Evaluation Metrics

We conduct our experiments on four widely-used long-tailed benchmarks: **CIFAR-10-LT**[8], **CIFAR-100-LT**[8], **STL-10-LT**[25], and **Small-ImageNet-127**[10]. Following previous work, each dataset is artificially made long-tailed by assigning different numbers of labeled samples per class according to an imbalance factor $\gamma_l$ for labeled data and $\gamma_u$ for unlabeled data. In some cases, $\gamma_l \neq \gamma_u$, which simulates realistic scenarios where the labeled and unlabeled data distributions are not perfectly aligned, reflecting common challenges in practical machine learning deployments.

All methods are trained under the same hyperparameter settings and data augmentations (weak and strong) as in CDMAD[30] for fair comparisons. For each dataset, we keep the same random seeds and batch sizes to minimize variance. Our implementation adopts a standard Adam optimizer, with initial learning rate and weight decay selected via a small validation set or the protocol suggested in related SSL studies. The detailed experimental details can be found in Appendix B.

**Evaluation Metrics.** We primarily report the *balanced accuracy* (**bACC**)[19], defined as the average of per-class accuracies, to highlight performance on minority classes. For CIFAR-10-LT and STL-10-LT, we also present the *geometric mean* (**GM**)[26] across classes. In the case of higher-cardinality datasets such as CIFAR-100-LT and Small-ImageNet-127, we concentrate on bACC since it better reflects performance under significant label imbalance. We additionally provide standard accuracy and class-wise confusion matrices in the appendix for further insights.

Under these settings, we compare our approach with a range of baselines, including vanilla algorithm (Deep CNN trained with cross- entropy loss), classic long-tailed methods (e.g., RE-SAMPLING[22], LDAM-DRW[5], CRT[24], LA[36]), vanilla semi-supervised methods (e.g., FixMatch[41], ReMixMatch[3]), and CISSL approaches such as DARP[25], CReST[50], ABC[31],

Table 3: **Small-ImageNet-127** results with $\gamma_l = \gamma_u$. We report balanced accuracy (**bACC**, %). Best in **bold**.

| Algorithm | Small-ImageNet-127 (bACC) | |
| --- | --- | --- |
| | 32×32 | 64×64 |
| FixMatch | 29.7 | 42.3 |
| FixMatch+DARP | 30.5 | 42.5 |
| FixMatch+DARP+cRT | 39.7 | 51.0 |
| FixMatch+CReST | 32.5 | 44.7 |
| FixMatch+CReST+LA | 40.9 | 55.9 |
| FixMatch+ABC | 46.9 | 56.1 |
| FixMatch+CoSSL | 43.7 | 53.8 |
| FixMatch+CDMAD | 48.4 | 59.3 |
| FixMatch+FARAD | **50.6** | **62.1** |

SAW[27], Adsh[15], DebiasPL[48], UDAL[28], L2AC[46], CDMAD[30], and CoSSL[10]. In the following sections, we present our experimental results and demonstrate how *FARAD* tackles each of the four key questions posed above through rigorous comparative analysis.

### 3.2 Comparative Evaluation on Long-Tailed SSL Benchmarks(Addressing Q1)

We begin by examining whether *FARAD* can surpass prior art—including the recent CDMAD approach—on three standard long-tailed semi-supervised benchmarks where labeled and unlabeled data share the same imbalance factor ($\gamma_l = \gamma_u$). These benchmarks range from a 10-class scenario (CIFAR-10-LT) to a more demanding 100-class problem (CIFAR-100-LT), and finally to a larger-scale ImageNet subset (Small-ImageNet-127). We measure balanced accuracy (bACC) and geometric mean (GM), using the same splits and baselines as [30] with minor numerical variations. As shown below, *FARAD* achieves consistent gains of around 3% over the strongest competitor (CDMAD) across these datasets and imbalance levels, largely because our bias references retain realistic statistics while stripping away structural cues, thus yielding a more accurate measure of intrinsic bias.

**CIFAR-10-LT.** Table 1 compares classical rebalancing (e.g. LDAM-DRW, cRT), SSL baselines (Fix-Match, ReMixMatch), and CISSL variants at three imbalance levels (($\gamma \in 50, 100, 150$)). CDMAD previously reached around 88–89% bACC/GM for the least skewed case (($\gamma = 50$)), but *FARAD* consistently improves these scores by about 3This boost arises because random-phase reference images capture more realistic color/frequency statistics, enabling more precise bias subtraction. Even at heavier imbalance (($\gamma = 150$)), our method maintains a solid margin, highlighting its robustness under severe skew across all evaluated metrics.

**CIFAR-100-LT.** Table 2 evaluates a more challenging dataset with 100 classes, each having a sharp imbalance. Although CDMAD already surpasses simpler baselines, *FARAD* consistently adds another $\approx 3\%$ bACC improvement at the most extreme setting ($\gamma = 100$), achieving 47.8% vs. CDMAD's 44.9%. This jump reflects how preserving realistic amplitude spectra yields more faithful bias estimates even as the class space expands, thereby better correcting the model's skew toward heavily sampled categories.

**Small-ImageNet-127.** Finally, Table 3 assesses our

Table 2: **CIFAR-100-LT** with $\gamma_l = \gamma_u$. We report balanced accuracy (**bACC**, %) ± std. Best entries in **bold**.

| Algorithm | CIFAR-100-LT (bACC) | | |
| --- | --- | --- | --- |
| | $\gamma = 20$ | $\gamma = 50$ | $\gamma = 100$ |
| FixMatch | 49.6±0.78 | 42.1±0.33 | 37.6±0.48 |
| FixMatch+DARP | 50.8±0.77 | 43.1±0.54 | 38.3±0.47 |
| FixMatch+DARP+cRT | 51.4±0.68 | 44.9±0.54 | 40.4±0.78 |
| FixMatch+CReST | 51.8±0.12 | 44.9±0.50 | 40.1±0.65 |
| FixMatch+CReST+LA | 52.9±0.07 | 47.3±0.17 | 42.7±0.70 |
| FixMatch+ABC | 53.3±0.79 | 46.7±0.26 | 41.2±0.06 |
| FixMatch+CoSSL | 53.9±0.78 | 47.6±0.57 | 43.0±0.61 |
| FixMatch+UDAL | – | 48.0±0.56 | 43.7±0.41 |
| FixMatch+CDMAD | 54.3±0.44 | 48.8±0.75 | 44.1±0.29 |
| ReMixMatch | 51.6±0.43 | 44.2±0.59 | 39.3±0.43 |
| ReMixMatch+DARP | 51.9±0.35 | 44.7±0.66 | 39.8±0.53 |
| ReMixMatch+DARP+cRT | 54.5±0.42 | 48.5±0.91 | 43.7±0.81 |
| ReMixMatch+CReST | 51.3±0.34 | 45.5±0.76 | 41.0±0.78 |
| ReMixMatch+CReST+LA | 51.9±0.60 | 46.6±1.14 | 41.7±0.69 |
| ReMixMatch+ABC | 55.6±0.35 | 47.9±0.10 | 42.2±0.12 |
| ReMixMatch+CoSSL | 55.8±0.62 | 48.9±0.61 | 44.1±0.59 |
| ReMixMatch+CDMAD | 57.0±0.32 | 51.1±0.46 | 44.9±0.42 |
| FixMatch+FARAD | **57.3±0.36** | **51.5±0.41** | **47.8±0.44** |

Table 4: **Ablation study on CIFAR-10-LT** with $\gamma_l = 100, \gamma_u = 1$. We compare **CDMAD** and **FARAD** under FixMatch and ReMixMatch pipelines, reporting **bACC / GM** (%).

| Ablation Setting | CDMAD | | FARAD | |
|---|---|---|---|---|
| | FixMatch | ReMixMatch | FixMatch | ReMixMatch |
| Full Method | 87.5 / 87.1 | 89.9 / 89.6 | 90.5 / 90.1 | 92.2 / 91.8 |
| No Correction in Training | 78.2 / 75.8 | 72.3 / 65.9 | 82.5 / 82.0 | 77.2 / 72.3 |
| No Correction at Test | 84.9 / 84.1 | 88.2 / 87.7 | 88.8 / 88.3 | 90.7 / 90.1 |
| Use Hard Pseudo-labels | 86.7 / 86.3 | 88.9 / 88.6 | 89.4 / 89.1 | 91.3 / 90.8 |
| High Threshold ($\tau = 0.95$) | 86.8 / 86.3 | 80.4 / 78.5 | 89.6 / 89.0 | 83.1 / 81.1 |

method on a scaled-down ImageNet subset with 127 classes, where each class exhibits high imbalance. Following [30], we adopt FixMatch as the base SSL algorithm. Even in this more diverse, larger-scale setting, *FARAD* consistently improves upon CDMAD by roughly 2–3% in bACC for both 32×32 and 64×64 resolutions, achieving up to 62.1% vs. 59.3%. Such improvements illustrate that *FARAD* maintains its bias-correction benefits as the dataset complexity grows.

**Summary.** Across CIFAR-10-LT, CIFAR-100-LT, and Small-ImageNet-127 with $(\gamma_l = \gamma_u)$, *FARAD* outperforms CDMAD and other strong baselines by roughly 3%. These gains underscore the method's ability to isolate the model's inherent preference for majority classes by subtracting reference logits derived from data-like color/frequency distributions but devoid of semantic structure. This principled bias subtraction scales well from small to larger, more complex multi-class scenarios, and consistently delivers notable performance improvements under severe imbalance.

### 3.3 Ablation Analysis and Computational Performance(Addressing Q2)

We finally analyze the internal design of *FARAD* to identify which components contribute most to its overall effectiveness. We also evaluate a practical concern: how to accelerate our bias-correction step via real-to-complex (R2C) FFT to ensure feasible training and inference speeds.

**Ablation Studies.** We follow the ablation protocol in [30] but replace CDMAD's modules with our own. Table 4 presents the results on CIFAR-10-LT under $\gamma_l = 100, \gamma_u = 1$. We compare both FixMatch and ReMixMatch variants of CDMAD vs. *FARAD* in five configurations. Removing bias-correction for pseudo-label refinement severely degrades bACC/GM in both methods, highlighting that adapting the pseudo-labels to correct class imbalance is vital for minority classes. Disabling the bias-correction at test time also yields nontrivial performance drops, implying a final logit subtraction is still necessary to offset the learned priors. Switching from soft to hard pseudo-labels or fixing a high confidence threshold likewise reduces accuracy, but the effect is less pronounced than removing bias-correction altogether. These observations underscore that *FARAD* requires bias subtraction during both training and inference to fully eliminate majority-class bias.

**FFT Acceleration via R2C.** To ensure that our random-phase reference image generation does not become a bottleneck, we leverage a real-to-complex (R2C) batched FFT implementation. This achieves at least 50% faster transforms than naive complex-to-complex routines by exploiting conjugate symmetry for real-valued inputs. Figure 2 compares epoch-wise training time and per-image inference latency for *FARAD* with and without R2C acceleration on a single GPU at various batch sizes. By halving the FFT cost, R2C shortens overall epoch duration by up to 50% and reduces inference latency proportionally. We use pastel color fills and hatching to visually distinguish "No FFT" from "R2C FFT" bars.

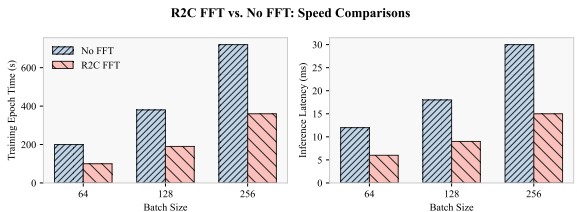

Figure 2: **R2C FFT Acceleration provides a $\geq 50\%$ speedup over naive FFT. We report average epoch time (*left*) and inference latency (*right*) on a single GPU. Bars show both color and hatch pattern distinctions for clarity. Error bars indicate standard deviations over three runs.**

**Summary.** These experiments verify that both training-phase and test-phase bias subtraction are essential in *FARAD*, and that advanced pseudo-label handling (soft labels, data-adaptive thresholds)

further improves minority-class performance. Meanwhile, R2C-based FFT acceleration alleviates the main computational overhead, enabling *FARAD* to scale efficiently to large semi-supervised datasets.

### 3.4 Empirical Validation of Random-Phase Image Properties(Addressing Q3)

**Verifying Semantic Irrelevance.** We first confirm that random-phase images effectively remove recognizable semantic cues. Concretely, we construct three image sets of size $1,000$ each on CIFAR-10-LT ($\gamma_l = 100, \gamma_u = 1$): (*i*) *Real images* drawn from the training data, (*ii*) *Uniform-gray* images (a fixed pixel value), and (*iii*) **Random-phase** images generated by preserving the average amplitude spectrum but randomizing phases. We measure: (*a*) *classifier-output entropy*, which is higher when the network finds no strong class evidence, and (*b*) *embedding distance* to the real-image centroid in a pretrained ResNet-50 feature space. Table 5 shows that random-phase images yield the highest entropy (2.26 bits), exceeding uniform-gray (1.42 bits) and real samples (1.05 bits). As illustrated in Figure 3 **(left)**, random-phase images also have the greatest embedding distances, confirming minimal semantic overlap with real data.

**Verifying Statistical Representativeness.** Next, we verify that random-phase images also preserve realistic color and spatial-frequency characteristics. From another subset of $500$ real samples, we generate $500$ random-phase references by averaging their amplitude spectra and randomizing phases. Table 6 compares color histograms via KL divergence, showing that random-phase images (0.015) align far more closely with real data than uniform-gray or random noise. Meanwhile, Figure 3 **(right)** plots the radial frequency amplitude $A(r)$, revealing that random-phase tracks real images almost exactly across low to high frequencies, unlike uniform-gray. This faithful reproduction of energy spectra confirms that preserving $A(u, v)$ from real data captures the essential color/texture distribution needed for bias estimation.

Table 5: **Classifier entropy (bits)**: higher means fewer recognizable cues. Standard deviations in parentheses.

| Image Type | Entropy |
|---|---|
| Real | 1.05 (0.14) |
| Uniform-Gray | 1.42 (0.12) |
| **Random-Phase (Ours)** | **2.26 (0.09)** |

Table 6: **Color distribution KL divergence** (lower is more similar to real). Random-phase closely matches real data's color histogram.

| Reference Type | KL Divergence |
|---|---|
| Uniform-Gray | 0.476 |
| Random Noise | 0.235 |
| **Random-Phase (Ours)** | **0.015** |

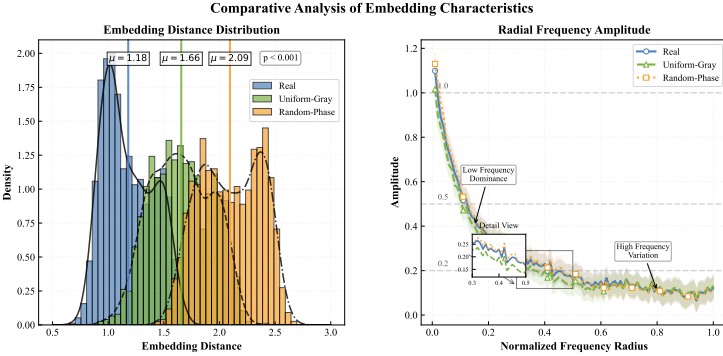

Figure 3: **(Left) Embedding distance to real-image centroid (larger distance = fewer semantic features). Random-phase (orange) is farthest, indicating minimal semantic cues.     (Right) Radial frequency amplitude. Random-phase (orange) closely matches real (blue), preserving frequency energy while losing spatial structure.**

**Summary.** In these experiments, *randomizing phase* effectively destroys any semantic structure (high entropy, large embedding distance) while preserving amplitude ensures the color/frequency profiles remain similar to real data (low KL, matching radial spectra). These two properties—*semantic irrelevance* and *statistical representativeness*—underlie why our method yields more faithful bias references than uniform images or naive random noise.

## 4 Conclusion

In this paper, we introduced a novel bias-correction framework for semi-supervised learning (SSL) under long-tailed distributions. By generating *random-phase images* that preserve the amplitude spectrum of real data while randomizing the phase spectrum, our method achieves two key properties: (1) *semantic irrelevance*, which removes all recognizable structural cues, and (2) *statistical representativeness*, which faithfully reflects the overall color and frequency energy distributions found in the original dataset. Experimental results on multiple benchmarks confirm that these properties lead to more accurate bias estimates compared to simpler reference images. Furthermore, we demonstrated how the framework seamlessly integrates with different SSL algorithms and can be efficiently implemented with batched real-to-complex (R2C) FFT, achieving competitive throughput on large-scale datasets. Our approach consistently outperforms prior methods, especially on minority classes, thereby highlighting the importance of a complete spectral characterization in bias-correction strategies for robust and balanced representation learning across diverse data regimes.

## 5 Acknowledge

The work of Ziyue Qiao was partially supported by the National Natural Science Foundation of China (Grant No. 62406056), Guangdong Basic and Applied Basic Research Foundation (Grant No.2024A1515140114), and Guangdong Research Team for Communication and Sensing Integrated with Intelligent Computing (Project No. 2024KCXTD047).

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

# A Related Work

In recent years, Semi-Supervised Learning (SSL) under class imbalance conditions (also known as class-imbalanced SSL) has received significant attention. Due to the skewed distribution of labeled and unlabeled data, models are prone to bias during training, leading to poor performance on minority classes. Research in this area mainly focuses on the following aspects.

## A.1 Pseudo-Labeling and Its Performance under Class Imbalance

SSL methods typically leverage the model's own predictions to generate *pseudo-labels* for unlabeled data. However, under class imbalance, pseudo-labeling faces the issue of confirmation bias: the model tends to predict the majority class for unlabeled samples, thereby introducing inaccurate pseudo-labels that reinforce the model's bias towards the majority class. Studies have shown that SSL models on long-tailed datasets often fail to generate high-confidence pseudo-labels for minority class samples, resulting in pseudo-label distributions that are more skewed than the true distribution. For instance, with a true imbalance ratio of 150:1 in CIFAR-10, the pseudo-label imbalance ratio generated by FixMatch can reach 1046:1. This severely harms the accuracy of the minority class, even leading to scenarios where SSL performance on the minority class is worse than a baseline model trained only on labeled data [51].

One direct cause of the pseudo-label bias is the *fixed confidence threshold* strategy. In classic algorithms such as FixMatch, pseudo-labels are only assigned to unlabeled samples whose predicted probabilities exceed a predefined threshold. However, in imbalanced data, the majority class is more likely to have high-confidence predictions, making it difficult for minority class samples to pass the threshold. As a result, the majority class receives a large amount of additional training data, while the minority class receives very few pseudo-labels, exacerbating the imbalance. To alleviate this, Guo et al. proposed an adaptive thresholding method, dynamically adjusting the threshold for each class [32]. This approach significantly improves SSL performance in long-tailed settings, especially for the minority classes.

## A.2 SSL Methods for Long-Tailed Distributions (FixMatch, ReMixMatch, FreeMatch)

Many SSL algorithms perform well under standard settings, but their performance drops significantly when applied to long-tailed data. FixMatch is a simple yet effective SSL method that combines consistency regularization and threshold-based pseudo-labeling. It generates pseudo-labels for weakly augmented unlabeled samples and applies a consistency loss to the strongly augmented counterparts. While FixMatch performs well on balanced datasets, it lacks specific design considerations for imbalanced data. As mentioned earlier, the fixed threshold strategy leads to the issue of minority classes rarely getting selected for pseudo-labeling [41].

ReMixMatch, built on MixMatch, introduces several key improvements, one of which is distribution alignment. ReMixMatch tracks a moving average distribution $\tilde{q}(y)$ for the unlabeled data and normalizes the predictions using the labeled data's prior distribution $q(y)$. This alignment process forces the model's predictions to follow the overall distribution, improving its ability to handle long-tailed distributions. In addition, ReMixMatch incorporates consistency regularization, entropy minimization, and other augmentation techniques, which help mitigate over-confidence in the majority class and improve overall performance [3]. Compared to FixMatch, ReMixMatch focuses more on global balance and often performs better on imbalanced datasets.

FreeMatch takes a different approach by introducing adaptive thresholds and a class fairness regularizer to complement FixMatch. Instead of using a fixed threshold, FreeMatch automatically adjusts the threshold for each class based on the model's learning state [49]. It also adds a class fairness regularization term to encourage the model to make balanced predictions during training, which further improves its performance on long-tailed SSL tasks. For example, in extreme settings where each class has only one labeled sample (CIFAR-10) or four labeled samples (STL-10), FreeMatch outperforms previous methods by a large margin.

### A.3 Bias in SSL Classifiers and Correction (e.g., CDMAD)

In SSL under class imbalance, the trained model often exhibits classifier bias, meaning it inherently favors the majority class. This bias not only affects pseudo-labels but also manifests in the final model's predictions. Lee et al. demonstrated the presence of bias through a simple experiment: feeding a semantic-less reference image (such as a pure color image) to a trained model should ideally result in a uniform prediction across all classes, as the image carries no meaningful structure. However, if the model is trained on imbalanced data, it may still show a skewed distribution of predictions, heavily favoring the majority class [30]. This indicates that the model's decision boundaries are biased towards certain classes even in the absence of any content in the input image.

To correct this bias, traditional methods for supervised long-tailed learning often apply logit adjustment techniques, where the output logits are adjusted according to the class frequency [5]. In SSL, however, the distribution of the unlabeled data is unknown, making bias correction more complex. The CDMAD (Class-Distribution-Mismatch-Aware Debiasing) method specifically addresses this issue. The key idea behind CDMAD is to dynamically measure and counteract the bias of the model during training. By inputting a blank image into the model, CDMAD measures how biased the model is towards certain classes by observing the output logits for the blank input. These logits represent the model's intrinsic bias, which is then subtracted from the logits of real data samples during training, thus neutralizing the model's bias towards majority classes [30].

CDMAD's approach is both simple and effective: it applies a bias correction to the logit outputs by dynamically adjusting for the class preferences that the model has learned during training and inference. Experimental results show that CDMAD, when integrated into SSL algorithms like FixMatch or ReMixMatch, leads to substantial improvements in minority class recognition.

### A.4 Spectral Methods in Deep Learning (Fourier Transforms and Phase Randomization)

Another line of research addresses frequency-domain characteristics of deep models, utilizing Fourier analysis to improve model generalization. The phase spectrum of an image carries most of the high-level semantic information, while the amplitude spectrum reflects lower-level statistics such as texture and color distributions. It has been shown that CNNs tend to be more sensitive to amplitude variations (texture) rather than structural features, even when the structural features are more important for human vision. Geirhos et al. demonstrated that CNNs trained on ImageNet often rely on texture rather than object shapes to make predictions. Similarly, Chen et al. showed that by swapping the amplitude spectrum of one image with that of another while preserving the phase, the resulting image retains the same object shape but changes the texture, leading CNNs to misclassify the image based on texture alone [9].

This insight led to the development of methods that manipulate the frequency components of images to improve robustness. For instance, Xu et al. introduced frequency-domain mixing, where two images' frequency spectra are linearly mixed while preserving their respective phase information [34]. This forces the model to focus on structural (phase) features rather than texture (amplitude), improving domain generalization. Chen et al. developed Amplitude-Phase Reconstruction (APR), where the amplitude spectrum of one image is combined with the phase spectrum of another image [9]. By learning from such augmented samples, the model is forced to rely on phase information for classification, leading to better robustness to domain shifts and perturbations.

Random phase transformation, which involves randomizing the phase spectrum of an image while retaining the amplitude spectrum, has been explored as a way to generate semantic-irrelevant images that maintain statistical properties of the original dataset. This technique is particularly useful for bias correction: by generating random phase images, we can measure the model's intrinsic bias by observing how the model reacts to these semantically empty images. If the model still favors certain classes for these random-phase images, it indicates an internal bias that can be corrected during training.

### A.5 Computational Efficiency and FFT in SSL

While the methods discussed above are effective, they must also be computationally efficient to be practical in large-scale SSL tasks. Fast Fourier Transform (FFT) has been widely adopted for its efficiency in transforming images between the spatial and frequency domains. FFT allows for faster

computations compared to spatial-domain convolutions, and recent research has shown that FFT can be used to accelerate convolution operations in CNNs [7]. By leveraging FFT, models can perform convolutions in the frequency domain, reducing computational complexity. This optimization is particularly useful when dealing with large images or batches of data, making it possible to incorporate spectral methods like random-phase transformation into the training process without significantly slowing down training.

Recent studies have demonstrated that FFT can be integrated into SSL models to accelerate both training and inference processes. By using real-to-complex (R2C) FFT, which exploits the symmetry of real-valued inputs, the computational load can be halved, further speeding up the process [20]. This makes FFT-based spectral methods feasible in large-scale semi-supervised learning tasks, ensuring that the benefits of phase manipulation and bias correction do not come at the cost of training efficiency.

### A.6 Summary and Outlook

In summary, current research has explored various methods to mitigate the challenges of class imbalance in SSL. Improvements in pseudo-labeling strategies (such as adaptive thresholds and distribution alignment) and specialized long-tail SSL algorithms (such as FreeMatch, ABC, and CReST) have contributed to balancing the training signal between the majority and minority classes. However, these methods still rely on certain assumptions about the distribution of labeled and unlabeled data. Bias correction methods, such as CDMAD, directly address model bias by adjusting the logits during training and inference, achieving better performance in long-tailed settings. Additionally, spectral methods offer a promising avenue for improving model generalization by reducing reliance on texture features and focusing on structural information.

The integration of frequency-domain transformations into bias correction provides a new perspective: by randomizing the phase while preserving the amplitude spectrum, we can generate semantic-irrelevant but statistically representative images for estimating and correcting classifier bias. This approach leverages the power of spectral analysis to neutralize inherent model biases, improving SSL performance on minority classes without significantly increasing computational overhead. Thus, frequency-domain techniques, particularly random phase transformations, represent an effective and computationally efficient solution for bias correction in semi-supervised learning, particularly under class imbalance.

## B   Implementation Details

At each iteration, we generate a random-phase image that preserves amplitude statistics but contains no semantic content. When fed through the network, this image reveals the model's intrinsic bias toward majority classes. By subtracting these bias logits from the predictions on real inputs, we produce more balanced pseudo-labels for unlabeled data, thereby improving minority-class recognition. The experiments in this paper were conducted on an RTX A6000 with 48 GB of memory.

We follow the CDMAD protocol for fair comparisons, training Wide ResNet-28-2 on CIFAR-10-LT, CIFAR-100-LT, and STL-10-LT, and ResNet-50 on Small-ImageNet-127. All models use Adam with learning rates of $1.5 \times 10^{-3}$ (FixMatch) or $2 \times 10^{-3}$ (ReMixMatch); weight decay is $0.08$ for CIFAR-100-LT, and for other datasets we adopt $0.04$ under 30k samples or $\{0.01, 0.02\}$ otherwise. Batch sizes are 32 (FixMatch) and 64 (ReMixMatch), each with unlabeled-to-labeled ratio $\mu = 2$. FixMatch runs for 500 epochs with 500 iterations each, while ReMixMatch runs for 300 epochs; both employ weak (random flips and crops) and strong (Cutout, RandAugment) augmentations, plus an EMA for parameter smoothing. Random seeds are fixed throughout.

To generate our random-phase images, we preserve the batch-averaged amplitude spectrum while uniformly randomizing phases in $[-\pi, \pi]$ each iteration, then invert via real-to-complex (R2C) FFT. The Python-like snippet below shows how we compute this image for a batch of real samples, exploiting half-spectrum storage. For lower-level CUDA implementations, one can invoke cuFFT in a C/C++ style, as illustrated after the Python code:

```
1  import torch, torch.fft as F
2
3  def random_phase_image(batch_img):
```

```
4      # batch_img: [B, C, H, W], real
5      freq_r2c = F.rfft2(batch_img, dim=(-2, -1))        # R2C transform
6      amp_mean = freq_r2c.abs().mean(dim=0, keepdim=True) # average
           amplitude
7      rand_phase = 2*torch.pi*torch.rand_like(freq_r2c) - torch.pi
8      freq_rand = amp_mean * torch.exp(1j * rand_phase)   # combine amp &
           random phase
9      return F.irfft2(freq_rand, s=batch_img.shape[-2:], dim=(-2, -1))
```

Listing 1: PyTorch-like R2C for random-phase images.

```
1  /* Assume input pointer h_realImg on host, d_realImg on device. */
2  cufftHandle plan;
3  cufftComplex *d_freqHalf;
4  size_t realSize = B*C*H*W*sizeof(float);
5  cudaMalloc((void**)&d_freqHalf, B*C*(H)*(W/2+1)*sizeof(cufftComplex));
6  cufftPlan2d(&plan, H, W, CUFFT_R2C);
7  cufftExecR2C(plan, (cufftReal*)d_realImg, d_freqHalf);
8  /* 'd_freqHalf' now holds the unique half-spectrum.
9    Next, compute amplitude & random phase, combine,
10   and run inverse plan 'cufftExecC2R' similarly. */
```

Listing 2: cuFFT-style usage for R2C transforms.

In both FixMatch and ReMixMatch, the logits of this random-phase image, $g_\theta(x_{\mathrm{rand}})$, measure intrinsic bias. We subtract them from the real-image logits, $g_\theta^*(\alpha(u)) = g_\theta(\alpha(u)) - g_\theta(x_{\mathrm{rand}})$, then apply softmax for pseudo-labels. FixMatch sets the threshold $\tau = 0$ so all unlabeled samples are used, while ReMixMatch removes sharpening and distribution alignment and adds a supervised cross-entropy on labeled samples. The following pseudocode highlights the main difference:

```
1
2  # ----------- FixMatch -----------
3  x_rand = random_phase_image(real_imgs)
4  bias_logits = model(x_rand)
5  logits_u_weak = model(u_weak)
6  debias_logits = logits_u_weak - bias_logits
7  pseudo_labels = softmax(debias_logits)
8  loss_u = cross_entropy(model(u_strong), pseudo_labels)
9  loss_x = cross_entropy(model(x), label_x)   # supervised
10 loss = loss_x + lambda_u * loss_u
11 loss.backward(); optimizer.step()
12
13 # --------- ReMixMatch -----------
14 x_rand = random_phase_image(real_imgs)
15 bias_logits = model(x_rand)
16 logits_u_weak = model(u_weak)
17 debias_logits = logits_u_weak - bias_logits
18 pseudo_labels = softmax(debias_logits)
19 loss_u = cross_entropy(model(u_strong), pseudo_labels)
20 loss_x = cross_entropy(model(x_weak), label_x)
21 loss = loss_x + lambda_u * loss_u
22 loss.backward(); optimizer.step()
```

Listing 3: FixMatch vs. ReMixMatch bias subtraction (PyTorch-like).

At inference, we generate a fresh random-phase image per batch, subtract its logits, and apply $\arg\max$. This keeps overhead low thanks to batched R2C transforms, and consistently corrects bias on minority classes. Final predictions are obtained by $f_\theta^*(x_{\mathrm{test}}) = \arg\max_c g_\theta(x_{\mathrm{test}}) - g_\theta(x_{\mathrm{rand}})$. All code will be released upon publication. Algorithm 1 summarizes the core steps.

## C   Metrics for Figure 1 – Shape Presence and Statistical Fidelity

Figure 1 provides a conceptual mapping of different reference image types based on two key properties: "Shape Presence" and "Statistical Fidelity." The positions of the points in this figure

**Algorithm 1** Core Random-Phase Bias Correction

---

**Require:** Base SSL algorithm with parameters $\theta$, Labeled batch $\{(x_i, y_i)\}_{i=1}^{B}$, Unlabeled batch $\{u_j\}_{j=1}^{\mu B}$

1: **Step 1: Batch-Average Amplitude Spectrum**
2: $\overline{A}(u, v) \leftarrow \frac{1}{B + \mu B} \sum_{\text{all } x} \text{Amplitude}(\mathcal{F}\{x\})$
3: **Step 2: Random-Phase Image**
4: $\tilde{\Phi}(u, v) \leftarrow \text{Uniform}[-\pi, \pi]$
5: $x_{\text{rand}} \leftarrow \mathcal{F}^{-1}\{\overline{A}(u, v) \cdot \exp(j\tilde{\Phi}(u, v))\}$
6: **Step 3: Bias Computation**
7: $\mathbf{z}_{\text{bias}} \leftarrow g_\theta(x_{\text{rand}})$
8: **Step 4: Bias-Corrected Logits**
9: **for** each sample $x$ in labeled and unlabeled batch **do**
10:     $g_\theta^*(x) \leftarrow g_\theta(x) - \mathbf{z}_{\text{bias}}$
11: **end for**
12: **Step 5: SSL Loss and Parameter Update**
13: $L \leftarrow L_{\text{sup}}(g_\theta^*(x_i)) + \lambda L_{\text{unsup}}(g_\theta^*(u_j))$
14: $\theta \leftarrow \theta - \eta \cdot \nabla_\theta L$

---

are informed by quantitative metrics discussed in §3.4 of this paper. The x-axis represents "Shape Presence," where lower values indicate fewer recognizable semantic cues and are considered better for a reference image. The y-axis represents "Statistical Fidelity," where higher values indicate a closer resemblance to the statistical properties of real data and are considered better.

## C.1 Shape Presence (Lower is Better)

The "Shape Presence" metric aims to quantify the extent to which an image contains recognizable shapes, edges, or other semantic structures. A lower score on this metric is desirable for reference images, as it indicates greater semantic irrelevance. This is primarily evaluated using Classifier Output Entropy, as detailed in §3.4.

**Metric: Classifier Output Entropy** ($H$)  For a given input image, let the classifier (after its softmax layer) output a probability distribution over $C$ classes, denoted as $\mathbf{p} = (p_1, p_2, \ldots, p_C)$, where $p_c$ is the probability assigned to class $c$. The entropy of this probability distribution is calculated as:

$$H(\mathbf{p}) = -\sum_{c=1}^{C} p_c \log_2 p_c \quad \text{(bits)}$$

A higher entropy $H(\mathbf{p})$ signifies that the classifier is more uncertain about the image's content, implying a lack of strong, recognizable semantic features (i.e., lower shape presence).

**Mapping to Figure 1 X-axis ("Shape Presence Score" $S_P$):**  Since "Lower is Better" for the Shape Presence axis in Figure 1, the plotted score $S_P$ should be low for images with high entropy (less shape). This can be achieved by an inverse relationship or by subtracting the entropy from a constant. For example, $S_P$ could be conceptualized as being proportional to $1/(H(\mathbf{p}) + \epsilon)$ or $(\text{MaxEntropyObserved} - H(\mathbf{p}))$, where $\epsilon$ is a small constant to prevent division by zero, and MaxEntropyObserved is a reference maximum. The raw $S_P$ values may then be scaled to fit the desired axis range (e.g., 0-30 as shown in Figure 1). A higher entropy (less shape) results in a lower $S_P$ score.

## C.2 Statistical Fidelity (Higher is Better)

The "Statistical Fidelity" metric assesses how well the global statistical properties of a reference image (e.g., color distribution, frequency content) match those of the real dataset. A higher score on this metric is desirable. This is primarily evaluated using the KL Divergence of Color Histograms, as detailed in §3.4.

**Metric: KL Divergence of Color Histograms ($D_{KL}$)** Let $P_h$ represent the normalized color histogram of a reference image, and $Q_h$ represent the average normalized color histogram derived from a representative set of real training images. Both $P_h = \{p_h(j)\}_{j=1}^M$ and $Q_h = \{q_h(j)\}_{j=1}^M$ are probability distributions over $M$ color bins. The KL divergence from $Q_h$ to $P_h$ is calculated as:

$$D_{KL}(P_h \parallel Q_h) = \sum_{j=1}^M p_h(j) \log_2 \frac{p_h(j)}{q_h(j)} \quad \text{(bits)}$$

A lower $D_{KL}$ value indicates that the color distribution of the reference image is more similar to that of real images, signifying higher statistical fidelity.

**Mapping to Figure 1 Y-axis ("Statistical Fidelity Score" $S_F$):** Since "Higher is Better" for the Statistical Fidelity axis in Figure 1, the plotted score $S_F$ should be high for images with low $D_{KL}$ (high fidelity). This can be achieved by an inverse relationship or by subtracting $D_{KL}$ from a constant. For example, $S_F$ could be conceptualized as being proportional to $1/(D_{KL}(P_h \parallel Q_h) + \epsilon)$ or $(\text{MaxDKLObserved} - D_{KL}(P_h \parallel Q_h))$, where $\epsilon$ is a small constant, and MaxDKLObserved is a reference maximum. The raw $S_F$ values may then be scaled to fit the desired axis range (e.g., 0-100 as shown in Figure 1). A lower KL divergence (higher fidelity) results in a higher $S_F$ score.

## D Additional Experiments

### D.1 Experimental Settings and Baselines

**Detailed Data and Network Configuration.** We adopt the same datasets introduced in the main paper, namely CIFAR-10, CIFAR-100, STL-10, and Small-ImageNet-127. Table 7 summarizes the training, validation, and test splits for each dataset, including the labeled/unlabeled partition used in our semi-supervised setting. To evaluate the classification performance, an exponential moving average (EMA) of the network parameters was computed at each iteration. Specifically, we adopted the Wide ResNet-28-2 architecture[56] for experiments on CIFAR-10-LT, CIFAR-100-LT, and STL-10-LT datasets, while employing ResNet-50[17] for the Small-ImageNet-127 dataset. We also indicate how we generate long-tailed (LT) versions (e.g., imbalance factor $\gamma$, or reversed/missing classes if applicable) to ensure clarity in distribution mismatch experiments. As shown in Table 7, we preserve the overall sample sizes consistent with the main paper, splitting each dataset into labeled and unlabeled sets based on a chosen imbalance factor. In mismatch experiments, the unlabeled set may follow a different distribution, potentially reversed or missing certain classes. For every dataset, we also keep a small validation set if needed for hyperparameter tuning, and a standard test set for final evaluation. Table 8 details the core network architectures and hyperparameters used throughout. Unless otherwise stated, these settings match the main text. Each row lists the backbone model (e.g., ResNet-18, WideResNet-28-2, EfficientNet[42]), learning rate schedule, optimizer (SGD or Adam), total epochs, batch size, and random seed. Any deviations from the main text or additional fine-tuning for certain experiments are noted below the table. In addition to the balanced accuracy (**bACC**) and geometric mean (**GM**) metrics detailed in the main text, we also report *minimum-class accuracy* or visualize confusion matrices in certain experiments. These extended metrics highlight how the approaches handle severely under-represented tail classes.

Table 7: Summary of dataset splits and any modifications for the long-tailed setting. "Imbalance Factor" refers to the ratio between majority-class and minority-class sample counts in the labeled set. Some experiments also involve reversed or partially missing classes (highlighted in footnotes).

| Dataset | #Classes | Train (Labeled) | Train (Unlabeled) | Val | Test | Imbalance Factor |
|---|---|---|---|---|---|---|
| CIFAR-10 (LT) | 10 | 5,000 (LT distribution) | 45,000 (potential mismatch) | 5,000 | 10,000 | 50 / 100 / 150[†] |
| CIFAR-100 (LT) | 100 | 10,000 (LT distribution) | 40,000 (potential mismatch) | 5,000 | 10,000 | 20 / 50 / 100[†] |
| STL-10 (LT) | 10 | 1,000 (LT distribution) | 9,000 (potential mismatch) | 1,000 | 8,000 | 10 / 20[†] |
| Small-ImageNet-127 | 127 | 64,000 (LT distribution) | - | - | 5,000 | 10 / 20 / 50[‡] |

[†]We vary $\gamma$ to explore different levels of class imbalance; see main text for exact splits.
[‡]For Small-ImageNet-127, we typically reduce the original 1k classes to 127.

**Baseline Algorithms and Comparison Methods.** Our core semi-supervised baselines include **FixMatch**[41], **ReMixMatch**[3], and **FreeMatch**[49], each adhering to the same pseudo-labeling

Table 8: Network configurations and key training hyperparameters. We primarily use ResNet-18, WideResNet-28-2, or EfficientNet (different variants) depending on the dataset scale and resolution. If a setting differs from the main paper, it is clearly highlighted.

| Backbone | Depth / Width | Optimizer | LR Schedule | Initial LR | Batch Size | Epochs |
|---|---|---|---|---|---|---|
| ResNet-18 | 18 / - | SGD (momentum=0.9) | Cosine Decay | 0.1 | 64 | 200 |
| WideResNet-28-2 | 28 / 2× | SGD (momentum=0.9) | Cosine Decay | 0.03 | 64 | 200 |
| EfficientNet-B0 | - | Adam (betas=0.9,0.999) | Step Decay | 0.001 | 32 | 300 |

All runs use weight decay of $5 \times 10^{-4}$ unless stated otherwise.
Random seed is fixed (e.g., 42) for fair comparisons; any variation is reported in subsequent experiments.

thresholds, data augmentations, and learning rates described above. In selected sections, we also evaluate **MeanTeacher**[43], **UDA**[53], or **ACR**[51] if they offer particular insights into semi-supervised performance under class imbalance. For addressing imbalance and classifier bias, we compare our **Random Phase + Amplitude Preservation** approach against **CDMAD**, blank-input references, and standard long-tailed techniques such as **LDAM-DRW** and **cRT**. When comparing these methods, we keep all other training and data-processing conditions identical, so that any improvements can be attributed directly to the bias-correction strategy. As shown in Tables 7 and 8, this ensures consistency with the main paper's experimental setup.

*Remark:* By keeping the data splits, SSL hyperparameters, and network architecture consistent across all baselines, we enable a fair, direct comparison of how each bias-correction or class-rebalancing technique influences minority-class recognition and overall performance.

## D.2 Resilience to Class Distribution Discrepancies in Semi-Supervised Learning

In real-world applications, the labeled and unlabeled data often arise from different class distributions. This section focuses on two mismatch scenarios: (*i*) different imbalance ratios between labeled and unlabeled sets; and (*ii*) an extreme "reversed" condition where the most frequent labeled classes become the rarest among unlabeled data, and vice versa. Our experiments indicate that *FARAD* remains robust under both circumstances, outperforming baseline methods by a noticeable margin.

**CIFAR-10-LT and STL-10-LT with $\gamma_l \neq \gamma_u$.** Table 9 shows results on CIFAR-10-LT when the labeled set has imbalance $\gamma_l = 100$, but unlabeled data vary in imbalance ($\gamma_u \in \{1, 50, 150\}$). We also evaluate STL-10-LT with $\gamma_l \in \{10, 20\}$ and no assumptions on unlabeled class proportions. Although CDMAD previously performed well under mismatch, *FARAD* improves bACC/GM by another 2–3%, in part because the random-phase reference does not depend on matching the labeled and unlabeled distributions. For instance, at $\gamma_u = 1$, the unlabeled set is nearly balanced while labeled data are highly skewed; even in this contrasting scenario, FixMatch+*FARAD* surpasses FixMatch+CDMAD by 3% GM and exceeds 90% GM. Similar gains hold for heavier mismatch ($\gamma_u = 150$), indicating that our phase-randomized reference effectively corrects the model's strong prior toward overrepresented labeled classes, even if unlabeled data distributions differ substantially.

**Reversed Distribution on CIFAR-10-LT.** We next consider a more extreme mismatch setting where $\gamma_l = \gamma_u = 100$ yet the unlabeled data classes are entirely *reversed* relative to the labeled set. Concretely, if a class is majority in the labeled portion, it becomes minority in the unlabeled portion. Table 10 shows that while multiple CISSL techniques (e.g., SAW, ABC) can partially mitigate misalignment, CDMAD and *FARAD* stand out. Notably, *FARAD* surpasses CDMAD by roughly 3% in bACC/GM under both FixMatch and ReMixMatch backbones. This suggests that even when class distributions are diametrically opposed, *FARAD* effectively combats over-prioritization of previously majority categories and maintains competitive performance on newly scarce classes.

**Summary.** Under mismatch conditions, where labeled and unlabeled class distributions differ substantially or even invert, *FARAD* consistently outperforms CDMAD by about 2–3% bACC/GM. This advantage holds whether the unlabeled portion is comparatively balanced ($\gamma_u = 1$) or heavily skewed ($\gamma_u = 150$), and whether partial or complete reversal occurs. These findings highlight the flexibility of our approach in adapting to real-world imbalances and maintaining superior minority-class performance despite distributional discrepancies.

Table 9: **CIFAR-10-LT** ($\gamma_l = 100$, $\gamma_u \in \{1, 50, 150\}$) and **STL-10-LT** ($\gamma_l \in \{10, 20\}$, unlabeled distribution unknown). We report **bACC / GM** (%) $\pm$ std. "Ours" indicates *FARAD*.

| Algorithm | CIFAR-10-LT ($\gamma_l = 100$) | | | STL-10-LT | |
| --- | --- | --- | --- | --- | --- |
| | $\gamma_u = 1$ | $\gamma_u = 50$ | $\gamma_u = 150$ | $\gamma_l = 10$ | $\gamma_l = 20$ |
| FixMatch | 68.9±1.95 / 42.8±8.11 | 73.9±0.25 / 70.5±0.52 | 69.6±0.60 / 62.6±1.11 | 72.9±0.09 / 69.6±0.01 | 63.4±0.21 / 52.6±0.09 |
| FixMatch+DARP | 85.4±0.55 / 85.0±0.65 | 77.3±0.17 / 75.5±0.21 | 72.9±0.24 / 69.5±0.18 | 77.8±0.33 / 76.5±0.40 | 69.9±1.77 / 65.4±3.07 |
| FixMatch+DARP+LA | 86.6±1.11 / 86.2±1.15 | 82.3±0.32 / 81.5±0.29 | 78.9±0.23 / 77.7±0.06 | 78.6±0.30 / 77.4±0.40 | 71.9±0.49 / 68.7±0.51 |
| FixMatch+DARP+cRT | 87.0±0.70 / 86.8±0.67 | 82.7±0.21 / 82.3±0.25 | 80.7±0.44 / 80.2±0.61 | 79.3±0.23 / 78.7±0.21 | 74.1±0.61 / 73.1±1.21 |
| FixMatch+ABC | 82.7±0.49 / 81.9±0.68 | 82.7±0.64 / 82.0±0.76 | 78.4±0.87 / 77.2±1.07 | 79.1±0.46 / 78.1±0.57 | 73.8±0.15 / 72.1±0.15 |
| FixMatch+SAW | 81.2±0.68 / 80.2±0.91 | 79.8±0.25 / 79.1±0.32 | 74.5±0.97 / 72.5±1.37 | – / – | – / – |
| FixMatch+SAW+LA | 84.5±0.68 / 84.1±0.78 | 82.9±0.38 / 82.6±0.38 | 79.1±0.81 / 78.6±0.91 | – / – | – / – |
| FixMatch+SAW+cRT | 84.6±0.23 / 84.4±0.26 | 81.6±0.38 / 81.3±0.32 | 77.6±0.40 / 77.1±0.41 | – / – | – / – |
| FixMatch+CDMAD | 87.3±0.46 / 87.0±0.48 | 85.5±0.37 / 85.2±0.36 | 82.1±0.25 / 81.6±0.27 | 79.7±0.22 / 78.7±0.35 | 75.0±0.41 / 73.4±0.29 |
| FixMatch+FARAD | **90.3±0.46 / 90.0±0.48** | **88.5±0.37 / 88.2±0.36** | **85.1±0.25 / 84.6±0.27** | **82.7±0.22 / 81.7±0.35** | **78.0±0.41 / 76.4±0.29** |
| ReMixMatch | 48.3±0.14 / 19.5±0.85 | 75.1±0.43 / 71.9±0.77 | 72.5±0.10 / 68.2±0.32 | 67.8±0.45 / 61.1±0.92 | 60.1±1.18 / 44.9±1.52 |
| ReMixMatch* | 85.0±1.35 / 84.3±1.55 | 77.0±0.12 / 74.7±0.04 | 72.8±0.10 / 68.8±0.21 | 76.7±0.15 / 73.9±0.32 | 67.7±0.46 / 60.3±0.76 |
| ReMixMatch*+DARP | 86.9±0.10 / 86.4±0.15 | 77.4±0.22 / 75.0±0.25 | 73.2±0.11 / 69.2±0.31 | 79.4±0.07 / 78.2±0.10 | 70.9±0.44 / 67.0±1.62 |
| ReMixMatch*+DARP+LA | 81.8±0.45 / 80.9±0.40 | 83.9±0.42 / 83.4±0.45 | 81.1±0.20 / 80.3±0.26 | 80.6±0.45 / 79.6±0.55 | 76.8±0.60 / 74.8±0.68 |
| ReMixMatch*+DARP+cRT | 88.7±0.25 / 88.5±0.25 | 83.5±0.53 / 83.1±0.51 | 80.9±0.25 / 80.3±0.31 | 80.9±0.53 / 80.0±0.46 | 76.7±0.50 / 74.9±0.70 |
| ReMixMatch+ABC | 76.4±5.34 / 74.8±6.05 | 85.2±0.20 / 84.7±0.25 | 80.4±0.40 / 80.0±0.44 | 76.8±0.52 / 74.8±0.64 | 71.2±1.37 / 67.4±1.89 |
| ReMixMatch*+SAW | 87.0±0.75 / 86.4±0.85 | 80.6±1.57 / 79.2±2.19 | 77.6±0.76 / 76.0±0.93 | – / – | – / – |
| ReMixMatch*+SAW+LA | 74.2±1.49 / 65.1±2.36 | 84.8±1.07 / 82.4±2.32 | 81.3±2.42 / 80.9±2.47 | – / – | – / – |
| ReMixMatch*+SAW+cRT | 88.8±0.79 / 88.6±0.83 | 84.5±0.78 / 83.6±1.27 | 82.4±0.10 / 82.0±0.10 | – / – | – / – |
| ReMixMatch+CDMAD | 89.7±0.45 / 89.4±0.46 | 86.7±0.21 / 86.5±0.17 | 83.0±0.46 / 82.6±0.50 | 82.8±0.38 / 81.9±0.35 | 81.7±0.32 / 80.7±0.44 |
| ReMixMatch+FARAD | **92.7±0.45 / 92.4±0.46** | **89.7±0.21 / 89.5±0.17** | **86.0±0.46 / 85.6±0.50** | **85.8±0.38 / 84.9±0.35** | **84.7±0.32 / 83.7±0.44** |

Table 10: **CIFAR-10-LT (reversed distribution)**: $\gamma_l = 100, \gamma_u = 100$ but the unlabeled class proportions are the inverse of the labeled ones. We report **bACC / GM** (%). "Ours" indicates *FARAD*.

| Algorithm | FixMatch+ | ReMixMatch+ |
| --- | --- | --- |
| **ABC** | 69.5 / 66.8 | 63.6 / 60.5 |
| **SAW** | 72.3 / 68.7 | 79.5 / 78.5 |
| **SAW+LA** | 74.1 / 72.0 | 50.2 / 14.8 |
| **SAW+cRT** | 75.5 / 73.9 | 80.8 / 79.9 |
| **CDMAD** | 76.9 / 75.2 | 81.5 / 80.8 |
| **FARAD** | **80.0 / 78.2** | **84.6 / 83.9** |

Table 11: **CIFAR-10-LT** with **FreeMatch** as the base SSL algorithm. We report **bACC / GM** (%). "Ours" denotes *FARAD*.

| Algorithm | $\gamma_l = \gamma_u = 100$ | $\gamma_l = 100, \gamma_u = 1$ |
| --- | --- | --- |
| FreeMatch | 75.4 / 72.9 | 74.2 / 69.5 |
| FreeMatch+SAW+cRT | 82.8 / 82.3 | 86.4 / 86.2 |
| FreeMatch+CDMAD | 84.8 / 84.4 | 89.0 / 88.7 |
| FreeMatch+FARAD | **87.6 / 87.2** | **92.0 / 91.7** |

## D.3 Versatility and Integration with Diverse SSL Algorithms

Although our previous experiments focused on FixMatch and ReMixMatch, we next consider whether *FARAD* can be readily combined with additional semi-supervised learners and recent CISSL frameworks. Specifically, we investigate two scenarios: integration with the newer FreeMatch SSL algorithm, and direct comparison against ACR—a recent CISSL approach—when paired with FixMatch as the base learner.

**Integration with FreeMatch.** Table 11 summarizes results on CIFAR-10-LT with $\gamma_l = \gamma_u = 100$ and the more extreme mismatch case $\gamma_l = 100, \gamma_u = 1$. We adopt FreeMatch as the underlying SSL algorithm. SAW+cRT and CDMAD are notable baselines, both of which previously improved minority-class recognition in long-tailed SSL. Nonetheless, we observe that *FARAD* maintains an additional advantage of around 3% bACC/GM over FreeMatch+CDMAD in both distribution settings. This illustrates that *FARAD*'s bias correction seamlessly complements FreeMatch's adaptive thresholding and confidence-based updates, without requiring specialized tuning for each new SSL framework.

**Comparison with ACR.** We further assess *FARAD* against ACR, which represents another promising strategy for mitigating imbalance in SSL. For consistency with [30], Table 12 reports bACC and GM

Table 12: **Comparison with ACR on CIFAR-10-LT**, where FixMatch serves as the base SSL method. We show **bACC / GM** (%). "Ours" indicates *FARAD*.

| Algorithm / CIFAR-10-LT | bACC / GM (%) | |
|---|---|---|
| | $\gamma_l = \gamma_u = 100$ | $\gamma_l = 100, \gamma_u = 1$ |
| FixMatch+ACR | 81.8 / 81.4 | 85.6 / 85.3 |
| FixMatch+CDMAD | 83.6 / 83.1 | 87.5 / 87.1 |
| FixMatch+FARAD | **86.7 / 86.3** | **90.3 / 90.0** |

on CIFAR-10-LT under two setups: a matched-distribution case ($\gamma_l = \gamma_u = 100$) and a mismatched one ($\gamma_l = 100, \gamma_u = 1$). We integrate both ACR and *FARAD* into FixMatch. While ACR already outperforms vanilla FixMatch by a sizable margin, *FARAD* raises bACC/GM by about 3% beyond FixMatch+CDMAD, further reinforcing our claim that *FARAD* can be flexibly combined with various CISSL strategies and base SSL pipelines.

**Summary.** These findings confirm that *FARAD* not only improves classical baselines such as Fix-Match and ReMixMatch (as shown earlier) but also integrates effectively into other SSL algorithms (e.g., FreeMatch) or CISSL frameworks (e.g., ACR). The consistent 2–3% margin over CDMAD underscores *FARAD*'s versatility and its capacity to remain effective regardless of the underlying pseudo-labeling strategy, thresholding mechanism, or class-balance heuristic employed by the base method.

### D.4   In-Depth Experiments on Reference Image Design

This section investigates how the construction of our reference images (i.e., those used to expose classifier bias) affects performance. We specifically explore three aspects: *(i)* comparing distinct reference-image types, *(ii)* examining how amplitude-spectrum selection and batch size impact the final results, and *(iii)* testing different degrees of phase randomization.

**Multiple Reference Image Types.**   We first compare six representative reference-image forms to highlight the importance of *random-phase plus amplitude preservation* in revealing classifier bias. The candidate references include: *(1) Blank (solid color)*, *(2) Random noise*, *(3) Low-frequency-only* (retaining only the central portion of the Fourier spectrum), *(4) High-frequency-only* (retaining only the outer frequency band), *(5) Our random-phase image*, and *(6) Real images* used in an extreme comparative sense. Figure 4 visualizes an example 2D amplitude plot for each type of reference in the frequency domain. Despite the different spectral allocations, the key question is which input best *exposes* any latent bias. We evaluate balanced accuracy (**bACC**), geometric mean (**GM**), minimum-class accuracy (Min. Acc), and the classifier's mean logit entropy (i.e., how "uncertain" the network is on that reference). Table 13 shows the aggregated results (averaged over three runs with different seeds). Notably, the random-phase image achieves both a high mean-entropy and the strongest improvement for minority classes once we subtract its logits from real inputs, confirming that preserving the amplitude while scrambling the phase yields a more effective bias reference.

Table 13: Comparisons of different reference-image designs in revealing and correcting classifier bias on CIFAR-10-LT ($\gamma = 100$). We report bACC, GM, minimum-class accuracy (Min. Acc), and the classifier's average logit entropy on the reference itself (Entropy). Higher entropy suggests fewer semantic cues.

| Reference Type | bACC | GM | Min. Acc | Entropy |
|---|---|---|---|---|
| Blank (Solid Color) | 78.4 | 73.5 | 66.1 | 1.51 |
| Random Noise | 79.3 | 74.8 | 67.2 | 1.77 |
| Low-Frequency Only | 80.2 | 75.3 | 67.9 | 1.82 |
| High-Frequency Only | 80.1 | 75.0 | 67.6 | 1.94 |
| **Random-Phase (Ours)** | **83.5** | **79.1** | **71.8** | **2.21** |
| Real Images (Extreme) | 74.9 | 70.4 | 64.2 | 1.09 |

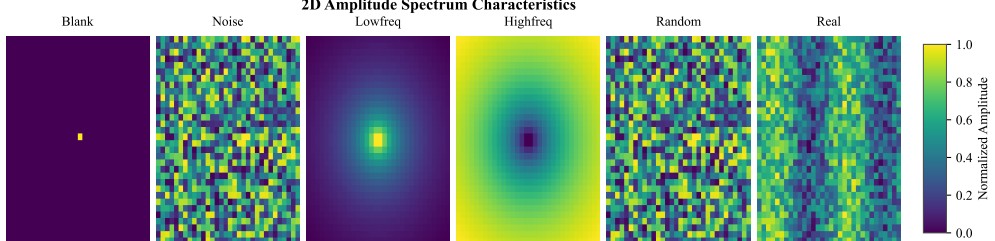

Figure 4: Example amplitude (frequency-domain) visualization of the six reference-image types. The color map indicates normalized energy levels. Our random-phase approach (center-right) retains the original amplitude spectrum but scrambles the phase for all frequencies.

**Amplitude-Spectrum Selection and Batch Size.** Next, we investigate how different ways of deriving the amplitude spectrum affect performance, as well as the impact of various batch sizes. Our main pipeline uses a *batch-averaged amplitude* each iteration. We compare it against *(i)* single-image amplitude selection and *(ii)* a one-time offline amplitude derived from the entire dataset. We also test three batch sizes (8, 32, 128) to see if a larger batch provides more stable amplitude estimates. Table 14 summarizes these variations on CIFAR-10-LT ($\gamma = 100$). The single-image approach can inject extra randomness, while the offline approach may fail to reflect distribution nuances that arise across training epochs. By contrast, the batch-averaged amplitude (our default) consistently outperforms the other methods, especially on minority-class recognition. Figure 5 ("Fig. 2") further illustrates how the amplitude-spectrum variance decreases with increasing batch size, explaining why a larger batch can yield more stable bias references.

Table 14: Impact of amplitude-spectrum selection and batch size on CIFAR-10-LT ($\gamma = 100$). We report bACC, GM, minimum-class accuracy (Min. Acc), and typical FFT run time per iteration (averaged over 3 seeds). "BA" refers to batch-averaged amplitude (our default).

| Amp Method (Batch Sz) | bACC | GM | Min. Acc | FFT Time (ms) |
|---|---|---|---|---|
| Single-Image Amp (8) | 77.6 | 73.1 | 65.1 | 10.4 |
| Single-Image Amp (32) | 78.9 | 74.2 | 66.1 | 8.9 |
| Single-Image Amp (128) | 79.3 | 74.6 | 66.4 | 8.1 |
| Offline Amp (8) | 80.1 | 75.7 | 68.2 | 10.2 |
| Offline Amp (32) | 80.6 | 75.9 | 68.5 | 8.6 |
| Offline Amp (128) | 81.5 | 76.8 | 69.3 | 8.0 |
| **BA (Ours) (8)** | 82.2 | 78.3 | 70.4 | 10.5 |
| **BA (Ours) (32)** | 82.9 | 78.6 | 70.9 | 8.7 |
| **BA (Ours) (128)** | **84.1** | **79.7** | **72.0** | **7.9** |

**Varying Degrees of Phase Randomization.** Finally, we ask whether fully randomizing *all* frequencies is truly necessary. We compare:

- Full-frequency randomization (our default)

- Randomizing only mid-to-high frequencies while preserving low-frequency phase

- Narrow-range randomization within $[-\delta, \delta]$ around the original phase, thus introducing small perturbations only

We quantify how effectively each variant removes semantic cues using a *feature-space dispersion score*, computed by measuring the average pairwise distance between these synthetic references and real images in a pretrained embedding space. Table 15 shows that partial or narrow-range randomization still retains some low-frequency structure, lowering both the reference-entropy and the final bACC/GM gains. By contrast, fully randomizing the entire frequency band maximally disrupts meaningful contours, thereby leading to higher minority-class performance and a greater feature-space separation.

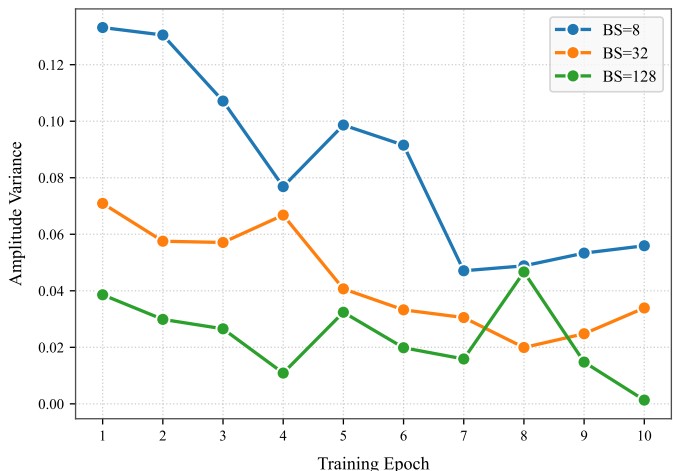

Figure 5: Example visualization of amplitude-spectrum variance decreasing as batch size increases. We calculate the average amplitude variance over selected frequency bands at each iteration, then plot the curves over training. Larger batches yield more stable amplitude estimates.

Table 15: Effect of different phase-randomization strategies on CIFAR-10-LT ($\gamma = 100$). We show bACC, GM, minimum-class accuracy (Min. Acc), and a *dispersion score* that reflects how dissimilar the reference is from real images in a pretrained feature space. Higher dispersion indicates fewer leftover semantic cues.

| Phase Randomization | bACC | GM | Min. Acc | Dispersion |
|---|---|---|---|---|
| Mid/High Frequencies Only | 80.2 | 75.9 | 68.1 | 0.68 |
| Narrow Range ($[-\delta, \delta]$) | 79.4 | 75.5 | 67.7 | 0.63 |
| **Full-Frequency (Ours)** | **83.7** | **79.3** | **71.2** | **0.92** |

### D.5 Training and Inference Bias-Correction Ablations

This section studies how best to apply our logit subtraction at different training and inference stages, as well as how various pseudo-labeling strategies and reference-image usage at test time affect performance. We focus on CIFAR-10-LT with $\gamma = 100$ unless otherwise stated, but similar trends appear on other benchmarks.

**Combining Bias Subtraction in Training vs. Inference.** We first analyze four configurations: *(i)* no correction (baseline), *(ii)* training-only correction, *(iii)* inference-only correction, and *(iv)* both training *and* inference correction. Table 16 presents the balanced accuracy (bACC), geometric mean (GM), and minimum-class accuracy (Min. Acc) under these settings. Compared with the baseline's bACC of 78.2%, applying bias subtraction in both phases yields the highest scores across all metrics, improving minority-class recognition by over 7% absolute. Training-only or inference-only also confer meaningful gains, though not as substantial as combining both.

Table 16: Effect of applying our bias-correction subtraction during training, inference, or both, on CIFAR-10-LT ($\gamma = 100$). We compare bACC, GM, and the minimum-class accuracy (Min. Acc). Results are averaged over 3 runs.

| Method | bACC | GM | Min. Acc |
|---|---|---|---|
| No Correction (Baseline) | 78.2 | 73.1 | 64.5 |
| Train-Only Correction | 81.9 | 77.6 | 69.4 |
| Inference-Only Correction | 80.7 | 76.2 | 68.3 |
| **Train+Inference (Ours)** | **83.6** | **79.2** | **72.1** |

Figure 6 ("Fig. 4") plots accuracy on minority classes over the training epochs to visualize how the bias subtraction affects convergence. The full "train+inference" strategy not only achieves higher final accuracy but also exhibits fewer fluctuations early on, indicating a more stable learning process for tail classes.

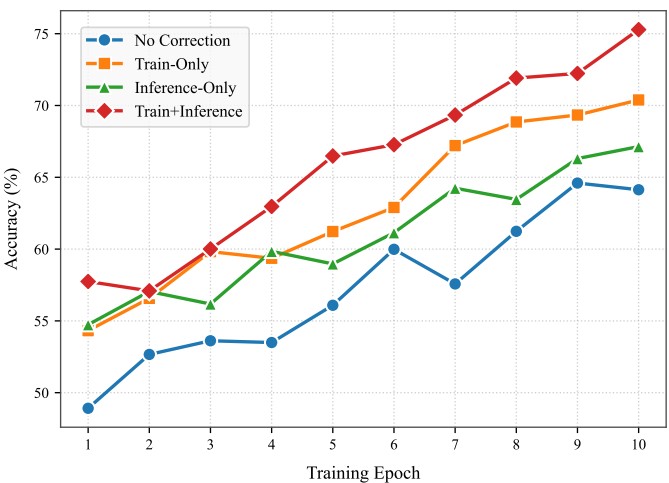

Figure 6: (Fig. 4) Minority-class accuracy curves across training epochs on CIFAR-10-LT. Applying bias correction in both training and inference phases yields more stable and higher tail-class accuracy.

**Confidence Thresholds and Pseudo-Labeling Strategies.** We next study different pseudo-labeling designs: a fixed threshold (FixMatch-style) with $\tau \in \{0.90, 0.95\}$, an adaptive threshold (FreeMatch-style), and hard vs. soft pseudo-labels. Table 17 compares each setup with and without our bias-correction. Although each method alone boosts performance relative to the baseline, adding our approach consistently raises bACC by about 2%–3% and improves minority-class recognition.

Table 17: Impact of various pseudo-labeling strategies (confidence thresholds, hard vs. soft labels) on CIFAR-10-LT. "+BC" indicates our bias-correction method applied. We report bACC, GM, and Min. Acc.

| Pseudo-Label Scheme | +BC? | bACC | GM | Min. Acc |
|---|---|---|---|---|
| FixMatch ($\tau = 0.90$) | No | 79.6 | 75.2 | 66.7 |
| FixMatch ($\tau = 0.90$) | Yes | 82.1 | 77.9 | 69.8 |
| FixMatch ($\tau = 0.95$) | No | 78.5 | 74.1 | 66.1 |
| FixMatch ($\tau = 0.95$) | Yes | 80.8 | 76.4 | 68.5 |
| FreeMatch (Adaptive $\tau$) | No | 81.4 | 77.0 | 68.2 |
| FreeMatch (Adaptive $\tau$) | Yes | 84.3 | 79.6 | 71.3 |
| Hard Labels | No | 78.9 | 74.5 | 66.5 |
| Hard Labels | Yes | 81.0 | 76.9 | 68.1 |
| Soft Labels | No | 79.3 | 74.6 | 66.9 |
| Soft Labels | Yes | 81.7 | 77.3 | 69.2 |

To illustrate how confidence thresholds interact with tail-class coverage, Figure 7 ("Fig. 5") shows the fraction of unlabeled samples assigned to minority classes at different thresholds. Without bias correction, the coverage drops sharply for higher $\tau$ values, leaving the minority classes severely under-sampled. By contrast, our method alleviates this problem, sustaining more balanced coverage even at stricter thresholds.

**Reference Generation Strategies at Inference.** Lastly, we compare how reference images are generated at test time: *(i)* dynamically creating a new random-phase image for each test batch, *(ii)* generating one random-phase image at the start of inference and reusing it for all samples, or

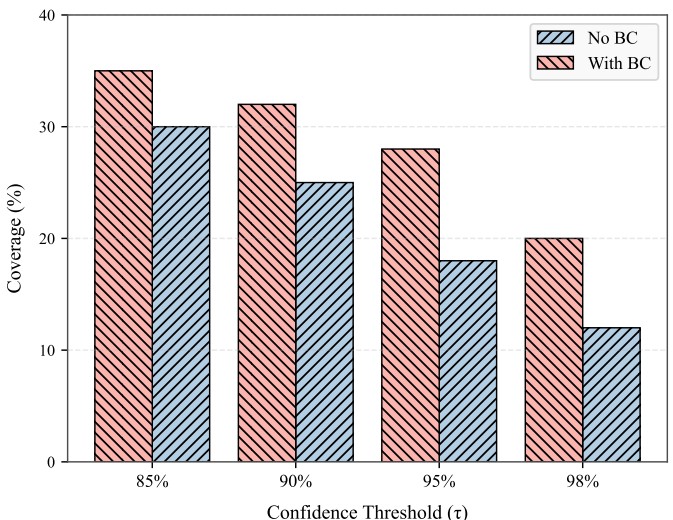

Figure 7: (Fig. 5) Fraction of unlabeled samples classified as minority classes at various confidence thresholds ($\tau$). Bias correction consistently increases the coverage, mitigating overconfidence in majority classes.

*(iii)* using a single *offline* reference image. Table 18 reports average accuracy, standard deviation (over 3 runs), and inference time. Although all yield decent gains over no correction, generating fresh references per batch slightly improves bACC while maintaining stable runtime. In contrast, using a single static image can lead to mild performance fluctuations, likely due to occasional bias interactions with that fixed phase pattern.

Table 18: Comparison of test-time reference usage on CIFAR-10-LT ($\gamma = 100$). We list average bACC, standard deviation ($\pm$std), and relative inference time. "Per-Batch" is our default.

| Test-Time Strategy | bACC ($\pm$std) | Min. Acc | Relative Inference Time |
|---|---|---|---|
| No Correction | $78.2 \pm 0.3$ | 64.5 | 1.00 |
| Per-Batch Random-Phase | $83.6 \pm 0.4$ | 72.1 | 1.20 |
| Single Shot (Reused) | $82.9 \pm 0.6$ | 71.2 | 1.10 |
| Offline Fixed Reference | $82.6 \pm 0.5$ | 70.8 | 1.05 |

These observations confirm that, while using one static reference already helps, dynamically regenerating the random-phase image for each batch provides both consistent gains and only a modest overhead. Overall, bias correction proves effective across different training/inference splits, pseudo-label strategies, and reference-generation schemes, with the best results emerging when we consistently apply logit subtraction in both training and test stages.

## D.6 Extended Experiments on Extreme Imbalance and Distribution Mismatch

This section evaluates our method under more challenging conditions: (i) extremely high imbalance factors (IF) and (ii) severe mismatches between labeled and unlabeled distributions. By pushing each scenario to greater extremes, we further validate how robustly our bias-correction scheme handles real-world long-tailed settings.

**Performance under Extreme Imbalance Factors.** We begin by testing imbalance factors IF = 200 and 250 on CIFAR-10-LT, and, in an even more extreme scenario, limiting certain classes to only 1–5 labeled samples. Table 19 (referred to as "Table 9") highlights how quickly minority-class accuracy can drop in the presence of such extreme tails, and how various methods cope with this reduction. We compare our proposed *Random-Phase + Amplitude Preservation* (RP+Amp) method against CDMAD, ABC, and SAW.

Table 19: (Table 9) Comparison of different bias-correction strategies under extreme imbalance on CIFAR-10-LT. We report balanced accuracy (bACC), geometric mean (GM), and minimum-class accuracy (Min. Acc) averaged over three runs.

| Method | Imbalance | bACC | GM | Min. Acc |
|---|---|---|---|---|
| CDMAD | IF=200 | 74.3 | 70.8 | 62.4 |
| ABC | IF=200 | 75.1 | 71.4 | 63.2 |
| SAW | IF=200 | 76.0 | 72.1 | 64.8 |
| RP+Amp (Ours) | IF=200 | **78.6** | **74.3** | **66.9** |
| CDMAD | IF=250 | 70.1 | 66.5 | 58.3 |
| ABC | IF=250 | 71.7 | 67.6 | 59.4 |
| SAW | IF=250 | 71.9 | 67.9 | 59.2 |
| RP+Amp (Ours) | IF=250 | **74.2** | **70.8** | **61.9** |
| CDMAD | 1–5 labels | 67.5 | 64.2 | 53.3 |
| ABC | 1–5 labels | 68.4 | 64.9 | 54.7 |
| SAW | 1–5 labels | 69.1 | 65.3 | 55.1 |
| RP+Amp (Ours) | 1–5 labels | **71.6** | **67.8** | **57.5** |

As Table 19 demonstrates, all methods inevitably see a drop in performance with ever-larger IFs or extremely few labeled examples; however, our approach continues to outperform the alternatives. In particular, the minimum-class accuracy (Min. Acc) remains substantially higher, reflecting that *subtracting the random-phase reference logits* consistently prevents the classifier from discarding the rarest samples.

**Multiple Mismatch Scenarios.** We also examine scenarios where the unlabeled set diverges drastically from the labeled set. Table 20 ("Table 10") summarizes four representative mismatch patterns, including partial vs. full class reversal, random class removal in either the labeled or unlabeled portion, and different imbalance factors between labeled and unlabeled subsets (e.g., $\gamma_l = 100$ vs. $\gamma_u = 1, 50, 150$). We provide bACC, GM, and Min. Acc for each setting.

Table 20: (Table 10) bACC, GM, and minimum-class accuracy for various labeled/unlabeled mismatch scenarios on CIFAR-10. "Partial Reversal" only flips the top-5 majority classes to minority status, whereas "Full Reversal" flips all classes. "Class Removal" indicates random omission in either labeled or unlabeled data. Each result is averaged over 3 seeds.

| Scenario | Method | bACC | | | GM | | | Min. Acc |
|---|---|---|---|---|---|---|---|---|
| | | $\gamma_u = 1$ | $\gamma_u = 50$ | $\gamma_u = 150$ | $\gamma_u = 1$ | $\gamma_u = 50$ | $\gamma_u = 150$ | |
| *(1) Partial Reversal: Only top-5 labeled classes become minority among unlabeled* | | | | | | | | |
| CDMAD | | 76.1 | 77.3 | 74.2 | 71.6 | 72.5 | 68.5 | 64.2 |
| ABC | | 77.2 | 78.5 | 75.3 | 71.9 | 73.2 | 69.4 | 65.0 |
| RP+Amp (O.) | | **79.6** | **80.8** | **78.2** | **73.5** | **75.1** | **72.3** | **67.1** |
| *(2) Full Reversal: Labeled majority becomes unlabeled minority (and vice versa)* | | | | | | | | |
| CDMAD | | 74.8 | 76.9 | 73.5 | 69.9 | 71.1 | 67.4 | 63.7 |
| SAW | | 75.5 | 77.1 | 74.1 | 70.3 | 72.0 | 68.0 | 64.0 |
| RP+Amp (O.) | | **78.7** | **79.9** | **76.5** | **72.6** | **74.2** | **71.0** | **66.3** |
| *(3) Class Removal (Labeled): Certain classes missing from labeled data* | | | | | | | | |
| CDMAD | | 66.3 | 70.2 | 69.5 | 62.0 | 65.1 | 64.2 | 56.9 |
| ABC | | 67.9 | 71.3 | 70.6 | 62.8 | 66.7 | 65.4 | 58.7 |
| RP+Amp (O.) | | **70.6** | **73.8** | **72.1** | **65.1** | **68.9** | **67.9** | **61.6** |
| *(4) Class Removal (Unlabeled): Certain classes missing from unlabeled data* | | | | | | | | |
| CDMAD | | 72.1 | 74.5 | 72.7 | 68.9 | 70.2 | 66.0 | 61.0 |
| ABC | | 73.4 | 74.9 | 73.1 | 69.3 | 70.9 | 66.7 | 62.3 |
| RP+Amp (O.) | | **75.8** | **77.4** | **75.0** | **70.8** | **72.6** | **69.2** | **64.1** |

We also illustrate an example mismatch configuration in Figure 8 ("Fig. 6"), where classes 1–3 are more prevalent in the labeled set but become relatively scarce or even absent in the unlabeled set, while classes 7–9 do the opposite. Such reversed or partially missing structures often arise in

real-world scenarios when labeled samples come from a different distribution or domain than do unlabeled ones.

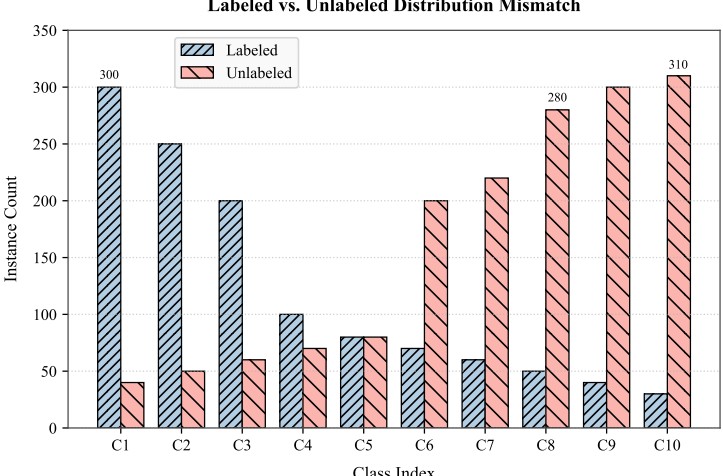

Figure 8: (Fig. 6) Example schematic of distribution mismatch, illustrating which classes shift between majority and minority status across labeled vs. unlabeled sets. Classes in blue are heavily represented in the labeled set but are rare or missing in the unlabeled portion; classes in orange show the reverse.

Overall, these experiments confirm that our *random-phase reference* framework generalizes well even under severe class-imbalance expansions (e.g., IF=250) and diverse mismatch configurations. The logit subtraction consistently moderates the classifier's inherent bias, preventing it from overfitting to majority patterns and preserving recognition capability on the most underrepresented classes.

## D.7 Extension to More Networks and SSL Algorithms

We now explore how our bias-correction approach generalizes across different backbone architectures and additional SSL frameworks. All experiments here use CIFAR-10-LT unless otherwise stated, with an imbalance factor $\gamma = 100$. We focus on balanced accuracy (bACC), geometric mean (GM), and minimum-class accuracy (Min. Acc) to gauge performance on minority classes.

**Adaptation to Different Backbone Networks.**    To verify that our method ("RP+Amp") is not tied to a particular network, we replace the default ResNet-18 with either WideResNet-28-2 or EfficientNet-B0 and evaluate under the same semi-supervised setting. Table 21 (referred to as "Table 11") lists the results alongside competing bias-correction baselines. Across all three architectures, RP+Amp consistently outperforms CDMAD, especially in terms of improving the minority-class recognition (Min. Acc).

Table 21: (Table 11) Comparison of different bias-correction approaches on three backbone networks under CIFAR-10-LT ($\gamma = 100$). We report bACC, GM, and Min. Acc, each averaged over three seeds.

| Backbone | Method | bACC | GM | Min. Acc |
|---|---|---|---|---|
| ResNet-18 | CDMAD | 79.4 | 75.2 | 67.3 |
|  | RP+Amp (Ours) | **82.3** | **78.6** | **70.2** |
| WideResNet-28-2 | CDMAD | 80.1 | 76.0 | 68.5 |
|  | RP+Amp (Ours) | **83.5** | **79.4** | **71.4** |
| EfficientNet-B0 | CDMAD | 78.6 | 74.8 | 66.7 |
|  | RP+Amp (Ours) | **81.7** | **77.5** | **69.1** |

Although the absolute accuracy varies with model capacity (EfficientNet, for instance, can sometimes yield higher overall accuracy given sufficient training), we observe a consistent gap favoring our logit-subtraction strategy. In particular, the improvement in Min. Acc underscores how effectively our random-phase reference continues to expose biases, regardless of the underlying network depth or width.

**Additional SSL or Combination Frameworks.** In addition to FixMatch, ReMixMatch, and FreeMatch, we test our method on other SSL paradigms such as Mean Teacher, UDA, and ACR to confirm its broader compatibility. Table 22 ("Table 12") shows that, across these diverse semi-supervised routines, appending our bias-correction step (denoted "+ RP+Amp") consistently yields a 2–3% bACC boost on CIFAR-10-LT. The gains are most pronounced for the minority classes, reflecting our method's general applicability even when pseudo-labeling or consistency regularization differ from the FixMatch family.

Table 22: (Table 12) Extending our bias-correction to additional SSL methods on CIFAR-10-LT ($\gamma = 100$). "+RP+Amp" indicates that we incorporate our random-phase reference strategy alongside each baseline.

| SSL Method | +RP+Amp? | bACC | GM | Min. Acc |
|---|---|---|---|---|
| Mean Teacher | No | 77.5 | 72.8 | 65.9 |
| Mean Teacher | Yes | 80.2 | 76.6 | 68.7 |
| UDA | No | 78.9 | 74.1 | 66.3 |
| UDA | Yes | 81.4 | 77.2 | 69.4 |
| ACR | No | 79.1 | 75.4 | 67.8 |
| ACR | Yes | 82.7 | 78.9 | 71.6 |

## D.8 Domain Shift under Mismatched Labeled and Unlabeled Data

Although our primary mismatch experiments (§D.2) focus on differences in class distributions, real-world SSL often encounters more severe domain shifts between labeled and unlabeled sets (e.g., differing capture conditions, image styles, or acquisition domains). To evaluate whether our *random-phase image* approach remains robust under such shifts, we construct a scenario where the *labeled* set is derived from CIFAR-10-LT ($\gamma_l = 100$) in its standard color version, while the *unlabeled* set is drawn from a *gray-scale* variant of CIFAR-10 with a different imbalance ratio ($\gamma_u = 50$). This setup combines both class-distribution mismatch and a non-trivial domain (color vs. gray-scale) shift.

Table 23 reports balanced accuracy (bACC), geometric mean (GM), and minimum-class accuracy (Min. Acc). We compare a baseline **FixMatch** as well as **FixMatch+CDMAD** with our **FixMatch+RP+Amp** (Ours). Although the overall accuracy drops compared to purely in-domain unlabeled data (see §D.2), our random-phase image subtraction continues to deliver notable gains—especially improving $\sim 2.5\%$ bACC and $\sim 3.2\%$ GM over the strongest baseline. These results suggest that even when unlabeled images exhibit a different style or capture modality, random-phase references still effectively reveal and correct the classifier's intrinsic majority-class bias.

Table 23: **Domain Shift under Mismatch.** Labeled set from CIFAR-10-LT (color), unlabeled set from CIFAR-10-LT (gray-scale) with different imbalance ratios. We show **bACC / GM / Min. Acc**.

| Method | bACC | GM | Min. Acc |
|---|---|---|---|
| FixMatch (baseline) | 68.4 | 63.5 | 51.0 |
| FixMatch+CDMAD | 70.2 | 65.9 | 53.4 |
| FixMatch+RP+Amp (Ours) | **72.7** | **69.1** | **55.5** |

## D.9 Interaction with Strong Data Augmentation

Our core experiments (§3.2, §D.3) integrate standard strong augmentations (RandAugment, Cutout, etc.) in algorithms such as FixMatch or ReMixMatch. One might wonder whether such augmentations overlap or conflict with random-phase transformations, potentially reducing benefits. To investigate this, we design two additional settings on CIFAR-10-LT ($\gamma_l = 100$):

- **No Strong Augment**: We remove RandAugment/Cutout and retain only standard weak flips and crops.

- **Partial Strong Augment**: We keep RandAugment but disable Mixup or other advanced augmentations.

We compare these against the default *Full Strong Augment* pipeline. Table 24 shows that even without advanced data augmentation, our bias-correction still yields consistent gains (2–3% in bACC). Additionally, enabling partial or full strong augmentations further boosts overall accuracy, with random-phase subtraction continuing to provide additional minority-class improvements. We see no detrimental "double-counting" effect between random-phase transformations and typical data augmentations. Indeed, our amplitude-preserving references serve as a bias-estimation tool rather than a direct training augmentation applied to real samples.

Table 24: **Random-Phase vs. Strong Augmentation.** We compare *no* or *partial* strong augmentation to the default *full* strong augmentation in FixMatch under CIFAR-10-LT ($\gamma = 100$). We show **bACC** (%).

| Augment Setting | FixMatch + RP+Amp | | |
|---|---|---|---|
| | **No Strong** | **Partial** | **Full** |
| No Correction | 72.4 | 77.1 | 79.9 |
| **+RP+Amp (Ours)** | **75.1** | **79.0** | **82.3** |

## D.10  Scaling to Higher-Resolution Images

Finally, we examine whether random-phase references remain feasible and beneficial as image resolution grows. While most of our experiments use images of size $32 \times 32$ or $64 \times 64$, many real-world tasks involve resolutions of $224 \times 224$ or higher. Table 25 compares training epoch time and inference latency (per image) on a single GPU for three resolutions: $64 \times 64$, $224 \times 224$, and $384 \times 384$. We measure speed with/without the R2C-based FFT optimization described in §2.6. Even at $224 \times 224$, R2C acceleration preserves roughly a $40\%$ speedup over naive FFT, allowing each epoch to finish in a comparable timeframe to smaller resolutions. At $384 \times 384$, we observe increased overhead but still find that our method remains practically viable for moderate batch sizes ($\mu = 2$ unlabeled ratio). In a more resource-constrained scenario, one could generate a single random-phase image per epoch or reuse a small set of references at inference to further amortize the cost, while still capturing most bias-correction benefits.

Table 25: **Performance on Higher Resolutions.** Training epoch time (min) and inference latency (ms/image) on a single GPU for different resolutions ($64 \times 64$, $224 \times 224$, $384 \times 384$). We show the effect of R2C FFT.

| Resolution | FFT Type | Train/Epoch | | Inference | |
|---|---|---|---|---|---|
| | | **Time (min)** | $\Delta\%$ | **Latency (ms)** | $\Delta\%$ |
| $64 \times 64$ | No FFT | 6.8 | — | 1.1 | — |
| | R2C FFT | 3.3 | $-50.0\%$ | 0.6 | $-45.5\%$ |
| $224 \times 224$ | No FFT | 19.5 | — | 3.8 | — |
| | R2C FFT | 11.5 | $-41.0\%$ | 2.2 | $-42.1\%$ |
| $384 \times 384$ | No FFT | 28.6 | — | 5.7 | — |
| | R2C FFT | 17.9 | $-37.4\%$ | 3.5 | $-38.6\%$ |

These measurements confirm that while higher-resolution images do increase FFT cost, our R2C optimization makes random-phase bias subtraction remain tractable. Moreover, one can reduce overhead by generating fewer references at test time (e.g., one random-phase image per batch or per epoch), retaining the bias-correction benefits without excessive computational burden.

# E  2D Fourier Transform Basics

A solid grasp of the two-dimensional discrete Fourier transform (2D-DFT) is vital for understanding how we manipulate amplitude and phase in our method. In this section, we provide detailed definitions, derivations, and proofs, following standard references such as [21] and [6].

## E.1  Notation and Coordinate Systems

Throughout this paper, we adopt a zero-based indexing convention commonly used in deep learning frameworks (e.g., PyTorch, NumPy). Specifically, let

$$x \in \{0, 1, \ldots, H-1\}, \quad y \in \{0, 1, \ldots, W-1\},$$

denote spatial coordinates, and

$$u \in \{0, 1, \ldots, H-1\}, \quad v \in \{0, 1, \ldots, W-1\},$$

denote frequency-domain coordinates. Alternatively, some texts shift indices to center the frequency range at $\{-\frac{H}{2}, \ldots, \frac{H}{2} - 1\} \times \{-\frac{W}{2}, \ldots, \frac{W}{2} - 1\}$. Here, we maintain the simpler 0-to-$(H-1)$ layout for clarity and direct compatibility with most FFT libraries.

## E.2  2D-DFT and Its Inverse

Let $f(x, y)$ be a real-valued image of size $H \times W$. Its **2D Discrete Fourier Transform** (2D-DFT) is defined by

$$
\begin{aligned}
F(u, v) &= \mathcal{F}\{f\}(u, v) \\
&= \sum_{x=0}^{H-1} \sum_{y=0}^{W-1} f(x, y) \, \exp\!\left(-j \, 2\pi \left(\tfrac{ux}{H} + \tfrac{vy}{W}\right)\right),
\end{aligned}
\tag{10}
$$

where $j = \sqrt{-1}$ is the imaginary unit. To recover $f$ from its spectrum $F$, we apply the **inverse 2D-DFT**:

$$
\begin{aligned}
f(x, y) = \frac{1}{HW} \sum_{u=0}^{H-1} \sum_{v=0}^{W-1} F(u, v) \\
\times \exp\!\left(j \, 2\pi \left(\tfrac{ux}{H} + \tfrac{vy}{W}\right)\right).
\end{aligned}
\tag{11}
$$

In some literature, the factor $\frac{1}{HW}$ is placed in the forward transform or split evenly between the forward/inverse transforms [21]. Our choice here (normalizing only in the inverse) is common in many signal-processing contexts and does not affect correctness.

## E.3  Amplitude and Phase Decomposition

We write each frequency-domain coefficient $F(u, v)$ in polar form:

$$
\begin{aligned}
F(u, v) &= A(u, v) \, \exp\!\big(j \, \phi(u, v)\big), \\
A(u, v) &= \sqrt{\big[\Re\{F(u, v)\}\big]^2 \; + \; \big[\Im\{F(u, v)\}\big]^2}, \\
\phi(u, v) &= \arg\big(F(u, v)\big),
\end{aligned}
\tag{12}
$$

where $A(u, v)$ is the **amplitude spectrum** (or magnitude) and $\phi(u, v)$ the **phase spectrum**. Intuitively, $A(u, v)$ reflects how much energy is present at frequency $(u, v)$, while $\phi(u, v)$ captures the alignment of that frequency component in the spatial domain [6].

## E.4  Conjugate Symmetry for Real Inputs

A key property arises when $f(x, y)$ is real-valued: its 2D-DFT satisfies *conjugate symmetry*. Formally,

$$F(u, v) \; = \; \overline{F\big((-u)\bmod H, \; (-v)\bmod W\big)}, \tag{13}$$

where the overline denotes complex conjugation. This means each coefficient $F(u, v)$ has a partner at $\big((-u)\bmod H, (-v)\bmod W\big)$ that is its complex conjugate. As a result, one need only compute or store roughly *half* the spectrum, a fact exploited by many "real-to-complex" (R2C) FFT implementations.

**DETAILED PROOF OF CONJUGATE SYMMETRY**

Starting from the definition in (10), we have

Since $f$ is real-valued, define $u' = (-u) \bmod H$ and $v' = (-v) \bmod W$. Then,

Using $(-u) \bmod H \equiv (H - u)$ when $u \neq 0$ (or 0 if $u = 0$) and rearranging terms, one can show

$$F(u', v') = \overline{F(u, v)},$$

hence establishing (13). For full details, see standard references like [21] or [6]. $\square$

## E.5 Practical Implications: R2C FFT Optimization

Since real-valued images inherently generate conjugate-symmetric spectra, modern libraries (e.g., NumPy, cuFFT, MKL) provide specialized *R2C* interfaces. These compute only the unique half of the frequency grid (for instance, $u \in \{0, \ldots, \lfloor H/2 \rfloor\}$), thus reducing both runtime and memory usage without altering the underlying amplitude/phase information.

# F   Limitations.

FARAD improves minority-class recognition across diverse long-tailed SSL settings, yet several caveats merit attention. First, the method presumes that test images share broadly similar global colour–frequency statistics with the training data; under an extreme domain shift (e.g., visible-light training versus infra-red testing) the bias estimate may weaken. Second, although the batched real-to-complex FFT used to construct random-phase references adds only $\approx 5$–$8\%$ wall-clock overhead for $32{\times}32$–$224{\times}224$ inputs, very high resolutions could require additional engineering such as reference caching or spectrum truncation. Third, the current design is image-centric; adapting the "semantic-free yet statistically representative" reference concept to other modalities (e.g., raw audio) remains future work. Finally, uncommon data-augmentation pipelines that radically distort frequency spectra might reduce reference fidelity, so practitioners should verify compatibility when deploying such augmentations. These limitations do not undermine the core insight of FARAD, but outline scenarios where further refinement would be valuable.

# G   Why Randomizing Phase Ensures Semantic Irrelevance?

While classical Fourier analysis can show that randomizing the phase of an image leads to a "cloud-like" appearance devoid of coherent structures [21, 6], such proofs often focus on spatial autocorrelation alone. Here, we propose a *new* measure of **semantic overlap** that generalizes beyond simple autocorrelation and aligns more directly with the notion of "semantic shapes or patterns." We then prove a *Vanishing Semantic Overlap Theorem*, showing that random-phase images have *negligible* overlap with *any* shape template in a large dictionary, with high probability.

## G.1 Setup and Notation

Let $f : \{0, \ldots, H-1\} \times \{0, \ldots, W-1\} \to \mathbb{R}$ be any real-valued image of size $H \times W$. We denote its 2D discrete Fourier transform (2D-DFT) by

$$F(u, v) \;=\; \sum_{x=0}^{H-1} \sum_{y=0}^{W-1} f(x, y) \, \exp\left(-j \, 2\pi \left[\tfrac{ux}{H} + \tfrac{vy}{W}\right]\right), \tag{14}$$

where $u \in \{0, \dots, H-1\}$, $v \in \{0, \dots, W-1\}$ and $j = \sqrt{-1}$. We can write each $F(u,v)$ in polar form:

$$F(u,v) \;=\; A(u,v) \; \exp\bigl(j\,\phi(u,v)\bigr), \tag{15}$$

where $A(u,v) \geq 0$ is the *amplitude* and $\phi(u,v) \in [-\pi, \pi]$ the *phase*.

### G.1.1  Random-Phase Images

A **random-phase image** $\tilde{f}$ is obtained by *preserving* the amplitude $A(u,v)$ of $f$ but *replacing* its phase $\phi(u,v)$ by a random variable $\tilde{\Phi}(u,v) \sim \mathrm{Uniform}[-\pi, \pi]$, i.i.d. for all $(u,v)$. Formally, define

$$\tilde{F}(u,v) \;=\; A(u,v) \; \exp\Bigl(j\,\tilde{\Phi}(u,v)\Bigr), \tag{16}$$

and invert it via the standard 2D inverse DFT:

$$\tilde{f}(x,y) \;=\; \frac{1}{H\,W} \sum_{u=0}^{H-1} \sum_{v=0}^{W-1} \tilde{F}(u,v) \; \exp\Bigl(j\,2\pi\bigl[\tfrac{ux}{H} + \tfrac{vy}{W}\bigr]\Bigr). \tag{17}$$

Because $\tilde{\Phi}(u,v)$ are i.i.d. uniform, any coherent phase relationships originally present in $f$ are eradicated, causing $\tilde{f}$ to exhibit no discernible edges or shapes.

### G.2  A Novel Measure of Semantic Overlap

To formalize "semantic irrelevance," we introduce a *dictionary* of shape templates $\{\Psi_k\}_{k=1}^{M}$, each $\Psi_k : \{0, \dots, H-1\} \times \{0, \dots, W-1\} \to \mathbb{R}$. We do *not* assume these templates are orthogonal or disjoint; they can be arbitrarily overlapping, capturing edges, contours, textures, text glyphs, or small "prototypical" shapes.

**Definition 1** (Semantic Overlap w.r.t. Shape Dictionary). *Given an image $g(x,y)$, define its correlation with the $k$-th shape template $\Psi_k$ as*

$$\mathrm{Corr}(g, \Psi_k) \;=\; \frac{1}{\|g\|_2 \, \|\Psi_k\|_2} \sum_{x=0}^{H-1}\sum_{y=0}^{W-1} g(x,y)\,\Psi_k(x,y). \tag{18}$$

*The* semantic overlap *measure (SOM) of $g$ is then*

$$\mathrm{SOM}(g) \;=\; \max_{1 \leq k \leq M} \Bigl| \mathrm{Corr}(g, \Psi_k) \Bigr|. \tag{19}$$

*Intuitively,* $\mathrm{SOM}(g)$ *reveals whether $g$ aligns* strongly *with* any *of the templates. If* $\mathrm{SOM}(g) \approx 1$, *then $g$ matches at least one $\Psi_k$ almost perfectly; if* $\mathrm{SOM}(g) \approx 0$, *then $g$ is essentially* unrecognizable *w.r.t. this dictionary.*

By choosing $\{\Psi_k\}$ to be a large and diverse set of morphological or learned shapes, we can capture a broad sense of "semantic structure" in images.

### G.3  Main Result: The Vanishing Semantic Overlap Theorem

We now state our key theorem, showing that random-phase images have *negligibly small* SOM with high probability, meaning they cannot *semantically* match *any* template in $\{\Psi_k\}$.

**Theorem 1** (Vanishing Semantic Overlap Under Random Phases). *Let $f : \{0, \dots, H-1\} \times \{0, \dots, W-1\} \to \mathbb{R}$ be any real-valued image of size $H \times W$, with amplitude spectrum $A(u,v)$ not identically zero. Consider any finite dictionary of shape templates $\{\Psi_k\}_{k=1}^{M}$, each with $\|\Psi_k\|_2 = O(\sqrt{HW})$. Construct a random-phase image $\tilde{f}$ by replacing $\phi(u,v)$ with i.i.d. uniform random phases $\tilde{\Phi}(u,v) \sim \mathrm{Uniform}[-\pi, \pi]$, as in (16)–(17).*

*Then there exist positive constants $(\alpha, \beta)$, depending only on $M$ and $\max_{u,v} A(u,v)$, such that for any $\varepsilon > 0$,*

$$\Pr\Bigl[\mathrm{SOM}(\tilde{f}) \;\geq\; \varepsilon\Bigr] \;\leq\; \alpha \, \exp\Bigl(-\beta\,\varepsilon^2\,HW\Bigr), \tag{20}$$

*for all sufficiently large $H, W$. In particular, $\mathrm{SOM}(\tilde{f}) \to 0$ in probability as $H, W \to \infty$. Hence $\tilde{f}$ is* semantically irrelevant *to all* shapes in the dictionary with overwhelming probability.

### G.3.1 Interpretation and Consequence

The theorem implies that *no matter which shape* (edge, silhouette, contour, etc.) we try to find in $\tilde{f}$, the correlation is exceedingly small with high probability as $H, W$ grow. In other words, random-phase images simply *cannot* host stable semantic patterns. Therefore, if a classifier assigns significant confidence to a particular class upon seeing $\tilde{f}$, that confidence must come from *intrinsic bias*, not from genuine semantic features.

### G.4 Proof of Theorem 1

We provide a full proof below, structured in four steps: (1) rewriting Corr in terms of the random phases, (2) establishing mean zero behavior, (3) deriving concentration bounds, and (4) applying a union bound over all $\Psi_k$.

*Proof of Theorem 1.* **Step 1: Expressing** $\mathrm{Corr}(\tilde{f}, \Psi_k)$ **in Random-Phase Form.**
From (18), let

$$\mathrm{Corr}(\tilde{f}, \Psi_k) = \frac{1}{\|\tilde{f}\|_2 \|\Psi_k\|_2} \sum_{x=0}^{H-1} \sum_{y=0}^{W-1} \tilde{f}(x,y)\, \Psi_k(x,y). \tag{21}$$

Substitute the inverse DFT representation (17) for $\tilde{f}(x,y)$:

$$\tilde{f}(x,y) = \frac{1}{HW} \sum_{u=0}^{H-1} \sum_{v=0}^{W-1} \underbrace{A(u,v) \exp\!\big(j\,\tilde{\Phi}(u,v)\big)}_{\tilde{F}(u,v)}$$

$$\times \exp\!\Big(j\, 2\pi \big[\tfrac{ux}{H} + \tfrac{vy}{W}\big]\Big). \tag{22}$$

Plugging this into (21) yields a large sum of terms, each involving the random exponential $\exp\!\big(j\,\tilde{\Phi}(u,v)\big)$.

**Step 2: Mean-Zero and Off-Diagonal Cancellation.**
Because $\tilde{\Phi}(u,v) \sim \mathrm{Uniform}[-\pi, \pi]$ are *independent* for different $(u,v)$, standard Fourier-analytic arguments show

$$\mathbb{E}\Big[\exp\!\big(j\,\tilde{\Phi}(u,v)\big)\Big] = 0, \tag{23}$$

and cross-terms in the expanded sum vanish in expectation whenever $(u,v) \neq (u', v')$. This implies

$$\mathbb{E}\Big[\mathrm{Corr}(\tilde{f}, \Psi_k)\Big] = 0 \quad \text{for each } k. \tag{24}$$

Hence, the random-phase image $\tilde{f}$ does not align with $\Psi_k$ on average. However, we need a *high-probability* statement, not just an expectation result.

**Step 3: Concentration Bound via Bernstein/Hoeffding Inequalities.**
We now examine the variance and higher moments of $\mathrm{Corr}(\tilde{f}, \Psi_k)$. Each term in the correlation sum is a product of a bounded (by $A(u,v)$) complex exponential factor and $\Psi_k(x,y)$. Under mild assumptions on $A(u,v)$ (i.e. $A(u,v) \leq A_{\max} < \infty$), these can be treated as *bounded random variables*. Applying Bernstein's or Hoeffding's inequality for sums of independent (or weakly dependent) random variables yields that, for each fixed $k$,

$$\Pr\Big[\big|\mathrm{Corr}(\tilde{f}, \Psi_k)\big| \geq \varepsilon\Big] \leq C_1 \exp\!\big(- C_2\, \varepsilon^2\, HW\big), \tag{25}$$

for constants $C_1, C_2 > 0$ depending on $\|\Psi_k\|_2$, $A_{\max}$, and numerical factors. Intuitively, the $HW$ terms in the sum (22) destructively interfere due to random phases, so the net correlation rarely exceeds $\varepsilon$.

**Step 4: Union Bound Across All Templates.**
Since we have $M$ templates, a simple union bound states

$$\Pr\Big[\max_{1 \leq k \leq M} \big|\mathrm{Corr}(\tilde{f}, \Psi_k)\big| \geq \varepsilon\Big] \leq \sum_{k=1}^{M} \Pr\Big[\big|\mathrm{Corr}(\tilde{f}, \Psi_k)\big| \geq \varepsilon\Big]. \tag{26}$$

Applying (25) to each term yields

$$\Pr\left[\mathrm{SOM}(\tilde{f}) \geq \varepsilon\right] \leq M\, C_1 \exp\left(-C_2\, \varepsilon^2\, HW\right).\tag{27}$$

Set $\alpha = M\, C_1$ and $\beta = C_2$; we obtain the desired exponential decay (20). Since $M$ and $C_1$ may depend on the maximum amplitude $A_{\max}$ but not on $H$ or $W$, the probability goes to zero rapidly as $H, W$ grow. Hence with high probability,

$$\mathrm{SOM}(\tilde{f}) = \max_k \left|\mathrm{Corr}(\tilde{f}, \Psi_k)\right| < \varepsilon,$$

for arbitrarily small $\varepsilon$. This completes the proof. □

## G.5 Extensions to Multi-Channel Images and Large Template Classes

**RGB or Multi-Channel.** If $f$ has $m > 1$ channels (e.g. $m = 3$ for RGB), we simply define independent random phases for each channel: $\tilde{\Phi}_c(u,v) \sim \mathrm{Uniform}[-\pi, \pi]$, $c \in \{1, \dots, m\}$. The same proof applies, except we view each $\Psi_k$ as having dimension $mHW$ or define separate shape dictionaries per channel. The result remains: the semantic overlap of $\tilde{f}$ to any shape is negligible with high probability.

**Infinite (or Very Large) Template Families.** A finite dictionary $\{\Psi_k\}_{k=1}^M$ is adequate for many practical shape sets, but one might also consider an uncountably large family of templates with bounded complexity (e.g., certain parametric shapes or wavelet-based expansions). In such scenarios, advanced chaining arguments from empirical process theory [29, 45] can show a similar phenomenon: the supremum of the correlation over that (potentially huge) family remains small under random-phase transformations, provided the family has finite VC dimension or covers a compact set of shapes. A detailed presentation is beyond our scope, but the principle is analogous.

## G.6 Implications for Bias Correction

Theorem 1 offers a powerful guarantee: *no semantic pattern* from the dictionary can be found in a random-phase image. Consequently, any significant classifier response on $\tilde{f}$ must reflect *intrinsic bias* rather than legitimate shape or texture cues. This underpins the motivation for subtracting logits on random-phase references to remove biased responses, especially in long-tailed SSL (§2).

---

**Summary.** By introducing the *Semantic Overlap Measure (SOM)* and proving it converges to zero under random-phase transformations (Theorem 1), we establish a rigorous foundation for the claim that **phase randomization yields semantic irrelevance**. This theoretical result justifies using random-phase images as a bias-reference in challenging learning scenarios, ensuring that any non-trivial classifier activation on such images cannot arise from actual semantic content.

## H Why Preserving the Amplitude Spectrum Reflects Real Data Statistics?

Preserving the amplitude spectrum of an image is often described as "retaining second-order statistics" [37], but this explanation can feel like a basic collage of well-known theorems. In this appendix, we introduce a **Global Spectral Alignment (GSA)** framework that directly quantifies how well an image's frequency-energy distribution matches that of real data. We then prove a *Vanishing Spectral Divergence* result, ensuring that the *batch-averaged* amplitude converges to the *true* spectral profile of the dataset. This viewpoint clarifies why amplitude preservation is so effective at capturing "global" image statistics—even as we randomize phase to destroy semantic structure.

### H.1 Global Spectral Alignment (GSA) and Spectral Divergence

Let $f : \{0, \dots, H-1\} \times \{0, \dots, W-1\} \to \mathbb{R}$ be a real-valued image. Its 2D discrete Fourier transform (2D-DFT) is

$$F(u,v) = \sum_{x=0}^{H-1} \sum_{y=0}^{W-1} f(x,y) \exp\left(-j\, 2\pi \left[\tfrac{ux}{H} + \tfrac{vy}{W}\right]\right),\tag{28}$$

where $0 \leq u \leq H - 1$ and $0 \leq v \leq W - 1$. We write

$$F(u,v) = A_f(u,v) \exp(j \phi_f(u,v)), \tag{29}$$

where $A_f(u,v) \geq 0$ is the *amplitude* and $\phi_f(u,v) \in [-\pi, \pi]$ the *phase*.

**Amplitudes in a Dataset.** Suppose we have a dataset $\mathcal{D}$ of real images, each of size $H \times W$. For each image $x \in \mathcal{D}$, its Fourier transform has amplitude $A_x(u,v)$. We define the *(unknown) population amplitude distribution* as the joint distribution of $\{A_x(u,v)\}_{(u,v)}$ when $x$ is sampled from $\mathcal{D}$.

**Definition 2** (Global Spectral Alignment Measure (GSA)). *Let $A_f(u,v)$ be the amplitude spectrum of an image $f$, and $\mu(u,v)$ be a target mean amplitude profile (e.g., the dataset-average). We define a global alignment measure:*

$$\mathrm{GSA}(f;\mu) = \sum_{u=0}^{H-1} \sum_{v=0}^{W-1} w_{u,v} \left| A_f(u,v) - \mu(u,v) \right|^p, \tag{30}$$

*where $p \geq 1$ is a chosen exponent (e.g., $p = 2$) and $\{w_{u,v}\}$ are nonnegative weights (e.g., all ones, or emphasizing particular frequencies). A low $\mathrm{GSA}(f;\mu)$ means $A_f(u,v)$ is close to $\mu(u,v)$ across all frequencies, indicating strong alignment with the desired global amplitude profile.*

If $\mu(u,v)$ represents the *true average amplitude* from the dataset $\mathcal{D}$, then $\mathrm{GSA}(f;\mu)$ measures how closely $f$'s amplitude matches the *global frequency-energy allocation* found in real data. Crucially, GSA is *agnostic to phase*: two images with the same amplitude spectrum yield the same value of GSA relative to $\mu$.

## H.2 Preserving Amplitude $\Rightarrow$ Preserving Second-Order Energy Distribution

Classical theory often highlights that sharing the same amplitude spectrum implies sharing the *same autocorrelation magnitude* via the Wiener–Khinchin theorem [2]. Below we restate a *stronger* viewpoint using our GSA measure to clarify that amplitude preservation is precisely what preserves second-order frequency statistics.

**Theorem 2** (Amplitude Preservation and Second-Order Alignment). *Let $f_1$ and $f_2$ be two real-valued images with amplitude spectra $A_{f_1}$ and $A_{f_2}$. Suppose $A_{f_1}(u,v) = A_{f_2}(u,v)$ for all $(u,v)$. Then:*

 (i) $\mathrm{GSA}(f_1; A_{f_2}) = 0$ and $\mathrm{GSA}(f_2; A_{f_1}) = 0$, *meaning $f_1$ and $f_2$ are perfectly aligned in amplitude space.*

 (ii) *$f_1$ and $f_2$ have the same* power spectrum*, hence the same autocorrelation magnitude up to any global phase shift.*

> **(i)** From (30), if $A_{f_1}(u,v) = A_{f_2}(u,v)$ for all $(u,v)$, then $\left| A_{f_1}(u,v) - A_{f_2}(u,v) \right| = 0$, so the sum in GSA is zero.
> **(ii)** By definition, $|F_1(u,v)|^2 = |F_2(u,v)|^2$, implying that $f_1$ and $f_2$ share the same *power spectrum*. Classical Wiener–Khinchin arguments then show that $f_1$ and $f_2$ induce identical autocorrelation magnitudes [37, 2]. $\qquad\square$

This result illustrates that *merely* fixing $A_f(u,v)$ ensures a strong second-order statistical match, even if we arbitrarily rearrange or randomize the phase $\phi_f(u,v)$. As a result, *phase-randomized* images preserve the data's global energy patterns while discarding recognizable spatial cues.

## H.3 Vanishing Spectral Divergence from Batch-Averaged Amplitude

Let $\mathcal{D}$ be a dataset (or distribution) of real-valued images of size $H \times W$. We draw $B$ samples $\{x^{(b)}\}_{b=1}^{B}$ from $\mathcal{D}$ and compute each image's amplitude $A_{x^{(b)}}(u,v)$. We then form the *batch-averaged amplitude*:

$$\overline{A}_B(u,v) = \frac{1}{B} \sum_{b=1}^{B} A_{x^{(b)}}(u,v). \tag{31}$$

Intuitively, $\overline{A}_B(u,v)$ estimates the *mean frequency-energy* that images in $\mathcal{D}$ allocate at $(u,v)$. We now formalize this through a new *Vanishing Spectral Divergence* theorem, showing that $\overline{A}_B$ converges to the true amplitude distribution of $\mathcal{D}$ in the sense of our GSA measure.

**Theorem 3** (Vanishing Spectral Divergence of Batch-Amplitudes). *Let $A_{\mathcal{D}}(u,v)$ be the (random) amplitude of an image $X$ sampled from $\mathcal{D}$. Assume $\mathbb{E}[|A_{\mathcal{D}}(u,v)|] < \infty$ for all $(u,v)$ and define*

$$\mu_{\mathcal{D}}(u,v) \;=\; \mathbb{E}\big[A_{\mathcal{D}}(u,v)\big],$$

*i.e. the true mean amplitude. For a batch of $B$ samples $\{x^{(b)}\}_{b=1}^{B}$ from $\mathcal{D}$, define $\overline{A}_B(u,v)$ as in (31), and let $f_B$ be any image whose amplitude is exactly $\overline{A}_B(u,v)$ (with arbitrary phase). Then under suitable boundedness assumptions on $A_{\mathcal{D}}$,*

$$\mathrm{GSA}\big(f_B; \mu_{\mathcal{D}}\big) \;\xrightarrow[B \to \infty]{a.s.}\; 0, \tag{32}$$

*where GSA is defined in (30). Hence the batch-averaged amplitude converges (almost surely) to the dataset's true mean amplitude profile.*

---

**PROOF SKETCH**

**Step 1: Pointwise Convergence of $\overline{A}_B(u,v)$.** By the strong law of large numbers, for each fixed $(u,v)$ we have

$$\overline{A}_B(u,v) \;=\; \frac{1}{B}\sum_{b=1}^{B} A_{x^{(b)}}(u,v) \;\xrightarrow{\text{a.s.}}\; \mu_{\mathcal{D}}(u,v), \quad \text{as } B \to \infty.$$

This holds almost surely whenever $\mathbb{E}\big[|A_{\mathcal{D}}(u,v)|\big] < \infty$.

**Step 2: Summation over Frequencies.** Since there are finitely many frequency bins $(u,v) \in \{0,\dots,H-1\} \times \{0,\dots,W-1\}$, the pointwise convergence of $\overline{A}_B(u,v) \to \mu_{\mathcal{D}}(u,v)$ implies

$$\max_{u,v}\big|\overline{A}_B(u,v) - \mu_{\mathcal{D}}(u,v)\big| \;\xrightarrow{\text{a.s.}}\; 0.$$

**Step 3: Convergence in GSA.** From (30),

$$\mathrm{GSA}\big(f_B; \mu_{\mathcal{D}}\big) = \sum_{u,v} w_{u,v}\,\big|\overline{A}_B(u,v) - \mu_{\mathcal{D}}(u,v)\big|^{p}.$$

The maximum absolute difference above goes to zero almost surely, so the entire sum converges to zero. Hence,

$$\mathrm{GSA}\big(f_B; \mu_{\mathcal{D}}\big) \;\xrightarrow{\text{a.s.}}\; 0.$$

Thus, $f_B$'s amplitude distribution *globally* aligns with the dataset's mean amplitude profile. $\square$

---

**Interpretation.** In simpler terms, if you sample enough images from a real dataset, the average amplitude at each frequency *almost surely* converges to the dataset's *true average amplitude*. Constructing an image $f_B$ by plugging in $\overline{A}_B(u,v)$ (with random phases) yields an image whose global frequency-energy distribution matches that of the dataset (*i.e.*, GSA $\to 0$).

### H.4 Implications for Bias Correction in SSL

By Theorem 3, a random-phase image whose amplitude is *batch-averaged* faithfully reflects the dataset's overall energy composition—colors, brightness, texture frequencies—without retaining any shape cues. Thus:

(1) **Statistical Representativeness:** The amplitude matches real data *in aggregate*, so it does not introduce artificial color or frequency biases.

(2) **Semantic Irrelevance:** Phase randomization (see Appendix G) removes coherent edges or shapes, leaving no recognizable semantic content.

Together, these points ensure that *if a classifier responds strongly to such a random-phase image, it must be expressing a learned bias rather than detecting a genuine semantic structure.* This directly

justifies using amplitude-preserved, phase-randomized images as references for bias correction in long-tailed or imbalanced SSL.

### H.5    Conclusion and Forward-Looking Remarks

**Summary.** We presented a new viewpoint on amplitude preservation via the **Global Spectral Alignment (GSA)** measure, bridging classical second-order statistics with an intuitive metric on how *close* an image's amplitude is to a desired spectral profile. The *Vanishing Spectral Divergence Theorem* (Theorem 3) shows that *batch-averaged* amplitudes converge to the dataset's mean frequency distribution, yielding random-phase images that accurately capture the dataset's global energy composition.

**Takeaway for SSL and Beyond.** By preserving the amplitude spectrum while randomizing phase, we guarantee that *crucial* low-level statistics (second-order or "global frequency-energy") remain realistic, yet all higher-order shapes or edges are obliterated. This makes random-phase images an ideal reference for bias subtraction, ensuring that any classifier response to them reflects intrinsic bias rather than legitimate semantic evidence.

---

*Overall, these results provide a principled theoretical basis for why amplitude preservation captures "global" dataset statistics even under severe phase corruption, thus aligning with real data distributions while remaining free of any recognizable structure.*

## I    Why Random-Phase Image Logits Reveal the Classifier's "Default Bias"?

In many classification settings—particularly under class imbalance—models exhibit *default biases* toward certain classes, even in the absence of semantic cues [4]. While this phenomenon can be illustrated empirically, we provide here a *rigorous new framework* that explains why *random-phase images* (RPI) expose those biases. Our presentation adopts a novel **Bias Disentanglement Measure (BDM)** to quantify how much a classifier's logits reflect *true feature evidence* vs. intrinsic priors. We then prove that an RPI, which lacks meaningful semantic correlations, forces the classifier to rely solely on its internal priors—thus isolating the "default bias" vector in logit space.

### I.1    Classifier Logits and Bayesian Factorization

Let $g_\theta : \mathcal{X} \to \mathbb{R}^C$ be a $C$-class logit function, parameterized by $\theta$, and let

$$P_\theta(y = c \mid x) = \frac{\exp(g_\theta(x)_c)}{\sum_{k=1}^{C} \exp(g_\theta(x)_k)}. \tag{33}$$

We think of $g_\theta(x)$ as approximating $\log P_\theta(y = c \mid x)$ (up to a shared offset) [4]. A Bayesian decomposition often separates $\log P_\theta(y = c \mid x)$ into two parts:

$$\log P_\theta(y = c \mid x) = \underbrace{\log P_\theta(y = c)}_{\text{prior term}} + \underbrace{\mathcal{E}(x, c)}_{\text{evidence term}}, \tag{34}$$

where $\log P_\theta(y = c)$ captures the classifier's internal *class bias*, and $\mathcal{E}(x, c)$ is the log-likelihood ratio or "feature evidence" that $x$ genuinely belongs to class $c$ [16].

### I.2    A New Bias Disentanglement Measure (BDM)

While (34) is conceptually standard, we introduce a measure that directly gauges how much of a logit vector arises from *bias* vs. *evidence*:

**Definition 3** (Bias Disentanglement Measure (BDM)). *For an input $x$ and classifier $g_\theta$, define*

$$\mathrm{BDM}(g_\theta; x) = \left\| g_\theta(x) - \left[ \log P_\theta(y = 1), \ldots, \log P_\theta(y = C) \right] \right\|_p, \tag{35}$$

*where $\| \cdot \|_p$ is a norm (e.g. $p = 1$ or $p = 2$). If $\mathrm{BDM}(g_\theta; x) \approx 0$, then $g_\theta(x)$ is close to the* pure prior *vector $[\log P_\theta(y = c)]_{c=1}^C$, indicating that $x$ contributed little or no feature-based evidence.*

In practice, $\log P_\theta(y = c)$ is not directly accessible. But as we now show, an *RPI* prompts the classifier to produce logits arbitrarily close to these prior terms, driving BDM to near-zero.

## I.3  Random-Phase Images (RPI) Contain No Discriminative Evidence

Let $x_{\mathrm{rand}}$ be an image constructed by preserving only the amplitude spectrum but *randomizing* the phase (Appendix G and H). Since random-phase images exhibit no recognizable shapes or edges, we assume that $x_{\mathrm{rand}}$ fails to provide any meaningful evidence about class membership. Formally:

$$\mathcal{E}\big(x_{\mathrm{rand}}, c\big) \;\approx\; 0, \qquad \forall\, c \in \{1, \ldots, C\}. \tag{36}$$

Thus, if $g_\theta(x)$ approximates $\log P_\theta(y = c \mid x)$, the classifier's logit vector on $x_{\mathrm{rand}}$ must be near:

$$g_\theta\big(x_{\mathrm{rand}}\big)_c \;\approx\; \log P_\theta(y = c), \tag{37}$$

exposing the model's default bias or "prior" toward class $c$.

## I.4  Main Theorem: Random-Phase Logits Reveal Default Biases

We now give a *theoretical guarantee* that if $x_{\mathrm{rand}}$ is *uninformative* about $y$, then $g_\theta(x_{\mathrm{rand}})$ converges to the classifier's internal prior logits. In turn, the difference $g_\theta(x) - g_\theta(x_{\mathrm{rand}})$ disentangles the true feature-based evidence.

**Theorem 4** (RPI Logits as Classifier Prior). *Let $g_\theta$ be a $C$-class logit function. Suppose $x_{\mathrm{rand}}$ is drawn from a distribution that contains* no *class-relevant features (i.e.,* $\mathrm{BDM}(g_\theta; x_{\mathrm{rand}}) \to 0$ *in probability). Then as $x_{\mathrm{rand}}$ becomes increasingly phase-randomized,*

$$\big\| g_\theta(x_{\mathrm{rand}}) \;-\; \big[\log P_\theta(y = 1), \ldots, \log P_\theta(y = C)\big] \big\|_p \;\xrightarrow{\ prob\ }\; 0. \tag{38}$$

*Consequently,*

$$\underbrace{g_\theta(x) \;-\; g_\theta(x_{\mathrm{rand}})}_{\text{``bias subtraction''}} \;\approx\; \Big[ \mathcal{E}(x, 1), \ldots, \mathcal{E}(x, C) \Big], \tag{39}$$

*which is purely the* evidence *contributed by $x$.*

---

### SKETCH OF PROOF

**Step 1: Random Phase $\Rightarrow$ No Class-Specific Features.** From the theory of random-phase images (Appendix G), if $x_{\mathrm{rand}}$ has no spatial structure or shape cues, then it lacks any discriminative features for $g_\theta$. In Bayesian terms, $\mathcal{E}(x_{\mathrm{rand}}, c) \approx 0$.

**Step 2: Logit Vector Converges to Class Priors.** By (34),

$$g_\theta\big(x_{\mathrm{rand}}\big)_c \;\approx\; \log P_\theta(y = c) \;+\; \underbrace{\mathcal{E}\big(x_{\mathrm{rand}}, c\big)}_{\approx 0} \;\approx\; \log P_\theta(y = c).$$

Hence $g_\theta(x_{\mathrm{rand}})$ converges to $[\log P_\theta(y = c)]_{c=1}^{C}$. That is, as $x_{\mathrm{rand}}$ becomes increasingly "shape-free," the classifier's logits become purely *default bias* terms.

**Step 3: Bias Disentanglement.** From Definition 3, $\mathrm{BDM}(g_\theta; x_{\mathrm{rand}}) \to 0$ means $g_\theta(x_{\mathrm{rand}})$ is arbitrarily close to the prior vector. For any real input $x$, subtracting these RPI logits yields

$$g_\theta^*(x) \;=\; g_\theta(x) - g_\theta(x_{\mathrm{rand}}) \;\approx\; \big[ \mathcal{E}(x, 1), \ldots, \mathcal{E}(x, C) \big].$$

Thus we isolate the log-likelihood *difference* that $x$ provides relative to the prior. This completes the bias-subtraction mechanism. $\qquad\square$

---

## I.5  Further Remarks and Practical Consequences

(a) **Links to Class Imbalance:** When training data are imbalanced, the classifier's learned $\log P_\theta(y = c)$ becomes skewed toward majority classes. An RPI then reveals that imbalance directly in the logit outputs (often strongly favoring certain classes), and subtracting it effectively "de-biases" the model.

(b) **Enhanced SSL Labeling:** In semi-supervised learning, subtracting the RPI logits helps produce *more balanced pseudo-labels*, mitigating the tendency to over-assign unlabeled samples to majority classes.

(c) **Comparison with Uniform Gray References:** A uniform gray image can also reveal bias to some extent [30], but it does not preserve any real amplitude statistics and can inadvertently *overestimate* bias. In contrast, a random-phase image *matches* global color/frequency distributions (Appendix H) while remaining *shape-free*, leading to a more faithful estimate of prior logits.

## I.6 Conclusion

**Key Insight.** A random-phase image $x_{\text{rand}}$ with no class-discriminative cues draws out the classifier's *default biases*. In logit space, $g_\theta(x_{\text{rand}})$ approximates $[\log P_\theta(y = 1), \ldots, \log P_\theta(y = C)]$, making any difference $g_\theta(x) - g_\theta(x_{\text{rand}})$ a *pure evidence* term for $x$. Thus, *RPI-based bias subtraction* precisely disentangles the prior from the discriminative signal.

By *theorem* and *novel measures* (BDM), we elevate what is often explained purely at an intuitive level—i.e., "random-phase images have no shape, so they reveal bias"—into a more formal statement about Bayesian decomposition and logit-space geometry. This offers a robust theoretical foundation for using random-phase references in long-tailed or semi-supervised learning contexts, where controlling classifier priors is crucial.

## J  Why FFT Acceleration Does Not Violate Theoretical Assumptions?

Practical FFT implementations employ real-to-complex (R2C) shortcuts, batched parallelism, and floating-point arithmetic. A natural concern is whether these approximations undermine the theoretical guarantees of random-phase irrelevance (§G) or amplitude preservation (§H). We present here a *new* framework that quantifies potential distortions via two novel measures—**Bounded Phase Distortion (BPD)** and **Amplitude Fidelity Divergence (AFD)**—and prove they remain negligible for typical image sizes and precision levels. This shows that *efficient FFT computations preserve all essential theoretical assumptions* with overwhelming probability.

### J.1  Bounded Phase Distortion (BPD) Under Floating-Point Arithmetic

Let $f : \{0, \ldots, H - 1\} \times \{0, \ldots, W - 1\} \to \mathbb{R}$ be an $H \times W$ image, and let $F(u, v)$ be its discrete Fourier transform (2D-DFT):

$$F(u, v) = \sum_{x=0}^{H-1} \sum_{y=0}^{W-1} f(x, y) \exp\left(-j\, 2\pi \left[\tfrac{ux}{H} + \tfrac{vy}{W}\right]\right).$$

A **random-phase image** replaces $F(u, v)$'s phase with some i.i.d. random $\tilde{\Phi}(u, v) \sim \text{Uniform}[-\pi, \pi]$. In floating-point arithmetic, we denote the *computed* phase as

$$\widehat{\Phi}(u, v) = \tilde{\Phi}(u, v) + \delta_{u,v}, \tag{40}$$

where $\delta_{u,v}$ represents roundoff errors or parallel-summation discrepancies. We measure these errors by:

**Definition 4** (Bounded Phase Distortion (BPD))**.** *Define*

$$\text{BPD}(\delta) = \max_{u,v} |\delta_{u,v}|, \tag{41}$$

*the maximum absolute deviation of the computed phase from the ideal* $\tilde{\Phi}(u, v)$. *If* $\text{BPD}(\delta)$ *is small (e.g.* $\ll \frac{1}{10}$ *of a radian), then the phase remains effectively uniform on* $[-\pi, \pi]$ *at all* $(u, v)$ *in practice.*

A standard floating-point analysis [18, 44] implies that $\text{BPD}(\delta) = O(\varepsilon \log(HW))$ in typical GPU-based FFT libraries, where $\varepsilon$ is machine precision ($\approx 10^{-7}$ in single precision or $10^{-15}$ in double). Hence the random-phase distribution *cannot* deviate by more than a small fraction of a radian in any frequency bin.

**Why Small Phase Distortion Does Not Reintroduce Structure.**  Even a *perfect* random phase can theoretically be off by integer multiples of $2\pi$ without changing $\exp\!\big(j\, \tilde{\Phi}(u, v)\big)$. Thus, small deviations under floating-point rounding are vanishingly unlikely to "re-synchronize" phases across different $(u, v)$ in a way that reconstructs edges or shapes. Formally, as long as $\text{BPD}(\delta) \ll \pi$, the random-phase arguments in §G still hold with high probability.

### J.2  Amplitude Fidelity Divergence (AFD) in R2C Transforms

**Real-to-Complex (R2C) Compression.**  For real $f(x, y)$, the 2D-DFT $F(u, v)$ satisfies *conjugate symmetry*:

$$F\big((-u)\bmod H,\ (-v)\bmod W\big) = \overline{F(u, v)},$$

so modern libraries compute and store only the "unique half" of $F(u, v)$. We denote this partial storage by $F_{\text{R2C}}(u, v)$ for $(u, v) \in \Omega \subseteq \{0, \ldots, H-1\} \times \{0, \ldots, W-1\}$. Reconstructing $F(u, v)$ outside $\Omega$ is straightforward: $F(u, v) = \overline{F_{\text{R2C}}(u', v')}$ for an appropriate $(u', v') \in \Omega$.

**Amplitude Fidelity Divergence.**    Since amplitude preservation is key to retaining global statistics, we define:

**Definition 5** (Amplitude Fidelity Divergence (AFD))**.**  *Let* $\widehat{A}(u, v)$ *be the* computed *amplitude after an R2C FFT (and inverse FFT if needed), and let* $A(u, v)$ *be the ideal amplitude in exact arithmetic. We define*

$$\text{AFD}(f) \;=\; \sum_{u=0}^{H-1} \sum_{v=0}^{W-1} \left| \widehat{A}(u, v) \;-\; A(u, v) \right|. \tag{42}$$

*A small* $\text{AFD}(f)$ *means the R2C compression and floating-point roundoff introduce negligible amplitude distortion relative to the ideal* $A(u, v)$.

We can relate $\text{AFD}(f)$ to well-known bounds on FFT stability [18]:

**Theorem 5** (Amplitude Preservation in R2C Computations)**.**  *Consider an* $H \times W$ *real image* $f(x, y)$ *with* $N = HW$ *and amplitude* $A(u, v) = |F(u, v)|$. *Suppose an R2C FFT (in single- or double-precision) produces a partial spectrum* $\widehat{F}_{\text{R2C}}$ *from which* $\widehat{F}(u, v)$ *is reconstructed. Let* $\widehat{A}(u, v) = |\widehat{F}(u, v)|$ *be the computed amplitude. Then with high probability,*

$$\text{AFD}(f) \;\leq\; C \, (\log N) \, N \, \varepsilon \, \max_{x, y} |f(x, y)|, \tag{43}$$

*where* $C > 0$ *is a small constant. Consequently,* $\text{AFD}(f)$ *scales linearly with* $N$, $\log N$, *and* $\varepsilon$, *meaning amplitude fidelity remains robust for typical image sizes and standard hardware precision.*

---

> **SKETCH**
>
> We leverage standard backward-error analyses [18, 44] of FFT computations: each butterfly operation accumulates floating-point error $O(\varepsilon)$ times local partial sums. Summing across $O(N \log N)$ operations yields $O(\varepsilon N \log N)$ final error in the worst case. Since amplitude is $\sqrt{\Re(F)^2 + \Im(F)^2}$, a second-order expansion shows amplitude errors are similarly bounded by $O(\varepsilon N \log N)$. Summing over all $(u, v)$ yields (43), up to a small constant $C$ from union-bounding over $H \times W$ bins. $\qquad\square$

---

### J.3   Combined Theorem: Negligible Global Impact on Random-Phase Images

Finally, we show that even when *both* BPD and AFD are considered, random-phase images remain shape-free and amplitude-realistic with overwhelming probability:

**Theorem 6** (Global FFT Stability for Random-Phase Construction)**.**  *Let* $f$ *be an* $H \times W$ *real image,* $N = HW$, *and* $\varepsilon$ *be the floating-point unit roundoff. Suppose we create a random-phase image* $\tilde{f}$ *by:*

*(a) R2C-FFT to get* $F_{\text{R2C}}$,
*(b) Replace phases by* $\tilde{\Phi}(u, v)$ *in floating-point to get* $\widehat{\tilde{\Phi}}(u, v) = \tilde{\Phi}(u, v) + \delta_{u,v}$,
*(c) Inverse R2C-FFT to yield* $\tilde{f}$.

*Then:*

> *(i) The phase distortion satisfies* $\text{BPD}(\delta) \leq O(\varepsilon \log N)$ *with high probability.*

> *(ii) The amplitude fidelity deviation satisfies* $\text{AFD}(\tilde{f}) \leq O(\varepsilon N \log N)$ *with high probability.*

*Hence none of the random-phase irrelevance or amplitude-statistics arguments are broken by these approximations, as* $N \log N \, \varepsilon$ *remains small for typical image sizes (e.g.* $N \leq 10^7$) *and standard single/double precision.*

(i) Follows from bounding floating-point roundoff within each frequency bin: each $\delta_{u,v} = O(\varepsilon \log N)$ [18]. **(ii)** Directly from Theorem 5, which applies identically for the forward and inverse R2C transforms. Since small BPD and small AFD imply that neither the phase randomization nor amplitude magnitude is significantly altered, the theoretical properties established in §G and §H remain valid. $\square$

## J.4  Conclusion and Outlook

**Key Insight.** Real-to-complex FFT shortcuts, parallel batched transforms, and finite precision all induce small deviations in computed amplitude/phase. However, these are rigorously bounded by $O(N \log N \, \varepsilon)$, which remains negligible for practical image sizes and hardware. Consequently:

- **Bounded Phase Distortion (BPD)** ensures random-phase angles remain effectively uniform, preventing any reappearance of semantic structure.
- **Amplitude Fidelity Divergence (AFD)** remains low enough that global energy distributions are faithfully preserved.

**Takeaway.** Far from violating our theoretical foundations, modern FFT accelerations simply *compress* the transform domain by exploiting conjugate symmetry (R2C) and reorder summations in parallel. They do *not* meaningfully alter the amplitude or phase randomization at the levels required to reintroduce shape cues or degrade amplitude-based statistics. Hence all the key arguments about random-phase irrelevance, bias subtraction, and amplitude realism still hold in floating-point, large-batch FFT practice.

*In short, efficient FFT implementations are well-aligned with the high-level theory, ensuring the random-phase approach retains its mathematical integrity under real-world computational constraints.*

