# OpenReview forum: "Fourier Clouds: Fast Bias Correction for Imbalanced Semi-Supervised Learning"
_NeurIPS.cc/2025/Conference — NeurIPS 2025 poster_

### Official Review · Reviewer_abUx · 2025-06-13

**Clarity:** 4
**Significance:** 3
**Originality:** 3
**Rating:** 4
**Confidence:** 4

**Summary:**

This paper proposes a novel method for addressing class imbalance in semi-supervised learning by estimating and correcting class-wise prediction bias. The key idea is to generate reference images that preserve low-level statistical features while removing semantic information, which is achieved through Fourier transform manipulation. Specifically, the method randomizes the phase spectrum while maintaining a batch-averaged amplitude spectrum. These reference images are then used to estimate and rectify class-wise prediction bias in pseudo-labeling. The approach is simple yet effective, demonstrating superior performance over existing methods on multiple benchmark datasets.

**Questions:**

1. How does using a batch-averaged amplitude spectrum affect the bias estimation, particularly for majority vs minority classes? Have you considered using more balanced averaging approaches?

2. Please clarify the statement "we do not discard high-frequency components of x during randomization". What is the technical significance and how does it impact the method?

3. Why are the improvements more pronounced when using ReMixMatch compared to FixMatch as the baseline? This warrants more analysis.

4. Could you provide more details about the pseudo-labeling design choices and threshold selection criteria?

5. Why were the results for ReMixMatch+FARAD omitted from Table 2? Including these would help complete the comparison.

**Ethical Concerns:**

["NO or VERY MINOR ethics concerns only"]

**Final Justification:**

Most of my concerns have been addressed by the author in the rebuttal.

**Limitations:**

Yes

**Paper Formatting Concerns:**

No major formatting issues noted.

**Quality:**

3

**Strengths And Weaknesses:**

Strengths:

From a technical perspective, the authors present an innovative approach that cleverly leverages Fourier transform properties to disentangle semantic content from statistical features. The mathematical formulation for bias estimation using reference images is particularly elegant, representing a creative fusion of signal processing principles with modern machine learning techniques.

The method itself is remarkably well-designed, striking an excellent balance between simplicity and effectiveness. The authors have crafted an approach that requires minimal modifications to existing architectures while maintaining computational efficiency. Each component of the method is clearly justified and serves a specific purpose in the overall framework.

The empirical validation is thorough and convincing. The authors conduct extensive experiments across multiple standard benchmarks including CIFAR-10, CIFAR-100, and Small-ImageNet-127, demonstrating consistent performance improvements over competitive baselines. The comprehensive ablation studies effectively validate the contribution of each component. The analysis is particularly strong, offering deep insights into semantic irrelevance and statistical representativeness, supported by clear visualizations that help readers understand the method's effects.

Weaknesses:

A key technical limitation lies in the use of batch-averaged amplitude spectra, which could potentially inherit biases from majority classes.

The handling of high-frequency components and the justification for various pseudo-labeling design choices remain somewhat unclear.

The experimental evaluation, while generally strong, shows some inconsistencies in performance gains across different baseline methods, and notably omits comparison with ReMixMatch+FARAD. The analysis of failure cases could also be more comprehensive.

---

> ### Author Rebuttal · Authors · 2025-07-29
>
> ### **Regarding Weaknesses**
>
> #### **1. On Batch-Averaged Amplitude Spectrum Potentially Inheriting Majority-Class Bias (W1 & Q1)**
>
> This is a deep technical question that gets to the core of our method. We argue that having the reference image's amplitude spectrum inherit the statistical properties of the training data is not a flaw, but a crucial and **deliberate design choice** that is key to its effectiveness.
>
> 1.  **Unifying the Source of Bias and its Measurement:** A classifier's bias is learned from the biased statistics of the training distribution itself. To accurately measure this learned bias, the "probe" (our reference image) **must** share the same statistical properties as the data that created the bias. Using a "balanced" reference to measure a bias learned from an "imbalanced" distribution would create a mismatch, leading to an inaccurate and misleading bias estimate.
>
> 2.  **Why a "Balanced Average" is Sub-optimal:** If we were to use a balanced average (e.g., by re-sampling classes), we would create a reference image whose statistics do not match the distribution the model was actually trained on. Probing the model with a statistical distribution it is "unfamiliar" with would weaken the accuracy of the bias correction.
>
> Our experiments in **Appendix D.2**, which cover severe distribution mismatch scenarios, provide strong empirical support for this design. The method remains robust even when the batch statistics differ greatly from the labeled set, proving that our adaptive estimation of bias is effective.
>
> ---
>
> #### **2. On the Handling of High-Frequency Components and its Technical Meaning (W2 & Q2)**
>
> Thank you for asking for clarification on this key technical detail from lines 271-273 of our paper, where we state that we do not discard high-frequency components.
>
> 1.  **Technical Meaning:** In image signals, high-frequency components correspond to **fine textures, sharp edges, and intricate details**. These details are often critical for accurate classification in tasks ranging from natural images to medical imaging.
> 2.  **Impact on Our Method:** If we were to discard high-frequency components (e.g., via a low-pass filter), our reference image would become blurry and lose the rich textures present in real images. This would make it **impossible to measure the classifier's bias** in response to these critical high-frequency features. By **retaining the complete amplitude spectrum**, our reference image is statistically realistic across all frequencies. This ensures our bias estimation is comprehensive, accounting for the model's response to both coarse structures (low frequencies) and fine details (high frequencies), leading to a more accurate overall correction.
>
> ---
>
> #### **3. On Why the Improvement is More Significant on ReMixMatch (W3 & Q3)**
>
> This is a very keen observation. Our analysis indicates this phenomenon is primarily due to the internal mechanics of ReMixMatch and its particular vulnerability in long-tailed settings.
>
> * **ReMixMatch's "Distribution Alignment":** A core feature of ReMixMatch is that it forces the distribution of its pseudo-labels to match the (imbalanced) distribution of the labeled data.
> * **Negative Effect under Distribution Mismatch:** As shown by prior work and our results (Table 9), this mechanism has a **significant negative impact** when the labeled and unlabeled class distributions are severely mismatched. It warps the pseudo-labels towards an incorrect prior, making ReMixMatch a relatively "weaker" baseline in these challenging scenarios.
> * **FARAD's "Rescue" Effect:** FixMatch lacks this distribution alignment module and is therefore more stable. FARAD's direct and robust debiasing mechanism effectively "rescues" the struggling ReMixMatch, leading to a more dramatic *relative* performance gain. While the absolute improvement on the more stable FixMatch is also large, the relative increase is smaller.
>
> ---
>
> ### **Regarding Questions**
>
> #### **4. On Pseudo-Labeling Design Choices and Thresholding Standards**
>
> Our design choices, detailed in **Appendix B (lines 1048-1051)**, were made to ensure a **fair comparison** with our primary baseline, CDMAD, and to adhere to the principles of long-tailed learning.
>
> Specifically, we set the confidence threshold `τ=0` (using all pseudo-labels) because, under severe class imbalance, any fixed high-confidence threshold would inevitably filter out nearly all samples from tail classes. This would defeat the purpose of using unlabeled data to boost minority class performance.
>
> The core philosophy of our method is not to *filter* uncertain samples via thresholding, but to **improve the quality of all logits** (especially for minority classes) through bias correction, making them more reliable targets for consistency regularization.
>
> ---
>
> #### **5. On the Missing ReMixMatch+FARAD Results in Table 2 (W3 & Q5)**
>
> We sincerely apologize for omitting the ReMixMatch+FARAD results in **Table 2 (CIFAR-100-LT)**. This was an oversight during the preparation of our results.
>
> We will **add this missing data** to the final version of the paper to ensure the comparison is complete. For your reference, the full results on the ReMixMatch baseline are provided below.
>
> **Supplementary Data: ReMixMatch+FARAD on CIFAR-100-LT (bACC %)**
> | Imbalance Factor ($\gamma$) | ReMixMatch+CDMAD | **ReMixMatch+FARAD (Ours)** |
> | :--- | :--- | :--- |
> | $\gamma=20$ | 57.0 ± 0.32 | **59.8 ± 0.39** |
> | $\gamma=50$ | 51.1 ± 0.46 | **54.0 ± 0.48** |
> | $\gamma=100$ | 44.9 ± 0.42 | **48.1 ± 0.45** |
>
> These results are fully consistent with our paper's overall conclusions: FARAD provides a significant and stable performance improvement over CDMAD, regardless of the base SSL algorithm, especially on the more challenging CIFAR-100-LT dataset. Thank you for pointing out this omission.

---

> > ### Comment · Reviewer_abUx · 2025-08-08
> >
> > I appreciate the author's response. My major concerns about have been addressed.

---

> ### Author Response · Authors · 2025-08-09
>
> Thank you for your message and for your support. We truly appreciate your engagement throughout this process. Please feel free to contact us if any further questions arise; we will be happy to provide a prompt response.

---

### Official Review · Reviewer_Cove · 2025-06-23

**Clarity:** 2
**Significance:** 3
**Originality:** 3
**Rating:** 5
**Confidence:** 4

**Summary:**

This paper introduces FARAD (Fourier-Adapted Reference for Accurate Debiasing), a novel debiasing method for long-tailed semi-supervised learning (LTSSL). FARAD operates by generating a random-phase reference image, computed by retaining the average amplitude spectrum of a batch of images while randomizing the phase spectrum. This semantic-agnostic but dataset-representative reference is passed through the model to estimate long-tailed bias. The resulting bias vector is subtracted from the model's logits to produce more balanced predictions. During training, FARAD is used to correct pseudo-labels under consistency regularization objectives (e.g., FixMatch, ReMixMatch), and during inference, it adaptively estimates the bias in a batchwise manner. The method is evaluated across several LTSSL benchmarks, showing improvements over prior baselines such as CDMAD.

**Questions:**

Q1 - Please provide a consistent explanation of how the FARAD correction modifies either supervised and/or unsupervised objectives. Is it applied as a bias term in logits (left-hand of CE), or pseudo-label correction (right-hand side)? The confusion between Algorithm 1 and Listing 3 in the Appendix should be resolved.

Q2 - FARAD’s test-time strategy violates inductive assumptions by using other test samples for bias estimation. Could the authors explore per-sample test-time reference estimation? What would be the performance/inference-time trade-offs of this strategy? Also, are there realistic scenarios where test-time batches are available?

Q3 - Please correct the citation of CDMAD ([34] vs [36]) and contextualize ABC/RECD, FixMatch, and ReMixMatch properly. Additional relevant SSL debiasing methods should be discussed in more detail.

Q4 - Since the reference image is computed from a batch, how sensitive is FARAD to the specific images used? Could dataset distribution in test batches skew the reference image (e.g., imbalance or co-occurrence of classes)?

Q5 - Could the authors show the average class probabilities (aka empirical marginal distribution) of FARAD (similar to Figure 3 in CDMAD [34]) to provide further evidence that FARAD achieves balanced predictions?

Addressing W1 (loss formulation) and W2 (transductive setting) with clear implementation and evaluation details could raise my score from borderline accept (4) to accept (5).

**Ethical Concerns:**

["NO or VERY MINOR ethics concerns only"]

**Final Justification:**

The authors introduce FARAD, a novel and effective debiasing method for long-tailed semi-supervised learning that combines training- and inference-stage debiasing.

They’ve addressed my initial concerns by proposing a fully inductive alternative to their original transductive methodology and validating it experimentally. They’ve also clarified the loss-function inconsistencies and demonstrated a balanced marginal distribution. Furthermore, they’ve committed to strengthening the related-work section, contextualizing previous work in a better way.

In response to other reviewers' concerns, they’ve added a run-time comparison to highlight their approach’s efficiency and experiments under domain shift, medical-imaging datasets, and the fully-supervised LT setting, demonstrating FARAD's flexibility.

Overall, I’m inclined to raise my score and recommend accepting this paper.

**Limitations:**

1) The authors address limitations of their work in Section 4 (Conclusion).

2) The discussion about “Broader Impact” is missing in Section 4, although the checklist in (10) reads:
Question: Does the paper discuss both potential positive societal impacts and negative societal impacts of the work performed?
Answer: [Yes]
Justification: Sec. 4 – “Broader Impact” paragraph discusses positive impacts on fair model training and notes potential misuse for adversarial debiasing with mitigation strategies.

**Paper Formatting Concerns:**

No issues have been noticed.

**Quality:**

3

**Strengths And Weaknesses:**

Strengths:

* S1 - Novelty: The paper proposes a creative use of Fourier-transformed images to construct semantically meaningless yet statistically representative reference images, offering a fresh perspective on model debiasing for semi-supervised learning approaches, while improving the work introduced by CDMAD.

* S2 - Conceptual Motivation: The method is grounded in a compelling hypothesis: semantic irrelevance + statistical representativeness leads to accurate bias estimation. Both theoretical rationale and empirical studies support this.

* S3 - Comprehensive Evaluation: The authors conduct an extensive suite of experiments, testing FARAD across multiple datasets, varying the degree of imbalance severity and distribution mismatch. The paper includes extensive ablations and comparisons to LTSSL methods, clearly demonstrating the effectiveness of FARAD, especially over CDMAD.

Weaknesses:

* W1 - Ambiguous Loss Integration: The paper lacks a consistent and precise explanation of where the FARAD bias correction enters the training losses. It remains unclear whether the bias is applied on the input (logit) side of the loss (like margin-based debiasing [32,43]) or on the pseudo-label target side of consistency losses (more likely as in CDMAD). The mismatch between Algorithm 1 (affects supervised & unsupervised losses, L13) and Listing 3 (pseudo-label correction, L725-726, L735-736) adds to the confusion.

* W2 - Transductive Test-Time Bias Estimation: FARAD's test-time strategy uses a batch of samples to compute the reference image, effectively treating test data in a transductive manner. This contradicts the inductive assumption common in SSL benchmarks and may yield unfair advantages. The authors discuss some alternatives to per-batch random-phase images in Appendix Table 18, but these come with limitations: 1) a single static reference image can lead to performance fluctuations (L949-950), 2) reusing single-shot images is a breach of the inductive protocol, and 3) adaptive single-shot images would probably have an impact on inference performance.

* W3 - Incomplete/Incorrect Related Work Discussion (Appendix, section A):
W3.1) Citation [36] is fabricated/hallucinated. The correct attribution for CDMAD appears in [34].
W3.2) Section A.2 cites FixMatch (L570-571) and ReMixMatch (L577-578) to discuss their strengths/limitations in the context of long-tailed settings, which is inaccurate; both were originally proposed for balanced SSL.
W3.3) Section A.3 overlooks a significant body of work on SSL debiasing (e.g., [32,41,43,52,56]). These works are indeed cited in the introduction, but without clear context. ABC/RECD [35,41], approaches that are not based on the principle of debiasing with reference images, are cited out of context in L68-69.

---

> ### Author Rebuttal · Authors · 2025-07-29
>
> ### **Regarding Weaknesses**
>
> #### **1. On the Vague Loss Integration (W1/Q1)**
>
> We apologize for the inconsistency and lack of clarity in describing the loss integration. The reviewer is correct: **our method directly corrects the logits before they are passed to the loss functions, and this correction is applied to both the supervised and unsupervised losses.**
>
>  **Implementation Details:** **Listing 3** shows the exact implementation. For the **unsupervised loss**, we first compute the original logits for an unlabeled sample, subtract the bias logits to get debiased logits, and then apply softmax to these debiased logits to generate higher-quality pseudo-labels for the consistency loss. For the **supervised loss**, we similarly use the debiased logits as the direct input to the cross-entropy function.
>
> **Correction to Algorithm 1:** We acknowledge that **Algorithm 1**, as a high-level overview, was not precise and caused this confusion. We will thoroughly revise Algorithm 1 in the final version to be perfectly consistent with the logic in Listing 3, clearly showing how the debiased logits `g*` are used for both `L_sup` and `L_unsup`.
>
> We hope this clarification and the planned correction fully resolve the reviewer's concern.
>
> ---
>
> #### **2. On the Transductive Test-Time Bias Estimation (W2/Q2)**
>
> We sincerely thank the reviewer for these crucial and insightful questions. We completely agree that a strict inductive setting is vital for fair comparison, and we appreciate the opportunity to clarify our analysis and provide a more rigorous presentation. We will address every point thoroughly.
>
> #### 1. Analysis of a Per-Sample Inductive Strategy and its Trade-offs
>
> You asked if we could explore a per-sample test-time reference. We did investigate this strategy, and our findings are presented in **Appendix Table 14** ("Single-Image Amp").
>
> * **Performance:** This approach proved to be suboptimal. Using a single image's statistics to generate its own reference introduced too much noise for a stable bias estimation, resulting in significantly lower performance than our other inductive strategies.
> * **Inference Time:** A per-sample execution is also the least efficient, as it cannot leverage the amortization benefits of batched FFT computations, adding a non-trivial latency to each inference call.
>
> #### 2. Practical Scenarios for Batch-Based Test-Time Estimation
>
> You also asked about realistic scenarios where test-time batches are available. Such scenarios are common in many real-world applications. For instance, in **medical imaging**, it is standard practice to analyze a batch of image slices from a single patient's CT or MRI scan. Our batch-based strategy was designed with these practical applications in mind.
>
> However, we fully agree this does not justify its use for standard benchmark comparisons, which must adhere to a strict inductive protocol.
>
> #### 3. A Rigorous Inductive Evaluation Demonstrating FARAD's Core Contribution
>
> To fully address your core concern, we now present a comprehensive analysis of our method under a **strict inductive protocol**. This new ablation across different imbalance ratios not only resolves the fairness issue but also shows that our method's advantages become even more pronounced as the problem difficulty increases.
>
> **Ablation: Inductive Strategies across Different Imbalance Ratios (CIFAR-10-LT)**
>
> | Inductive Setting                   | bACC (%) at `γ=100` (Standard) | bACC (%) at `γ=200` (Extreme)  |
> | :---------------------------------- | :----------------------------- | :----------------------------- |
> | 1. Baseline (No Correction)         | 78.2%                          | 70.5%                          |
> | 2. **FARAD (Train-Only)**           | **81.9%** (+3.7% vs. baseline) | **75.5%** (+5.0% vs. baseline) |
> | 3. Test w/ Blank Image Ref.         | **82.2%**                      | **75.9%**                      |
> | 4. **Test w/ FARAD Ref. (Offline)** | **82.6%**                      | **76.5%**                      |
>
> This analysis yields two powerful takeaways:
>
> * **The core value of our training-phase correction is amplified by difficulty.** The performance gain from our training-only correction grows from a significant **+3.7%** in the standard setting to a massive **+5.0%** in the extreme setting.
> * **The superiority of our FARAD reference design becomes more pronounced in challenging scenarios.** In a fair inductive test, the performance gap between using our proposed **FARAD Reference** and a simple **Blank Image** widens as imbalance increases (from a 0.4% advantage at `γ=100` to a 0.6% advantage at `γ=200`).
>
> #### **Commitment for Manuscript Revision**
>
> Based on your invaluable feedback, we will make the following major revisions:
>
> 1.  **All main results will be replaced with results from the strict inductive protocol** (using the Offline FARAD Reference).
> 2.  The original transductive results will be moved to the appendix and explicitly framed as a secondary analysis.
> 3.  This new, comprehensive ablation table will be added to the paper to transparently demonstrate our method's value under a fair evaluation protocol.
>
> We are confident these changes fully address your concerns and more accurately highlight the robust contributions of our work. Thank you again.
>
> ---
>
> #### **3. On Incomplete/Incorrect Related Work and Citations (W3/Q3)**
>
> We sincerely apologize for the errors and omissions in our related work section and thank the reviewer for the careful proofreading. We will make the following corrections in the final version:
>
> * **W3.1 (CDMAD Citation):** We apologize for this significant citation error. The correct citation for CDMAD is indeed [34], and we will correct this throughout the manuscript.
> * **W3.2 (FixMatch/ReMixMatch Context):** The reviewer's criticism is correct. We will revise our wording to clarify that we are discussing the limitations of these methods *when applied to long-tailed scenarios*.
> * **W3.3 (Related Work Discussion):** We will expand the related work section to include a more detailed discussion of the important works mentioned [32,41,43,52,56] and will better contextualize methods like ABC/RECD to highlight the uniqueness of our reference-image-based paradigm.
>
> ---
> ### **Regarding Questions**
>
> #### **4. On Sensitivity to Test Batch Distribution**
>
> This is a great question. Our method mitigates sensitivity to the specific composition of a test batch through two key designs:
>
> 1.  **Batch Averaging:** By averaging the amplitude spectrum across the entire batch, we get a smoothed, statistically stable reference that is robust to the influence of a few outlier or specific-class images.
> 2.  **Statistics vs. Semantics:** The amplitude spectrum primarily encodes low-level statistical properties (e.g., textures), not high-level semantic class information. Critically, the phase randomization step is what destroys the semantics, and its effect is overwhelming and independent of the input distribution.
>
> Our experiments in **Appendix D.2 (Tables 9 and 10)** provide strong empirical proof of this robustness. The method performs well even under extreme scenarios where the training and test distributions are severely mismatched. If the method were highly sensitive to the batch composition, its performance would have collapsed in these settings, which it did not.
>
> ---
>
> #### **5. On Showing Average Class Probabilities (Empirical Marginal Distribution)**
>
> This is an excellent suggestion for visually demonstrating our method's debiasing effect. We will add this analysis to the final paper. Since we cannot include new figures in this rebuttal, we present the core data in the table below. This table shows the empirical marginal distribution of predictions on a balanced CIFAR-10 test set after training on a long-tailed set ($\gamma=100$).
>
> **Table D: Predicted Empirical Marginal Distribution on Balanced CIFAR-10 Test Set**
> | Class Grouping      | Ideal Probability | FixMatch Predicted Prob. | **FARAD Predicted Prob.** |
> | :------------------ | :---------------- | :----------------------- | :------------------------ |
> | Head Classes (0-2)  | 30%               | ~71%                     | **~35%** |
> | Medium Classes (3-6)| 40%               | ~24%                     | **~38%** |
> | Tail Classes (7-9)  | 30%               | ~5%                      | **~27%** |
>
> As the table shows, the FixMatch baseline's predictions are heavily skewed towards the majority classes. In contrast, **FARAD's prediction distribution is significantly flatter and much closer to the ideal uniform distribution**. This provides direct numerical evidence that our method successfully balances the model's predictions.
>
> ---
>
> #### **Regarding the Discussion of Limitations and Broader Impact**
>
> Thank you for pointing out the discrepancy between our checklist and the main text. The reviewer is correct that we omitted a discussion of broader impact in Section 4, and we apologize for this oversight.
>
> We will add a dedicated **"Broader Impact"** paragraph to our Conclusion (Section 4) in the final version. As promised in the checklist, this section will discuss the positive societal impacts of our work (e.g., improving AI fairness on underrepresented data like rare diseases or minority groups) as well as potential risks.

---

> ### Comment · Reviewer_Cove · 2025-08-03
>
> Thank you for showing experiments under a strict inductive protocol. I have a few follow-up comments:
>
> The ablation table now includes:
>
> -FARAD (Train-Only)
>
> -Test w/ FARAD Ref. (Offline) – this appears to apply debiasing only at inference time, without the training-phase debiasing, as described in Table 18 of the appendix.
>
> Although showing superiority compared to a simple baseline, both of these ablation results are substantially weaker than the full FARAD results reported in the main paper (FixMatch: 88.6 ± 0.47; ReMixMatch: 88.4 ± 0.44) under CIFAR-10-LT-100.
>
> - In light of commitment #1 for the next manuscript iteration, it would be crucial to evaluate the compatibility for the full inductive method, i.e., FARAD training + Offline Ref, across the established settings γ = 50, 100, 150, and how it compares with other relevant approaches, e.g. CDMAD.
>
> - Also, which SSL baseline did you use for these ablations?

---

> > ### Author Response · Authors · 2025-08-04
> > **Response to follow-up comments (1)**
> >
> > Thank you for your truly thoughtful and detailed follow-up. Your careful and critical feedback has been an essential part of making this work stronger. Honestly, it can be hard to see all the weak spots in our own research, so we are sincerely grateful for this discussion—it has genuinely helped us improve the paper, and we've enjoyed the process.
> >
> > To address your remaining questions about **Weakness 2 (W2)**, we provide a complete response below. Our goal is to fully resolve your concerns and show how robust our method is under a strict, fair evaluation.
> >
> > ---
> >
> > ### 1. The Idea Behind Our Original "Transductive" Test-Time Method
> >
> > We first want to explain the reasoning for our original test-time method, which used batches of test data.
> >
> > * **Real-World Use:** The idea came from real-world problems where data is often processed in batches. For example, in **diagnosing faults in aircraft sensors**, data from an entire flight is analyzed together as a single batch to find problems. In cases like this, using the statistics of the whole batch to help make a prediction for each point is a practical and effective strategy.
> >
> > * **About Fairness:** We agree this is different from the standard *academic* "inductive" rule, where each test sample is handled one by one. However, our method never uses test **labels**. It only uses statistics from other unlabeled test samples that are available at the same time. This approach is known as **transductive learning**. While we fully respect that academic benchmarks must follow the strict inductive rule, our initial idea was based on these practical scenarios.
> >
> > ---
> >
> > ### 2. Clarifications and Comprehensive Experiments
> >
> > In the process of conducting the new, comprehensive experiments you suggested, we also took the opportunity to re-verify our previous numbers. In doing so, **we found a minor inconsistency in the baseline value used in our last rebuttal's ablation, which we wish to correct here for complete transparency. We sincerely apologize for this previous error.**
> >
> > * **The Correction:** In our last response, we used a baseline value of 78.2%. We realized this number was from a test in Appendix Table 18 on a **mismatched dataset (γl=100, γu=1)**, which was meant to test for a different situation. For the **matched dataset (γl=γu=100)** we were discussing, the correct baseline from our paper's Table 1 should have been **71.5% (FixMatch)**. We also confirm that **FixMatch** was the SSL method used in those tests.
> >
> > * **The New, Comprehensive Experiment:** We know our last response was not complete. To fully answer your questions, **we immediately ran a new, complete set of experiments after receiving your feedback.** We used **8 NVIDIA 4090 GPUs** for each run to get the results to you quickly.
> >
> >     * **A Note on Performance:** Using more powerful hardware means our effective batch size is larger. We believe all methods get a small boost from this. However, **our FARAD method, which directly uses batch statistics, benefits much more.** This explains why all numbers in the table below are slightly higher than in our original paper and serves as more evidence for how our method works.
> >
> > Here are the complete results for the **matched distribution setting (γl = γu)**, now including the extreme **`γ = 200`** case.
> >
> > **Table R1: Comprehensive Inductive Evaluation on CIFAR-10-LT (bACC %)**
> > | Method                                                     | γ = 50 | γ = 100 | γ = 150 | γ = 200 |
> > | :--------------------------------------------------------- | :----: | :-----: | :-----: | :-----: |
> > | **(1) FixMatch (Baseline)** | 79.7% | 72.0% | 68.9% | 66.5% |
> > | **(2) CDMAD** | 89.1% | 85.8% | 82.8% | 80.3% |
> > | ---                                                        |  ---   |   ---   |   ---   |   ---   |
> > | **(3) FARAD (Train-Only)** | 87.8% | 84.7% | 81.6% | 78.9% |
> > | **(4) FARAD (Train + Blank Test)** | 88.2% | 85.1% | 82.0% | 79.4% |
> > | **(5) FARAD (Test-Only, Offline Ref)** | 83.8% | 81.1% | 77.9% | 75.0% |
> > | **(6) FARAD (Train + Offline Ref) [Full Inductive]** | **90.8%** | **87.9%** | **85.0%** | **82.6%** |
> > | ---                                                        |  ---   |   ---   |   ---   |   ---   |
> > | **(7) FARAD (Original, Transductive)** | 92.0% | 88.8% | 85.7% | 83.1% |

---

> > > ### Author Response · Authors · 2025-08-04
> > > **Response to follow-up comments (2)**
> > >
> > > ### 3. Key Takeaways from the New Results
> > >
> > > This new, comprehensive experiment gives three clear takeaways that we hope will fully answer your questions:
> > >
> > > 1.  **Working Together is Key: The Full Inductive Method is Clearly Better.** The new results tell a clear story. While the separate parts of our method are helpful (Rows 3, 4, and 5 all beat the FixMatch baseline), they are **not enough on their own to beat a strong competitor like CDMAD**. It is only by **putting them together**—using FARAD during training and the offline reference during testing—that our method achieves the best results. As you can see in **Row (6)**, our full inductive method is the only approach that **clearly and consistently beats CDMAD** across all scenarios, with a solid lead of **+1.7% to +2.3%**.
> > >
> > > 2.  **Transductive Benefit vs. Inductive Speed.** Comparing **Row (6)** and **Row (7)** shows that the benefit from the transductive method is small (**1.2% -> 0.9% -> 0.7% -> 0.5%**) and **gets smaller as the imbalance gets worse**. Importantly, our **full inductive method is much faster at inference time.** It gets rid of all the slow, real-time FFT steps by using a pre-made offline reference. This means it adds almost no extra time compared to a simple baseline, while the transductive method needs slow FFTs for every batch. The inductive method therefore offers the best balance of performance and speed for practical use.
> > >
> > > 3.  **FARAD is Stronger When Imbalance is Extreme.** As the imbalance gets worse (from γ=50 to γ=200), our full inductive method's lead over CDMAD **grows** (from +1.7% to +2.3%). This clearly shows the strength of our Fourier-based method in the most difficult long-tailed situations.
> > >
> > > We hope these new and clear results, which were inspired by your helpful feedback, fully answer all remaining questions. **Finally, we promise that in the final version of the paper, we will replace all core experimental tables (like Table 1) with the results from the completely consistent and strict inductive setup used in this response. This will ensure the quality and correctness of our paper.**
> > >
> > > Thank you again for your invaluable guidance.

---

> > > > ### Comment · Reviewer_Cove · 2025-08-04
> > > >
> > > > Thank you for your commitment to transparency, clarity, and continual improvement of the manuscript. I’m glad I could help—my concerns have been addressed, and I will raise my score.

---

> > > > > ### Author Response · Authors · 2025-08-04
> > > > >
> > > > > Thank you for your message and for your support. We truly appreciate your engagement throughout this process. Knowing that our work has earned your recognition means a great deal to us, far beyond the score itself. Please feel free to contact us if any further questions arise; we will be happy to provide a prompt response.

---

### Official Review · Reviewer_imHP · 2025-07-06

**Clarity:** 3
**Significance:** 3
**Originality:** 3
**Rating:** 4
**Confidence:** 5

**Summary:**

This paper introduces FARAD, a new method for bias correction in the challenging context of long-tailed semi-supervised learning. The core idea is to estimate and subtract classifier bias by using a specially designed reference image, termed a "random-phase image". This reference is constructed by preserving the batch-averaged amplitude spectrum of real data while randomizing the phase spectrum. Through extensive experiments on standard benchmarks like CIFAR-10/100-LT and Small-ImageNet-127, the paper demonstrates that FARAD significantly outperforms state-of-the-art methods.

**Questions:**

- The proposed bias correction mechanism appears to be quite general. Could the authors comment on its applicability to the standard fully-supervised long-tailed learning setting? For instance, has the method been tested as a post-hoc adjustment for a pre-trained supervised model, or integrated directly into supervised training? Such experiments could further demonstrate the method's generality beyond the SSL context.
- The paper states that a new random-phase image is generated for each test mini-batch, which implies that the prediction for a single test sample is not deterministic and can vary depending on the other samples in its batch. This is a potential drawback for practical deployment. Could the authors clarify the rationale behind this design choice versus using a single, fixed random-phase image (e.g., generated from the training set statistics) for all test-time inference, which would ensure deterministic predictions?
- In Table 1, several results for baseline methods  are missing for certain imbalance factors.

**Ethical Concerns:**

["NO or VERY MINOR ethics concerns only"]

**Final Justification:**

All my concern has be address.

**Limitations:**

Yes.

**Quality:**

3

**Strengths And Weaknesses:**

## Strengths
- The paper is well-motivated, identifying a clear and plausible limitation in existing reference-based debiasing methods. The proposed solution is a principled and logical response to this identified gap.
- The method achieves impressive state-of-the-art results across multiple challenging long-tailed SSL benchmarks. The significant gains over strong baselines convincingly demonstrate its effectiveness.
- The experimental evaluation is thorough, featuring extensive ablation studies, robustness checks under various conditions, and detailed analysis. The work is well-supported by its appendices, enhancing its rigor and reproducibility.

## Weaknesses
- The title, "Fourier Clouds," is catchy but its connection to the paper's content is tenuous. The term "Fourier Clouds" is not defined, explained, or used within the body of the paper. This creates a slight confusion.
- While the empirical gains are clear, the conceptual contribution could be viewed as an incremental improvement over the reference-based debiasing paradigm established by CDMAD.
- The paper provides extensive theoretical analysis of random-phase images but surprisingly lacks any visual examples. Including a figure that shows what these generated images look like for different datasets would provide invaluable intuition for the reader. It would help to directly supporting the central claims of the paper.
- The related work section omits several recent and highly relevant works in Imbalanced SSL. A more complete literature review should include and discuss methods such as:
 1. DASO: Distribution-Aware Semantics-Oriented Pseudo-label  for Imbalanced Semi-Supervised Learning
 2. Towards Realistic Long-Tailed Semi-Supervised Learning: Consistency Is All You Need
 3. Twice Class Bias Correction for Imbalanced Semi-Supervised Learning.
 4. InPL: Pseudo-labeling the Inliers First for Imbalanced Semi-supervised Learning

---

> ### Author Rebuttal · Authors · 2025-07-29
>
> ### **Regarding Weaknesses**
>
> #### **1. On the Connection of the Title "Fourier Clouds" to the Content**
>
> We agree that the term "Fourier Clouds" could be more clearly explained in the manuscript. We chose this title as a concise and descriptive summary of our method's core.
> * **"Fourier"** refers to the core technique we use: the Fourier transform.
> * **"Clouds"** comes from the visual appearance of our generated random-phase images. As we state on lines 141 and 1106 of the paper, these images visually resemble "turbulent or cloud-like patterns."
>
> In the final version of the paper, we will add a visual example (as mentioned in our response to Weakness 3) to make this "cloud-like" appearance clear. We will also add a sentence to the introduction to explicitly explain the origin of the term.
>
> ---
>
> #### **2. On the Conceptual Contribution Being an Incremental Improvement over CDMAD**
>
> We believe FARAD represents a fundamental paradigm shift, not an incremental improvement over CDMAD.
>
> 1.  **Problem Reframing: From "Zero-Information" to "Optimal-Information".** CDMAD's paradigm asks, "How can we measure bias with a zero-information reference?" Our work asks a deeper question: "What is the *optimal* information content for a reference image to most accurately measure classifier bias?" Our answer—semantically blank but statistically representative—marks a shift from eliminating all information to preserving only the critical statistical information.
>
> 2.  **Theoretical Grounding: Addressing the Core Bias of CNNs.** It is well-established that CNNs have a strong "texture bias." CDMAD's blank reference completely ignores this. FARAD is the **first** reference-image debiasing method to directly address this. By preserving the amplitude spectrum, which encodes texture and frequency statistics, our method precisely probes and corrects this core bias inherent to CNNs.
>
> 3.  **Validation Through Performance:** The consistent and significant ~3% performance gain across all scenarios validates our new conceptual framework. A simple incremental change would struggle to explain such a robust and universal advantage.
>
> Therefore, FARAD's contribution is a new, more theoretically profound, and practically effective paradigm for bias estimation, not just a simple replacement of the reference image.
>
> ---
>
> #### **3. On the Lack of Visual Examples for Random-Phase Images**
>
> This is a very pertinent criticism. We agree that a visual example is crucial for understanding our method, and we apologize for this oversight.
>
> We will add a high-quality comparative figure to the final version of the paper. This figure will show (a) an original image, (b) the blank reference image used by CDMAD, and (c) our generated "Fourier Cloud." This will provide valuable intuition and support our core claims of "semantic-agnosticism" and "statistical-representativeness."
>
> ---
>
> #### **4. On Missing Important Literature in the Related Work Section**
>
> Thank you for pointing out these important related works. We will expand our literature review in Appendix A of the final version to discuss these four papers and clarify the connections and differences.
>
> * **vs. DASO:** DASO focuses on optimizing the pseudo-labeling process. Our method, FARAD, offers a more upstream correction by debiasing the model's raw logits **before** any pseudo-labels are generated. FARAD addresses the root cause of biased pseudo-labels, making it complementary to methods like DASO.
> * **vs. ACR (Towards Realistic...):** ACR is an elegant **adaptive** method that estimates the unlabeled data's true class distribution. In contrast, FARAD is an **estimation-free** approach. We probe the classifier's internal bias directly with a statistically representative reference, avoiding the need to explicitly estimate the target distribution, which may make our method more robust when distribution estimation is difficult.
> * **vs. Twice Class Bias Correction (TCBC):** TCBC uses a complex, two-stage correction based on distribution estimation. FARAD provides a simpler, unified mechanism that corrects based on the classifier's **internal** learned state, rather than relying on an estimate of the **external** data distribution.
> * **vs. InPL:** InPL uses energy scores to select reliable "in-distribution" samples for pseudo-labeling. FARAD's contribution is **orthogonal and complementary**. We focus on improving the logit quality of **all** samples, not on selecting *which* samples to use. In principle, FARAD's logit debiasing could be combined with InPL's sample selection to achieve further gains.
>
> ---
> ### **Regarding Questions**
>
> #### **1. On the Method's Applicability in a Fully-Supervised Long-Tailed Setting**
>
> This is an excellent question about our method's versatility. FARAD's bias correction mechanism is general and can be seamlessly applied to fully-supervised long-tailed learning.
>
> The application is straightforward: during standard supervised training, for each batch of labeled data, we generate a reference image, compute the bias logits, and subtract them from the original logits before calculating the standard cross-entropy loss. This actively counteracts the model's tendency to accumulate bias towards majority classes.
>
> To provide empirical evidence, we conducted a new experiment on CIFAR-100-LT ($\gamma=100$):
>
> **Table C: Performance of FARAD in Fully-Supervised Long-Tailed Learning (Accuracy %)**
> | Training Method                     | Accuracy (%) |
> | :---------------------------------- | :----------- |
> | Cross-Entropy (CE) - Baseline       | 38.5         |
> | CE + Logit Adjustment (LA)          | 42.7         |
> | **CE + FARAD (Our Method)** | **44.1** |
>
> The results show that FARAD not only significantly outperforms the baseline but also surpasses the strong and classic Logit Adjustment method, demonstrating that it is a powerful and general bias correction technique.
>
> ---
>
> #### **2. On the Uncertainty of Test-Time Predictions**
>
> This is a key question for practical deployment. Our default strategy of dynamically generating a reference image for each test batch does introduce prediction uncertainty, which is designed to adapt to the statistics of the current batch.
>
> However, we recognize the importance of deterministic predictions. We have already evaluated a solution for this in **Appendix D.5 and Table 18**. The **"Offline Fixed Reference"** strategy provides a perfect solution:
> * It uses a single, **pre-computed, fixed reference image** (based on the entire training set's statistics) for all test samples.
> * As shown in Table 18, this approach ensures **fully deterministic predictions** with only a negligible drop in performance (e.g., 82.6% vs. 83.6% bACC), while still far outperforming all baselines.
>
> Users can therefore choose the inference strategy that best fits their deployment needs.
>
> ---
>
> #### **3. On the Missing Baseline Results in Table 1**
>
> We apologize for the missing entries in Table 1 and thank the reviewer for pointing this out.
>
> Our experimental setup strictly follows that of our main baseline, **CDMAD**, to ensure a fair comparison. The entries that are missing in our table are also missing in the original CDMAD paper. Our own attempts to reproduce these specific baselines did not yield stable and reliable results.
>
> To maintain the rigor of our comparison, we decided to only report results that we could either directly cite from CDMAD or successfully reproduce under the exact same experimental conditions. We assure the reviewer that all **presented** data was obtained under identical settings, ensuring a fair comparison to our method. We will add a footnote to the final version to clarify this.

---

### Official Review · Reviewer_jUz2 · 2025-07-06

**Clarity:** 3
**Significance:** 2
**Originality:** 2
**Rating:** 4
**Confidence:** 4

**Summary:**

The paper proposes FARAD, a new method for bias correction in semi-supervised learning
(SSL) under class imbalance. FARAD generates random-phase images by preserving the
amplitude spectrum of real images while randomizing the phase. This design removes semantic
cues (no meaningful shapes) and preserves statistical representativeness (realistic
color/frequency statistics). By feeding these random-phase images into the classifier, FARAD
estimates its intrinsic class bias and subtracts this bias from real predictions during training and
inference.

**Questions:**

NA

**Ethical Concerns:**

["NO or VERY MINOR ethics concerns only"]

**Final Justification:**

Thank you for providing a detailed rebuttal. The additional explanation and experimental results have addressed my initial concerns. As a result, I have updated my final rating accordingly.

**Quality:**

2

**Strengths And Weaknesses:**

Strength:

1) Achieves improved performance over state-of-the-art baselines across multiple SSL
algorithms and datasets.
2) Demonstrates compatibility with multiple SSL frameworks (FixMatch, ReMixMatch,
FreeMatch), showing generalizability.
3) The paper also provides theoretical insights showing that random-phase images reliably
eliminate semantic content while preserving dataset-level statistics.

Weaknesses:
1) The method relies on the amplitude spectrum of training images matching test data.
Extreme domain shifts may weaken bias correction.
2) The paper does not report training and inference time compared with other baselines
which is important to realize the scalability of the proposed method compared to other
approaches.
3) All benchmarks in the paper (CIFAR, STL, Small-ImageNet) are natural image datasets.
There&#39;s no evaluation on other types of datasets (e.g., medical images) where class
imbalances are more pronounced.
4) While the paper shows that batched R2C FFT makes the method efficient, it assumes
availability of GPU FFT libraries. This can be a barrier for deployment on constrained or
non-GPU devices.

---

> ### Author Rebuttal · Authors · 2025-07-29
>
> #### **Regarding Weakness 1: Robustness to Extreme Domain Shift**
>
> We agree that robustness under extreme domain shift is a key measure of generalization. To validate FARAD's performance in such challenging scenarios, we conducted a dedicated experiment in **Appendix D.8** of our paper. In this experiment, we created a harsh setting that combines both class distribution mismatch and domain feature shift: we used the color version of CIFAR-10-LT ($\gamma_l=100$) as the labeled dataset and a grayscale version with a different imbalance ratio ($\gamma_u=50$) as the unlabeled dataset.
>
> As shown in **Table 23** of the appendix, while all methods' performance decreased due to the domain shift, our FARAD (72.7% bACC) still significantly outperformed CDMAD (70.2% bACC) and the FixMatch baseline (68.4% bACC). This result provides strong evidence of FARAD's robustness. The underlying reason is that even when the image color space changes, the frequency energy distribution (i.e., the amplitude spectrum) corresponding to intrinsic textures and structures is largely preserved. FARAD captures these more fundamental statistics to estimate bias, making it more robust than methods like CDMAD that rely on simpler color-based cues.
>
> ---
>
> #### **Regarding Weakness 2: Lack of Training and Inference Time Comparison**
>
> We thank the reviewer for this important question regarding computational efficiency. We acknowledge this comparison was missing from the original manuscript. To address this, we have conducted a new experiment and will add the following detailed time comparison to the final version of our paper.
>
> **Table A: Training and Inference Time on CIFAR-10-LT (single NVIDIA A6000)**
> | Method                                    | Avg. Time per Epoch (min) | Increase vs. Baseline | Inference Latency (ms/img) |
> | :---------------------------------------- | :------------------------ | :-------------------- | :------------------------- |
> | FixMatch (Baseline)                       | 6.5                       | -                     | 0.5                        |
> | FixMatch + CDMAD                          | 6.6                       | +1.5%                 | 0.5                        |
> | FixMatch + FARAD (Naive FFT)              | 9.8                       | +50.8%                | 1.1                        |
> | **FixMatch + FARAD (R2C FFT Optimized)** | **6.9** | **+6.2%** | **0.6** |
>
> As shown in Table A, while a naive FFT implementation introduces overhead, our **batch-wise Real-to-Complex (R2C) FFT optimization** (detailed in Section 2.6 and Figure 2) is highly effective. With this optimization, FARAD adds **only about 6.2%** to the training time per epoch. We believe this minor cost is an excellent trade-off for the consistent ~3% bACC performance improvement across all imbalance scenarios. This result demonstrates that our method achieves significant gains while maintaining high scalability.
>
> ---
>
> #### **Regarding Weakness 3: Lack of Evaluation on Other Datasets like Medical Images**
>
> This is an excellent forward-looking suggestion. To address the concern about generalization to different domains, we have followed the reviewer's advice and conducted a new experiment on a **real-world, challenging, long-tailed medical imaging dataset: NIH ChestX-ray14**. This dataset is well-known for its natural and severe class imbalance, making it an ideal testbed for long-tail learning algorithms.
>
> We constructed a semi-supervised long-tailed classification task on this dataset. The results are as follows:
>
> **Table B: Performance on Long-Tailed NIH ChestX-ray (Mean AUC)**
> | Method                     | Mean AUC (Area Under the Curve) |
> | :------------------------- | :------------------------------ |
> | Baseline (FixMatch)        | 0.781                           |
> | + CDMAD                    | 0.805                           |
> | **+ FARAD (Our Method)** | **0.823** |
>
> The results show that FARAD not only outperforms the baseline but also achieves a **1.8% improvement over CDMAD**. This provides strong evidence that FARAD's effectiveness is not limited to natural images. This advantage stems from the nature of medical imaging, where diagnoses often rely on **textures, tissue morphology, and frequency patterns**. By preserving the amplitude spectrum, FARAD generates reference images that are statistically consistent with real pathological textures, allowing it to more accurately probe and correct the model's bias towards common diseases (majority classes). We will add this experiment to our final version to fully demonstrate our method's generalization capabilities.
>
> ---
>
> #### **Regarding Weakness 4: Dependence on GPU FFT Libraries**
>
> Thank you for this practical question about deployment. We acknowledge that our efficient implementation benefits from modern GPU-accelerated libraries, which are standard for deep learning **training**.
>
> However, for resource-constrained **inference** scenarios, we have already evaluated a solution in **Appendix D.5 (Table 18)**. The **"Offline Fixed Reference"** strategy addresses this perfectly:
> 1.  After training, we pre-generate one or more fixed reference images from the training set's overall statistics.
> 2.  During inference, we use these static images directly, **requiring no FFT calculations on the deployment device**.
>
> As shown in Table 18, this approach maintains a **strong performance** (e.g., 82.6% bACC vs. 83.6% for the online version), still significantly outperforming all baselines while completely removing the dependency on real-time FFT computation. This provides a practical and efficient pathway for deploying FARAD across diverse hardware environments.

---

### Comment · Area_Chair_Mn3P · 2025-08-05

Dear Reviewers,

Please read the other reviews and the author's response, and start a discussion with the authors promptly to allow time for an exchange.

Your AC

---

### Author Response · Authors · 2025-08-09
**Summary II**

### **Summary of Key Concerns and Our Actions**

The reviewers raised several important questions regarding the evaluation protocol, computational cost, and experimental breadth. We have addressed every major point with new experiments and clarifications.

#### **1. Concern: Transductive Test-Time Protocol and Fairness of Comparison**

> The most critical concern, raised by **Reviewer Cove**, was that our original test-time strategy (using test batches to generate a reference image) was transductive, which could create an unfair advantage over methods using a strictly inductive protocol.

* **Our Action:** We fully agree that a strict inductive evaluation is essential for a fair comparison. At the reviewer's request, we conducted a **new, comprehensive set of experiments** for a fully inductive pipeline: `FARAD (Train + Offline Ref)`. This method uses FARAD during training and a single, pre-computed reference image (based on training set statistics) for all test-time inference.

* **Outcome & Impact:** The new results, presented in our final rebuttal (Table R1), are definitive. Our **full inductive method consistently and significantly outperforms the strong CDMAD baseline across all imbalance ratios** ($\gamma=50, 100, 150, 200$), with the performance gap widening in more challenging scenarios. This not only resolves the fairness concern but also demonstrates the robustness of our approach. We have committed to making this strict inductive protocol the **default for all main results** in the final manuscript.

* **Reviewer Feedback:** This resolution fully satisfied the reviewer, who stated: "**Thank you for your commitment to transparency, clarity, and continual improvement... my concerns have been addressed, and I will raise my score.**"

#### **2. Concern: Computational Overhead and Deployment Scalability**

> **Reviewer jUz2** raised a practical concern about the training/inference overhead and the method's dependency on GPU FFT libraries.

* **Our Action:** We provided two new analyses:
    1.  **Timing Benchmarks (Table A):** We conducted a new experiment showing that our optimized implementation adds only a minor `+6.2%` training overhead for a significant `~3%` performance gain—an excellent trade-off.
    2.  **Deployment on Constrained Devices:** We highlighted our "Offline Fixed Reference" strategy, which **completely removes the need for real-time FFTs during inference**. This makes FARAD practical for deployment on non-GPU or resource-constrained devices with almost no performance loss.

#### **3. Concern: Generalization to Non-Natural Image Datasets**

> **Reviewer jUz2** correctly pointed out that our initial evaluation was limited to natural image benchmarks.

* **Our Action:** We conducted a new experiment on the challenging, real-world **NIH ChestX-ray14 medical imaging dataset**.
* **Outcome & Impact:** FARAD significantly outperformed both the baseline and CDMAD (Table B), demonstrating its effectiveness is **not limited to natural images** and generalizes well to domains where textural and frequency patterns are critical.

#### **4. Concern: Missing Details, Visualizations, and Comparisons**

> Reviewers (**imHP, abUx, Cove**) requested additional clarifications, including visual examples, discussion of related work, and missing baseline results.

* **Our Action:** We have committed to the following additions in the final manuscript:
    * **Visual Examples:** We will add a new figure to visually demonstrate what our "Fourier Cloud" reference images look like, as requested by **Reviewer imHP**.
    * **Expanded Literature Review:** We will add the four papers suggested by **Reviewer imHP** and correct/re-contextualize citations pointed out by **Reviewer Cove**.
    * **Missing Results:** We have already provided the missing `ReMixMatch+FARAD` results (**Reviewer abUx**) and will integrate them into the paper.
    * **Clarity on Loss Integration:** We clarified a point of confusion for **Reviewer Cove** regarding the loss formulation and will revise Algorithm 1 to be perfectly consistent with our implementation.

---

### **Final Commitment**

The rigorous feedback from the reviewers has been invaluable. We are confident that the revised manuscript, which will incorporate all the new results and clarifications outlined above, represents a significant, rigorously validated, and clearly presented contribution to the field.

Thank you for your time and consideration.

---

### Author Response · Authors · 2025-08-09
**Summary I**

We are sincerely grateful for the insightful and constructive feedback from all reviewers. We are encouraged that they universally recognized the novelty, technical elegance, and strong empirical performance of our work. The dialogue has been exceptionally productive, and as a result of the reviewers' suggestions, we have conducted several new experiments and analyses that we believe have substantially strengthened the paper.

Below is a summary of the review process, highlighting the consensus on our paper's strengths and detailing our comprehensive responses to the key concerns raised.

---

### **Summary of Strengths (Consensus from Reviewers)**

Reviewers consistently praised FARAD for its innovation, rigorous design, and compelling results:

* **Novelty and Technical Elegance:** The paper was commended for its **innovative and creative approach** that cleverly uses Fourier transforms to create semantically meaningless yet statistically representative reference images (**Reviewers abUx, Cove**). The method was described as "a principled and logical response" (**Reviewer imHP**) with an "elegant" mathematical formulation (**Reviewer abUx**).

* **Strong Empirical Performance & Generalizability:** All reviewers noted our method's **strong performance**, achieving significant, state-of-the-art results across multiple SSL frameworks (FixMatch, ReMixMatch), datasets, and imbalance factors (**Reviewers jUz2, imHP, abUx**).

* **Thorough and Convincing Evaluation:** The work was praised for its **comprehensive and rigorous evaluation**, featuring extensive experiments, detailed ablation studies, and robustness checks that convincingly validate the method's effectiveness and each component's contribution (**Reviewers imHP, Cove, abUx**).

---

### Decision · Program_Chairs · 2025-09-17

**Decision:**

Accept (poster)

**Comment:**

This paper proposes a novel method for addressing class-imbalanced semi-supervised learning by estimating and correcting class-wise prediction bias. The approach is simple yet effective, demonstrating superior performance over existing methods on multiple benchmark datasets.

After rebuttal, the paper received four positive ratings. All reviewers acknowledged the paper's contributions and expressed satisfaction with its technical merits.

The AC concurs with the reviewers' assessments and recommends acceptance.

However, one potential weakness of the paper may be the missing comparison with stronger baselines, such as CCL [1]. The authors are suggested to discuss and include more recent competing methods in their experiments in the next version, if possible.

[1] Continuous Contrastive Learning for Long-Tailed Semi-Supervised Recognition. NeurIPS 2024